# MORC2 is a phosphorylation-dependent DNA compaction machine

Winnie Tan [1,2,3], Jeongveen Park [4], Hariprasad Venugopal [5,10], Jieqiong Lou [6,10], Prabavi Shayana Dias [1,7,10], Pedro L. Baldoni [1,2,10], Kyoung-Wook Moon [4,10], Toby A. Dite [1,2,10], Christine R. Keenan [1,2,6], Alexandra D. Gurzau [1,2], Joonyoung Lee [4], Timothy M. Johanson [1,2], Andrew Leis [1,2], Jumana Yousef [1,2], Vineet Vaibhav [1,2], Laura F. Dagley [1,2], Ching-Seng Ang [8], Laura D. Corso [1,2], Chen Davidovich [5], Stephin J. Vervoort [1,2], Gordon K. Smyth [1,9], Marnie E. Blewitt [1,2], Rhys S. Allan [1,2], Elizabeth Hinde [6], Sheena D'Arcy [7], Je-Kyung Ryu [4] ✉ & Shabih Shakeel [1,2,3,6] ✉

The Microchidia (MORC) family of chromatin-remodelling ATPases is pivotal in forming higher-order chromatin structures that suppress transcription. The exact mechanisms of MORC-induced chromatin remodelling have been elusive. Here, we report an in vitro reconstitution of full-length MORC2, the most commonly mutated MORC member, linked to various cancers and neurological disorders. MORC2 possesses multiple DNA-binding sites that undergo structural rearrangement upon DNA binding. MORC2 locks onto the DNA using its C-terminal domain (CTD) and acts as a clamp. A conserved phosphate-interacting motif within the CTD was found to regulate ATP hydrolysis and cooperative DNA binding. Importantly, MORC2 mediates chromatin remodelling via ATP hydrolysis-dependent DNA compaction in vitro, regulated by the phosphorylation state of its CTD. These findings position MORC2 CTD phosphorylation as a critical regulator of chromatin remodelling and a promising therapeutic target.

Chromatin remodelling is essential to maintain cell type-specific function. Chromatin remodellers function by either condensing regions of the genome into transcriptionally inactive heterochromatin or by loosening the chromatin into transcriptionally active euchromatin. To date, the most well-studied chromatin remodelers are the Inositol requiring 80 (INO80), the Chromodomain Helicase DNA-binding (CHD), Switching defective/Sucrose Non-Fermenting (SWI/SNF) and the Imitation Switch (ISWI), all of which possess ATPase activity to remodel chromatin[1–4]. The Microchidia CW-type zinc finger (MORC) proteins are the newest member of the chromatin remodelling family[5]. The number of MORCs varies from species to species, for example, there are four MORCs in human (MORC1-4), seven in

[1]WEHI, 1G Royal Parade, Parkville, VIC 3052, Australia. [2]Department of Medical Biology, The University of Melbourne, Melbourne, VIC 3052, Australia. [3]ARC Centre for Cryo-electron Microscopy of Membrane Proteins, Bio21 Molecular Science and Biotechnology Institute, University of Melbourne, Parkville, VIC, Australia. [4]Department of Physics and Astronomy, Seoul National University, Seoul, Republic of Korea. [5]Department of Biochemistry and Molecular Biology, Monash University, Clayton, VIC 3168, Australia. [6]Department of Biochemistry and Pharmacology, The University of Melbourne, Melbourne, VIC 3052, Australia. [7]Department of Chemistry and Biochemistry, The University of Texas at Dallas, Richardson, TX 75080, USA. [8]The Bio21 Molecular Science and Biotechnology Institute, The University of Melbourne, Melbourne VIC 3052, Australia. [9]School of Mathematics and Statistics, The University of Melbourne, Melbourne VIC 3010, Australia. [10]These authors contributed equally: Hariprasad Venugopal, Jieqiong Lou, Prabavi Shayana Dias, Pedro L. Baldoni, Kyoung-Wook Moon, Toby A. Dite. ✉e-mail: prof.love@snu.ac.kr; shakeel.s@wehi.edu.au

*Arabidopsis thaliana* and one in *Caenorhabditis elegans*[6–8]. MORC proteins are increasingly associated with epigenetic silencing across various eukaryotic organisms[7]. All MORCs share three features: an N-terminal **g**yrase, **h**eat shock protein 90, histidine **k**inase and Mut**L** (GHKL) ATPase domain, a central CW-type zinc finger (CW) domain, and a divergent C-terminal domain (CTD) with predicted coiled coils (CCs)[5]. While the GHKL and CW domains are implicated in transcription suppression and chromatin binding, respectively[9,10], the CC domain is proposed to facilitate MORC's homo-dimerisation[5]. Despite crystal structures of MORC2-4 ATPase domains showing dimerization propensity upon ATP binding or DNA interaction[11–13], an essential requirement for chromatin remodelling by MORCs, the molecular mechanism by which this is achieved remains unclear due to a lack of structural and biochemical data on full-length MORCs.

MORCs are increasingly recognised as oncoproteins, among which MORC2 is frequently mutated, possibly due to its ubiquitous expression in all cell types. MORC2 expression is linked to several cancers, including breast, gastric, liver and colorectal cancers[14–17]. Knockdown of MORC2 inhibited cancer cell proliferation[18], hinting at its role as a tumour suppressor gene. MORC2 variants are also found in neurological disorders like Charcot-Marie Tooth disease. Mice carrying a CMT-related MORC2 S87L heterozygous variant exhibited axonal neuropathy and skeletal muscle weakness, displaying the clinical symptoms expected in human CMT2Z patients[19]. MORC2's impact on different disease type underscores the importance of understanding its molecular mechanisms for therapeutic interventions.

Physiologically, MORC2 is involved in two cellular functions – epigenetic silencing and DNA damage response. In the epigenetic silencing pathway, MORC2 operates within the human silencing hub (HUSH) complex to represses LINE-1 retrotransposons through histone 3 lysine 9 trimethylation (H3K9me3) binding[20]. The neuronal MORC2 mutation, R252W results in reduced ATPase activity and hyperactive transcriptional silencing[11,21], suggesting that ATP hydrolysis impairs HUSH-mediated silencing by promoting chromatin relaxation. In the context of DNA damage response, MORC2 is recruited to DNA damage sites through PARylation by PARP1[22] and phosphorylation at S739 by PAK1 kinase in the CTD[23]. These modifications promote chromatin remodelling and DNA damage response. The importance of MORC2 CTD is underscored by its extensive post-translation modifications (PTMs) including phosphorylation, acetylation, PARylation and SUMOlyation, indicating a mechanistic cooperation between MORC2 PTMs, chromatin remodelling and DNA damage response. So far, only dysregulation of phosphorylation is implicated in disease manifestation, where S739E mutation led to gastric cancer progression[15], suggesting phosphorylation has regulatory function in MORC2 (Fig. 1a). Despite mounting evidence showing PTMs affect MORC2 function, the molecular mechanisms of MORC2-mediated chromatin remodelling remain poorly understood. This is largely because there has been no successful in vitro reconstitution of full-length MORC2 (or other human MORCs) to date. Since all MORCs have similar structural organisation and are implicated in gene regulation[24–26], understanding the molecular function of MORC2 in chromatin remodelling will have an impact on our understanding of pathological implications of other MORC proteins.

Here, using a comprehensive analysis of full-length MORC2 employing structural, biochemical, biophysical, single molecule and cellular approaches, we show that MORC2 function is regulated by its CTD. We report the formation of a symmetric MORC2 dimer and structural rearrangements in all its domains including CTD upon DNA binding. We show that MORC2 possesses multiple DNA-binding sites across its length, with different binding affinities. We reveal MORC2 is a DNA clamp that preferentially binds to open chromatin region. Importantly, our findings reveal that MORC2 causes chromatin remodelling by DNA compaction. The ATP binding and subsequent hydrolysis coupled with DNA binding to its CTD is indispensable for DNA compaction. Surprisingly, alanine mutations of six conserved serine sites in its CTD, which we show are frequently phosphorylated, accelerates ATP hydrolysis-driven DNA compaction, independent of DNA binding. Thus, MORC2 function is driven by ATP hydrolysis and regulated by its CTD phosphorylation. This functional dependency of chromatin remodelling on PTMs in MORC2 could be a common feature not just in MORCs but in all chromatin remodellers.

## Results

### Full-length MORC2 is a dimer

To understand the molecular mechanism of MORCs and how they are regulated, we purified recombinant, functionally active, full-length MORC2 from insect cells (Supplementary Fig 1a) and conducted a comprehensive analysis employing structural, biochemical, biophysical, single molecule, and cellular approaches. Size exclusion chromatography multi-angle light scattering (SEC-MALS) analysis of full-length MORC2 (MORC2^WT) reveals a homodimer of 264 kDa, which differs from the ATPase domain (1-603 amino acids, MORC2^1-603) that dimerises only when an ATP analog, AMP-PNP, is present (Supplementary Fig 1b). SEC-MALS also showed that all MORC2 truncation variants that contained CTD formed dimers (Supplementary Fig 1c), confirming that MORC2 CTD enables its dimerisation. To observe whether MORC2 forms dimers in cells, we mapped and quantified the number of molecules and brightness (NB) of the monomeric eGFP (meGFP)-tagged MORC2 protein in a MORC2 knockout HEK293T cell line using fluorescence fluctuation spectroscopy, which allowed us to measure fluctuation of meGFP fluorescence intensity and meGFP oligomerisation dynamics at the focal adhesions with pixel resolution[27]. We observed about 10 % dimers of MORC2, which increased by ~2-fold upon addition of neocarzinostatin, a DNA damage agent that is known to cause double stranded (ds) DNA breaks (Supplementary Fig 1d-e). These data suggest dimerised MORC2 may be the functional form, at least in DNA repair.

### Recombinant MORC2 is phosphorylated at multiple sites

The purified MORC2 came phosphorylated from insect cells, and the level of phosphorylation further increased upon incubation with PAK1 kinase[23] (Supplementary Data 1). Tandem mass spectrometry (MS/MS) analysis revealed extensive phosphorylation at multiple sites within MORC2, with the top 6 hits S725, S730, S739, S743, S777, and S779, which were further phosphorylated by the PAK1 kinase. Of these residues, all except S725 are evolutionarily conserved and reside within the CTD region, which we coined as phosphate interacting motif (PIM) (Supplementary Fig 2). To understand how phosphorylation may impact MORC2 interaction with DNA, we first determined the minimal length of DNA required for binding. We performed electrophoretic mobility shift assay (EMSA) with MORC2^WT and MORC2^1-603 in the presence of IRDye-700 labelled 10, 14, 19, 24, 29, 34, 39, 44, 60 and 90 bp double-stranded (dsDNA). Both constructs of MORC2 bind to dsDNA longer than 29 bp (Supplementary Fig 3). To evaluate the impact of phosphorylation on MORC2's DNA binding capability, we incubated the purified MORC2 with λ-protein phosphatase (λPPase) or recombinant PAK1 kinase before performing EMSA in the presence of dsDNA. EMSA showed increased retention of dephosphorylated MORC2 on DNA compared to the pre-treated (MORC2 in λPPase and PAK1 kinase reaction buffer) and PAK1-treated MORC2 (Supplementary Fig 4a-b). Additionally, surface plasmon resonance (SPR) analysis showed that dephosphorylated MORC2 binds DNA with >10-fold greater affinity ($K_D = 15.3 \pm 9.1$ nM) than pre-treated ($K_D = 160 \pm 12.5$ nM) and >30-fold greater affinity than phosphorylated MORC2 ($K_D = 493.3 \pm 41.8$ nM) (Supplementary Fig 4c). These results suggest dephosphorylated MORC2 may be a stronger DNA binder.

### Full-length MORC2 undergoes structural changes upon DNA binding

As a MORC2 neuropathic mutant S87L was shown to have residual ATP hydrolysis activity[11], we have identified and utilised ATP hydrolysis

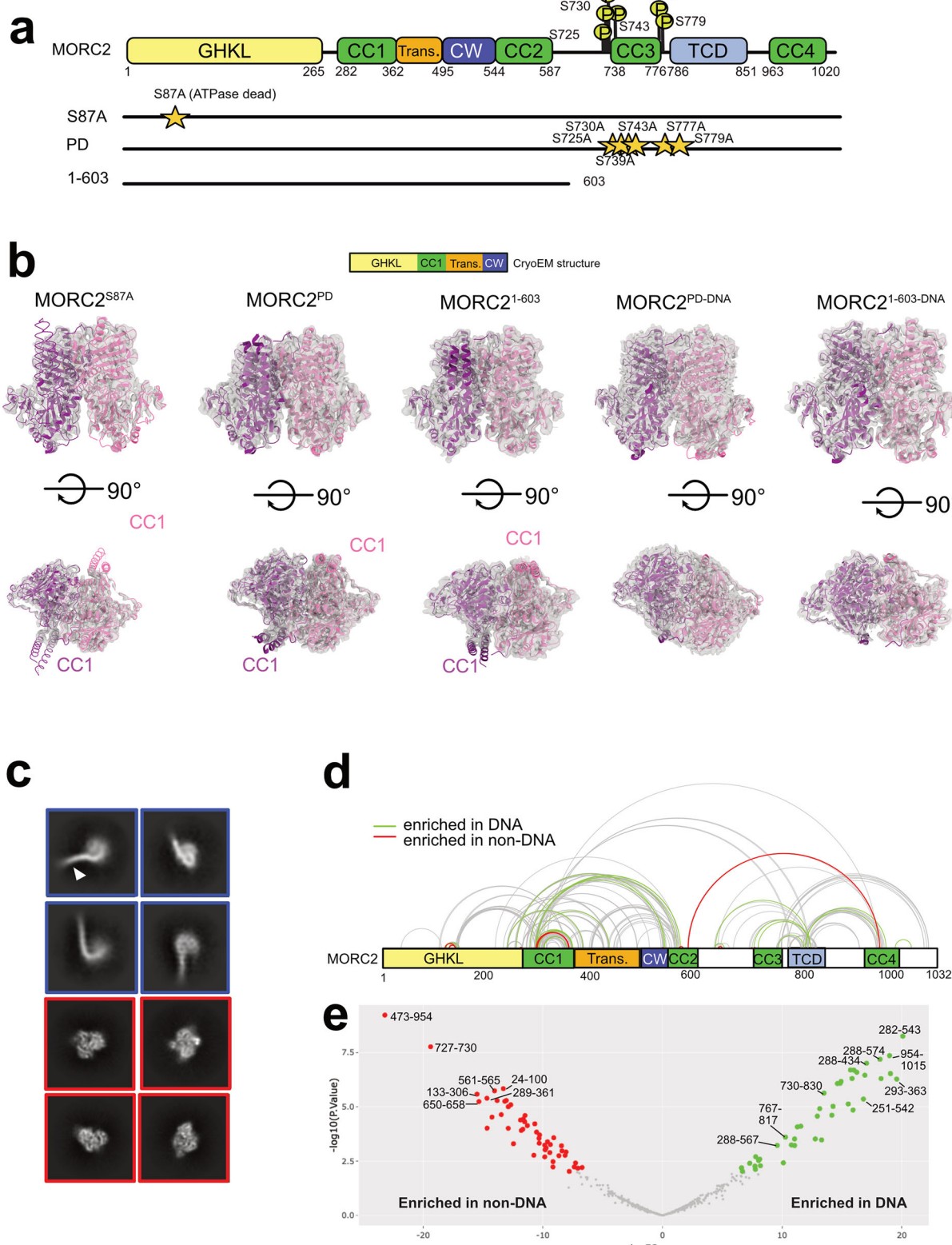

mutant S87A. We determined a 2.4 Å cryo-electron microscopy (cryoEM) structure of full-length MORC2 ATPase-dead mutant (MORC2^S87A) in the presence of AMP-PNP (Fig. 1a, b, Supplementary Fig 5a-d). The AMP-PNP density was visible in the cryoEM map and at the same location as found in the crystal structures (Supplementary Fig 5e). The presence of AMP-PNP density near the ATP lid in MORC2^S87A model, confirmed that this mutant can bind but not

hydrolyse ATP, mimicking S87L variant where some residual ATPase activity is still detected (Supplementary Fig 5f). S87A mutant had similar DNA binding affinity as wild-type MORC2 (Supplementary Fig 5g-h), indicating that ATP binding did not affect MORC2 DNA binding.

To determine how phosphorylation of MORC2 affects DNA binding, we purified a MORC2 phosphodead (MORC2^PD) mutant

**Fig. 1 | DNA binding triggers structural changes in full-length MORC2. a** Domain diagram of MORC2 constructs used in this study. The GHKL indicates Gyrase, Hsp90, Histidine Kinase, MutL domain, CC1-4 indicates coiled coil domains, Trans indicates Transducer-like domain, CW indicates CW-type zinc finger domain and TCD indicates Tudor-chromodomain. Phosphorylation sites studied in this work are marked as 'P' with the residue number indicated. **b** The domain diagram on top shows the extent of the model built in MORC2 cryoEM maps. The cryoEM structures of MORC2 S87A (MORC2[S87A]) mutant, phosphodead (MORC2[PD]) mutant, MORC2 ATPase (MORC2[1-603]), phosphodead MORC2 with DNA (MORC2[PD-DNA]), and MORC2 ATPase with DNA (MORC2[1-603-DNA]). Coiled coil 1 is marked as CC1, which is only seen in DNA-free samples. Pink and purple colour models fitted in the cryoEM maps indicate one protomer each of the MORC2 homodimer. **c** Selected 2D reference-free class averages of MORC2[PD-DNA]. The blue box shows the 2D classes where DNA (marked by white arrowhead in one of the classes) is seen bound to fuzzy MORC2 density. The red box indicates high-resolution 2D classes where secondary structure elements are clearly visible. **d** Map of quantified crosslinks in MS analysis of MORC2 DNA and non-DNA bound samples. Crosslinks with a similar abundance in DNA and non-DNA (grey lines), crosslinks considered significantly enriched (adjusted $p$-value ≤ 0.05) in DNA (green lines) and in non-DNA (red lines) are shown. **e** Volcano plot of quantified crosslinks, where the log2 DNA/non-DNA fold changes are plotted against the -log10 p-value. Crosslinks that were considered significantly enriched (adjusted $p$-value ≤ 0.05) in DNA (green) and in non-DNA (red) are highlighted; some of the top hits have the peptide residue numbers indicated. We adjusted for multiple comparisons with Benjamini–Hochberg (BH) correction and the statistical test was two-sided moderated t-test. The full list of all enriched crosslinks is in Supplementary Data 1.

comprising S725A, S730A, S739A, S743A, S777A and S779A mutations (Fig. 1a, Supplementary Fig 1a). We determined the structures of MORC2[PD] and MORC2 ATPase domain with (MORC2[PD-DNA], MORC2[1-603-DNA]) and without DNA (MORC2[PD], MORC2[1-603]), in presence of AMP-PNP, at overall resolutions of 1.9, 2.5, 2.4 and 3.2 Å, respectively, using cryoEM (Fig. 1b, Supplementary Fig 6, Supplementary Data 2). Although there were 2D classes where MORC2 was bound to the DNA in the MORC2[PD-DNA] samples (Fig. 1c), these could not be resolved into a 3D structure due to low resolution and absence of diverse views, suggesting that there may be an increase in flexibility upon DNA binding despite similar global resolution observed in DNA or DNA-free samples (Supplementary Fig 6b)[11].

The full-length WT MORC2 displayed strong preferred orientation where the symmetric dimer view is conspicuously absent despite the presence of the AMP-PNP, indicating that phosphorylation at the CTD may affect flexibility of MORC2 dimer (Supplementary Fig 7a-b, view of symmetric dimer is marked with red box in panel b). We used crystal structures of MORC2[1-603] protein (PDB:5OF9 and 5OFB)[11] to build models into our cryoEM maps. The dimerised ATPase domain is well-resolved in all the cryoEM maps, but no density is seen for CTD to model in the full-length MORC2[PD], MORC2[PD-DNA] and MORC2[S87A] maps. Upon overlaying these models, we observed no conformational changes within the ATPase domain. Interestingly, almost all CC1 was missing in the DNA-bound cryoEM structures indicating high flexibility of this region upon DNA binding (Fig. 1b, Supplementary Fig 6c). Overall, these structures show that the core of the ATPase domain does not undergo large conformational changes in the presence of DNA except CC1, and that MORC2 CTD is flexible.

As DNA binding led to an apparent loss of density corresponding to MORC2 CC1 domain, we employed quantitative crosslinking mass spectrometry to interrogate the conformational changes upon DNA binding. We used a high density crosslinker, sulfosuccinimidyl-4,4'-azipentanoate ('sulfo-SDA'), where its NHS ester group binds to primary amines and the diazarine ring to the side chain of any residue within its vicinity. We crosslinked full-length MORC2 on its own (MORC2[Apo]), in the presence of AMP-PNP with DNA (MORC2[AMPPNP-DNA]) and without DNA (MORC2[AMPPNP]).

We identified 337 inter- and intra- crosslinks in the MORC2[Apo], 214 in MORC2[AMPPNP], and 359 in MORC2[AMPPNP-DNA], at 5% False Discovery Rate (FDR, Supplementary Data 1). We used the AlphaFold2 multimer-predicted models for full-length MORC2 dimer. Model 1 closely resembles the MORC2 crystal structure (PDB:5OF9) and was used for validation by crosslinking mass spectrometry data (Supplementary Fig 7c-e). We used XMAS[28], which identifies homodimeric interaction based on peptide sequence overlap, to map the crosslinks from MORC2[AMPPNP] and MORC2[AMPPNP-DNA] on Model 1 within the proximity of 20 Å, which is the expected median length for sulfo-SDA crosslinker (Supplementary Fig 7f-g, Supplementary Data 1). There were 54 intermolecular crosslinks identified between the two protomers in MORC2[AMPPNP] and 56 in MORC2[AMPPNP-DNA]. There were 5 crosslinks between the CTD (residues 904-1029) from the two protomers in both

samples, which are paired next to each other in the model, thus confirming that CTD forms dimeric coiled coils in presence or absence of DNA (Supplementary Data 1). A large cluster of crosslinks was observed within the ATPase domain, indicating that it is the most well folded domain across all the regions of MORC2, consistent with our cryoEM structures where we were able to resolve the ATPase domain, with the rest being too flexible.

We quantified the crosslinks for a statistical-based comparison of MORC2[Apo], MORC2[AMPPNP] and MORC2[AMPPNP-DNA] conformational changes (Supplementary Data 1). Between MORC2[Apo] and MORC2[AMPPNP], there were 93 common crosslinks, 30 unique crosslinks in MORC2[Apo] and 47 in MORC2[AMPPNP]. These unique crosslinks appeared throughout the length of MORC2, indicating that ATP-dependent MORC2 dimerisation leads to increase in proximity of two protomers (Supplementary Fig 8a-b). Without ATPase dimerisation, the CC1 domain interacts with the CC3 domain (Supplementary Fig 8a). However, upon dimerisation, CC1 contacts the TCD domain, as evidenced by enrichment of crosslinks in the AMP-PNP sample (Supplementary Fig 8a, Supplementary Data 1). The most striking domain-level changes involved crosslinks in the GHKL domain near S87, ATP binding site where we observed 32 crosslinks in the GHKL domain in presence of AMP-PNP (Supplementary Fig 8c, Supplementary Data 1). In contrast, the MORC2[AMPPNP] and MORC2[AMPPNP-DNA] share many crosslinks suggesting that the two complexes adopt similar conformations. However, by focusing on significantly enriched crosslinks between MORC2[AMPPNP] and MORC2[AMPPNP-DNA], the abundance of some crosslinks differed, suggesting subtle changes in MORC2 conformation upon DNA binding (Fig. 1d, e). MORC2[AMPPNP] has long range contacts including 176-749, 290-561 and 473-954, which disappear in presence of DNA, indicating re-organisation of contacts between ATPase and CTD regions (Fig. 1d). Particularly, consistent with our cryoEM studies, the residues within CC1 make different contacts in presence or absence of DNA. By mapping the crosslinks on the MORC2 Model 1 we found that the median crosslinking length was about 22 Å, where 38 (out of 77) crosslinks were within 20 Å length in MORC2[AMPPNP] and 34 (out of 68) in MORC2[AMPPNP-DNA] (Supplementary Fig 7g, Supplementary Data 1, Supplementary Movies 1-2). We observed a loss of crosslinks within the dimeric interface upon DNA binding and formation of crosslinks between CC1 region and the CW-CC2 domains, indicating that MORC2 CC1 engages with a patch on the CW-CC2 domain that was distal to the dimerisation interface mediated by AMP-PNP (Supplementary Fig 7f **inset**). Enrichment of these crosslinks suggests that the CC1 region is more accessible in the absence of DNA.

## MORC2 has multiple DNA binding sites

To further characterise MORC2 DNA binding, we performed hydrogen-deuterium exchange mass spectrometry (HDX) on MORC2[1-603] and MORC2 CTD (MORC2[496-1032]) with and without DNA in the presence of AMP-PNP (Supplementary Data 2 **and** Supplementary Table 4). In MORC2[1-603], we observed a significant increase in deuterium uptake in the ATP lid upon DNA binding (Fig. 2a, peptides 81 to 109), indicating an

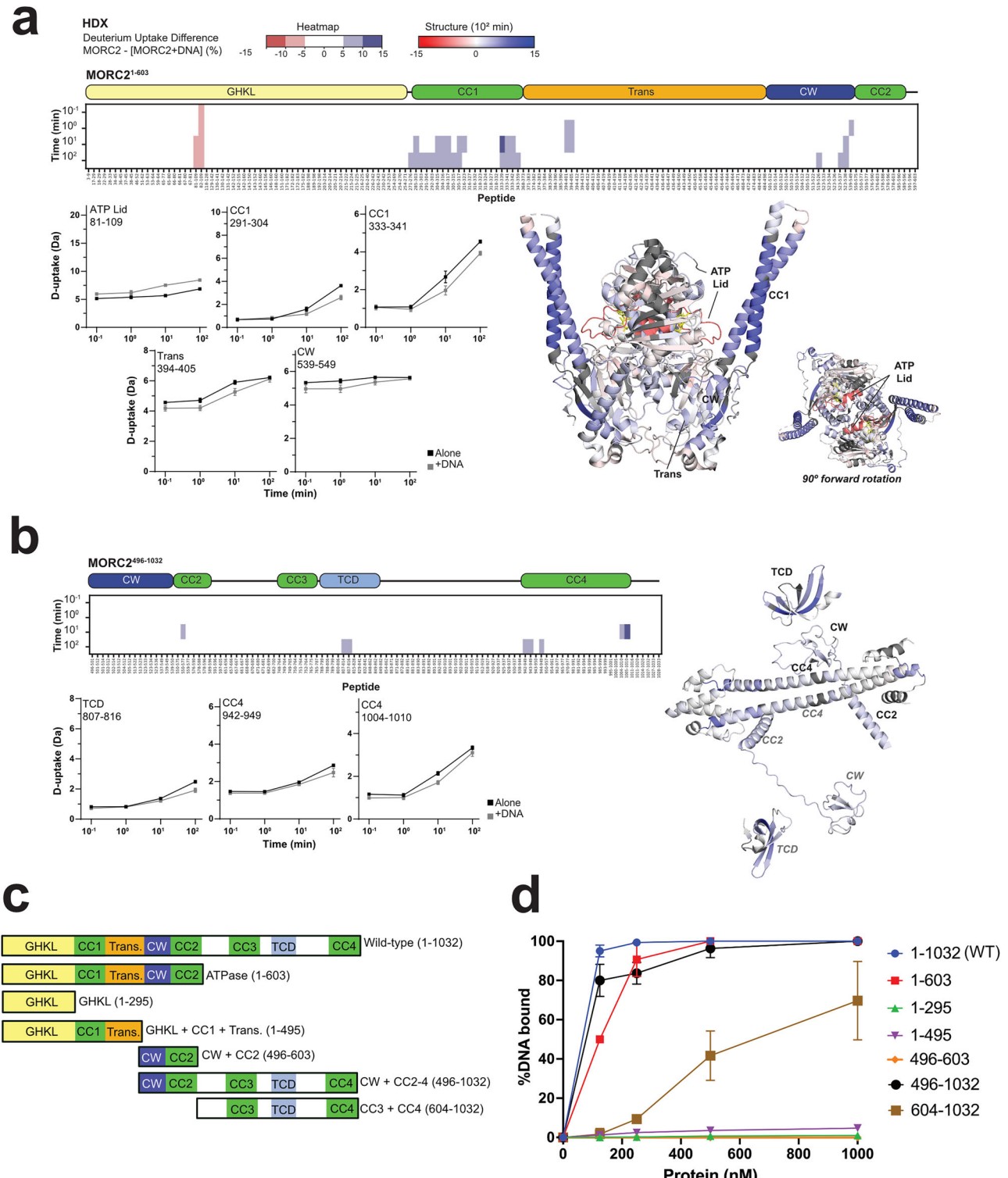

**Fig. 2 | There are multiple DNA binding sites, with different affinities, across MORC2. a** Heatmap showing the difference in deuterium uptake between MORC2$^{1-603}$ alone and with three-fold excess 60 bp dsDNA (top). Deuterium uptake plots for example peptides (left). The MORC2 ATPase crystal structure (PDB: 5OF9) coloured by difference after $10^2$ min exchange based on DynamX residue-level scripts without statistical filters (right). Residues without coverage are grey. **b** Heatmap showing the difference in deuterium uptake between MORC2$^{496-1032}$ alone and with three-fold excess 60 bp dsDNA (top). Deuterium uptake plots for example peptides (left). The MORC2$^{496-1032}$ Alphafold Model 1 coloured by difference after $10^2$ min exchange based on DynamX residue-level scripts without statistical filters (left). Residues without coverage are grey. For the model, only residues with >40 confidence

are shown. For **a** and **b**, blue/red heatmap colouring indicates a difference ≥5% with a $p$-value ≤ 0.01 in Welch's one-sided t-test (number of replicates ($n$) =3). Plots show deuterium uptake for representative peptides for MORC2 alone (black) or with DNA (grey). Error bars are mean ± 2 standard deviation (SD, $n$ = 3) and the y-axis is 80% of the maximum theoretical deuterium uptake, assuming the complete back exchange of the N-terminal residue. Source data are provided as a Source Data file. **c** Schematic for MORC2 constructs used in DNA binding EMSA. **d** Quantification of percentage of 60 bp dsDNA bound to MORC2 WT (1-1032), ATPase (1-603), GHKL (1-265), GHKL + CC1 (1-495), CW domain (496-603), CW + CC1 + CTD (496-1032) and CCW + CTD (604-1032). The points are shown as mean +/− standard deviation; number of independent experiments ($n$) = 3. Source data are provided as a Source Data file.

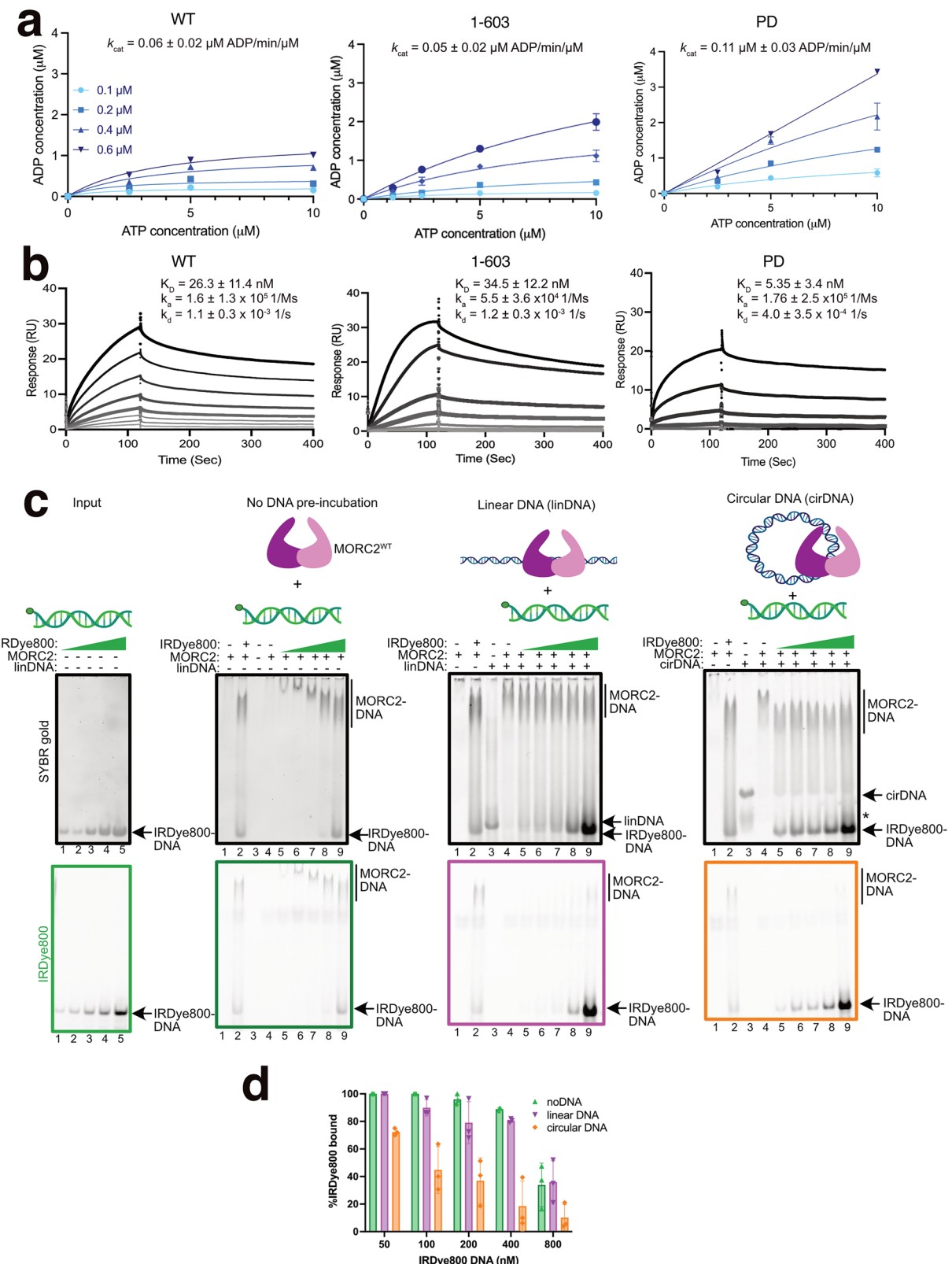

increased flexibility in the ATP lid region. Our MORC2[1-603] cryoEM structure and the previous crystal structure (PDB:5OF9)[15] showed that upon AMP-PNP binding, the ATP lid covers the active site and modulates rotation of CC1, which contributes to stability of MORC2 ATPase dimerisation interface. Our ATPase-dead S87A mutant (Supplementary Fig 5e) and S87L Charcot-Marie-Tooth patient mutant[15] structures also show a change in the ATP lid. Thus, the DNA binding may reduce the stability of the ATP lid, leading to increased flexibility of the MORC2 dimer, which may have functional implications in MORC2 chromatin remodelling.

The presence of DNA with MORC[1-603] also reduced deuterium uptake in CC1 and nearby regions, transducer-like and CW. This suggests CC1 and nearby regions form a high-affinity DNA binding site, somewhat consistent with previous mutagenesis of the CC1 loop (point mutation of R326, R329 and R333) that prevented DNA

**Fig. 3 | MORC2 acts as a DNA clamp with phosphorylation-dependent ATPase activity. a** In vitro Fluorescence Polarisation ATPase activity of MORC2 WT, ATPase (1-603) and PD mutant. Individual measurements ($n$ = 3) are shown as points, and the solid line represents the non-linear fit of the data. The indicated $k_{cat}$ values are mean ± standard error of mean (SEM) representative of three experiments. **b** Surface plasmon resonance analysis of MORC2 WT, ATPase (1-603) and PD mutant with protein concentrations of 4, 8, 16, 33, 63, 125, 250 and 500 nM. Values for $K_D$ (equilibrium dissociation constant), $k_a$ (association rate constant) and $k_d$ (dissociation rate constant) are shown as mean ± SEM, representative of three experiments. **c** Schematic of competition EMSA with input IRDye800dsDNA (column 1), pre-incubated with no DNA (column 2), 101 bp linear (column 3) and 101 bp circular DNA (column 4) is shown on top. The 100 nM MORC2 WT protein was pre-

incubated with no DNA, 200 nM 101 bp linear or 200 nM 101 bp circular DNA for 30 min, followed by 10 min incubation with 50, 100, 200, 400 or 800 nM 90 bp IRDye800-labelled dsDNA. The protein to IRDye800-labelled DNA concentration (nM) in controls (lane 2 in Column 2, Column 3 and 4) is 100:800. The reaction is resolved on 6% PAGE gel. SYBR gold (top) and IRDye800 (bottom) channel images are shown. *indicates background circular DNA bands. The MORC2 protein and linear and circular DNA cartoons were created in BioRender. Tan, W. (2025) https://BioRender.com/xfcicda. **d** Quantification of the IRDye800-labelled DNA bound to MORC2, calculated as described in Supplementary Fig 10a. The points are shown as mean ± standard deviation (SD) from three independent experiments; number of independent experiments ($n$) = 3. For Fig. 3a-d, source data are provided as a Source Data file.

binding[11]. We saw only modest decreases in deuterium uptake in the CC1 loop (below our significance thresholds) as this loop is disordered and reached maximal deuteration at our initial timepoint. The reduced deuterium uptake in CC1 and CW may also be attributed to an intramolecular interaction induced by DNA, consistent with our cross-linking experiments (Fig. 1d, e). Notably, we did not see a DNA-induced change in deuterium uptake in the CW when CC1 was not present in MORC2[496-1032]. For MORC2[496-1032], we saw reduced deuterium uptake in the TCD and at opposite ends of CC4 (Fig. 2b), hinting that the CTD contains secondary, low-affinity DNA binding sites.

To further investigate our HDX observations, we performed EMSA with various MORC2 truncations, in the presence of 60 bp dsDNA (Fig. 2c). All constructs that lacked CW and CC2 showed lower DNA binding in comparison to the full-length MORC2 (Fig. 2d, Supplementary Fig 9a). The constructs containing only GHKL and CC1 domains, or only CW and CC2, showed negligible DNA binding, whereas some DNA binding was rescued in the 604-1032 construct that lacks CC1-2. Taken together, the HDX and EMSA data show that CC1 DNA binding requires CW-CC2, and that CC1 and CW-CC2 alone are insufficient for DNA binding. We further tested the DNA binding ability of isolated CC2, CC3 and CC4. Interestingly, CC2 (544-587), and CC4 (963-1020) domains were able to bind DNA but not the CC3 region that contains the PIM domain (Supplementary Fig 9b-c). This is consistent with the HDX data where the CW adjacent to CC2 has measurable DNA binding (Fig. 2b). Thus, this work confirms that MORC2 has multiple DNA binding sites in the ATPase and CTD domains.

## MORC2 ATPase activity is phosphorylation-dependent

The rate of ATP hydrolysis by chromatin remodellers may have an impact on chromatin regulation[12,13]. As the MORC2 CTD is extensively post-translationally modified, we hypothesised that MORC2 phosphorylation may affect its function. We performed Fluorescence Polarisation ATPase assays on MORC2[WT], MORC2[1-603] and MORC2[PD]. MORC2[WT] and MORC2[1-603] were able to hydrolyse ATP at a turnover rate ($k_{cat}$) of 0.06 ± 0.02 μM ADP/min/μM and 0.05 ± 0.02 μM ADP/min/μM, which is similar to other GHKL ATPases[12,13,29]. Surprisingly, MORC2[PD] hydrolysed ATP ~2-fold faster than MORC2[WT] at $k_{cat}$ of 0.11 ± 0.03 μM ADP/min/μM (Fig. 3a), and addition of DNA did not alter MORC2 ATPase activity (Supplementary Fig 9d-e). To assess the DNA binding activity of MORC2, we first performed EMSA showing that MORC2[PD] mutant binds better than MORC2[WT] (Supplementary Fig 4d). Next, we quantified this DNA binding by SPR on MORC2[WT], MORC2[1-603] and MORC2[PD] mutant. We found all MORC2 variants were able to bind to 60 bp DNA within a nanomolar affinity range (Fig. 3b). MORC2[PD] mutant has ~5-fold increase in affinity to DNA ($K_D$ = 5.35 ± 3.4 nM) compared to wild-type ($K_D$ = 26.3 ± 11.4 nM (Fig. 3b). Together, these data show that the MORC2 CTD, which harbours the phosphorylation sites, regulates the ATP hydrolysis activity and DNA binding of MORC2.

## Full-length MORC2 acts as a DNA clamp

Given that the SPR experiments showed a remarkably low off-rate of MORC2 from DNA (Fig. 3b, Supplementary Fig 4c), we investigated

whether it locks onto DNA as has been proposed previously for MORC3[30] and MORC1[31], respectively. We incubated full-length MORC2 (MORC2[WT]) or its ATPase domain (MORC2[1-603]) with two-fold molar excess of 101 bp linear dsDNA or its corresponding circular dsDNA at room temperature for 30 minutes to promote occupancy of all available DNA binding sites. Following incubation, we assessed the ability of a 90 bp IRDye800-labeled dsDNA (competitor DNA) to compete with the pre-incubated DNA. The extent of competitor DNA uptake in MORC2-DNA complexes was quantified by comparing it to DNA-only controls via titration against different concentrations of IRDye-labeled linear DNA (Fig. 3c, Supplementary Fig 10).

Our results demonstrate that when MORC2[WT] is preincubated with unlabeled linear DNA, a secondary DNA species is able to effectively occupy more binding sites across all tested concentrations. In contrast, when MORC2[WT] is preincubated with circular DNA, the secondary DNA species binds to a much lesser extent. A similar trend was observed in the case of MORC2[1-603], albeit with a difference in outcompeting the circular DNA, where the uptake of the secondary DNA species was consistently higher than for MORC2[WT]. This suggests that MORC2[WT] exhibits different binding dynamics depending on DNA topology. Given that MORC2[WT] has multiple DNA-binding sites, linear DNA dissociates more readily from regions with lower binding affinity, allowing competitor DNA to bind to them, and/or the MORC2 is binding multiple DNA molecules. Conversely, in the presence of circular DNA, MORC2[WT] appears to clamp onto the DNA, making it more resistant to competition by the IRDye-labeled linear DNA.

These findings suggest a mechanism wherein MORC2[WT] engages in dynamic interactions with linear DNA but forms a more stable, possibly topologically constrained, complex with circular DNA. This is likely due to the additional, HDX-detected DNA binding sites on the CTD being responsible for formation of a tight clamp.

## MORC2 targets accessible chromatin sites

To identify the preferred genomic regions of MORC2 binding, we performed chromatin immunoprecipitation sequencing (ChIP-seq) in the HEK293T cell line utilising antibodies specific for MORC2 and H3K9me3, a mark of constitutive heterochromatin. MORC2 is known to associate with the HUSH complex that is a H3K9me3 reader[21]. MORC2 binding sites have previously been reported to be enriched at promoters in the HeLa cell line[21]. Therefore, we performed assays for transposase-accessible chromatin with sequencing (ATAC-seq) to determine if MORC2 was indeed binding open chromatin. We identified 15,366 MORC2 binding sites in HEK293T cells of which 8,317 correlated with gene promoters according to ChIP-seq (Fig. 4a). The majority of such regions were accessible in ATAC-seq and depleted of H3K9me3 (Fig. 4b, c). In contrast, 3095 MORC2 peaks overlapped H3K9me3 outside of promoter regions, indicating binding in heterochromatin (Supplementary Fig 11a-b) and hinting a weaker colocalization between MORC2 and H3K9me3 regions. To further explore the global binding profile of MORC2, we examined the relationship of MORC2 binding sites with other genomic locations such as repetitive elements including LINE-1 elements and endogenous retroviruses

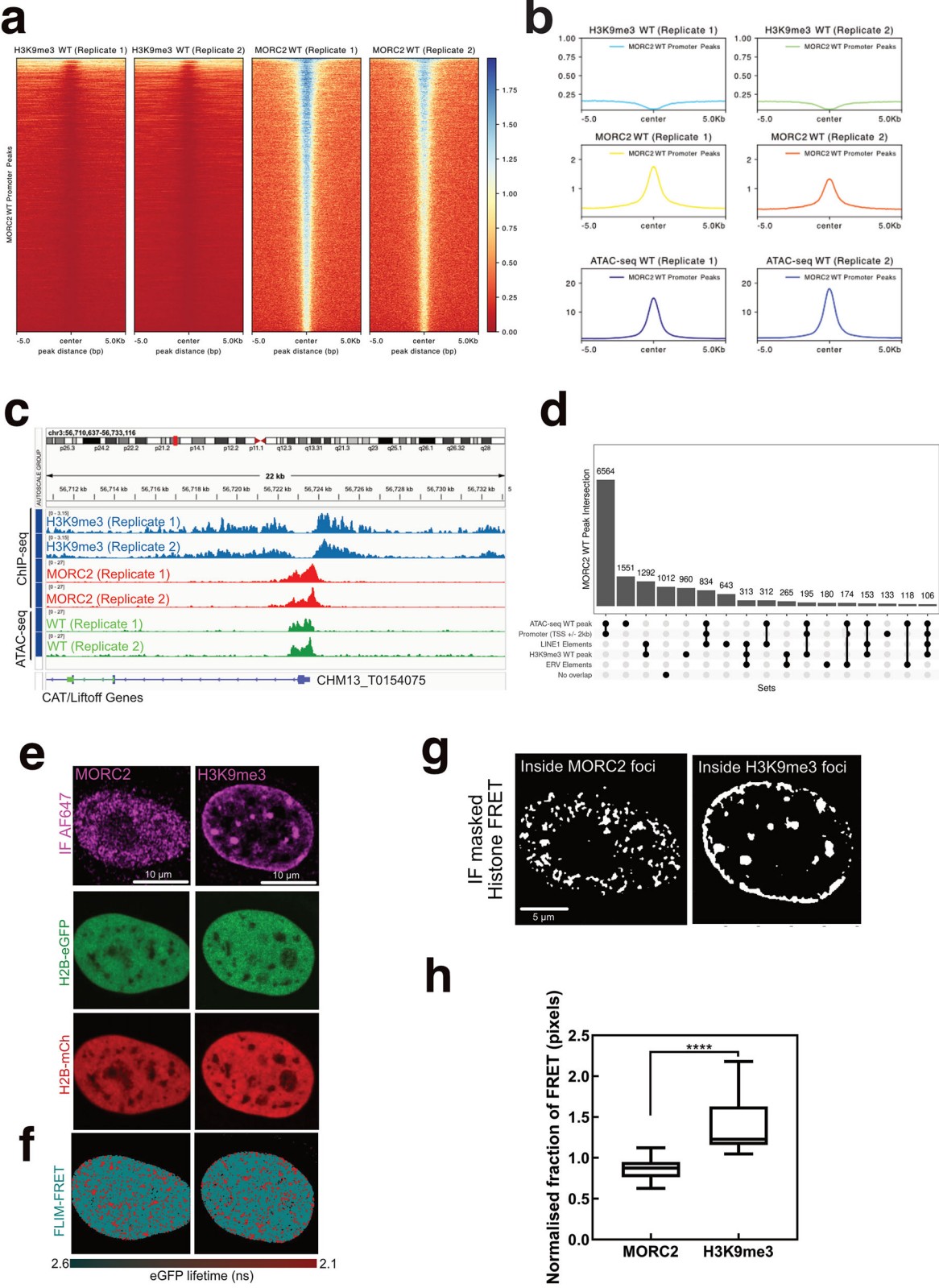

(ERV) where MORC2 is also known to play a role[21]; however, we found limited binding at these repetitive elements (Fig. 4d). A similar observation was reported previously, where out of 4500 MORC2 peaks, only 490 peaks overlapped with H3K9me3[21], suggesting a non-canonical epigenetic silencing pathway mediated by MORC2.

To investigate the compaction of the chromatin environment that MORC2 foci colocalise with in an intact cell, we used the phasor approach to fluorescence lifetime imaging microscopy (FLIM) of Förster resonance energy transfer (FRET) between fluorescently labelled histones (H2B-eGFP and H2B-mCh)[32,33] and immunofluorescence (IF) of MORC2 versus H3K9me3[34] (Fig. 4e, f). Quantification of MORC2 foci localisation throughout recovered histone FRET maps shows that MORC2 foci localise with areas of lower histone FRET compared to the surrounding nucleoplasm, which indicates an association with the

**Fig. 4 | MORC2 occupies open chromatin regions. a** Heatmaps showing H3K9me3 and MORC2 ChIP-seq coverage of promoter-overlapping MORC2 WT peaks (total number of peaks = 8317), sorted by MORC2 intensity. **b** Mean coverage of H3K9me3 and MORC2 ChIP-seq and ATAC-seq signals from 5 kb upstream of promoter-overlapping MORC2 WT peaks (total number of peaks = 8317) to 5 kb downstream. **c** Representative Integrative Genomics Viewer (IGV) image of H3K9me3 and MORC2 ChIP-seq and ATAC-seq tracks in HEK293T cells showing chr3:56,710,637-56,733,116. For (**a–c**), two biological replicates from ChIP- and ATAC-seq assays were used for visualization purposes. **d** Upset plot of MORC2 WT peaks showing the number of peaks overlapping H3K9me3 peaks, promoter regions and ATAC-seq peaks. The number of peaks without any overlap is also presented. **e** FLIM-FRET analysis of HEK293T co-expressing H2B-eGFP and H2B-mCh, with MORC2 and H3K9me3 immunofluorescence labelled with Alexa Fluor dye 647 (AF647). **f** The corresponding pseudo-coloured maps show chromatin compaction, where red colour represents compact chromatin region. **g** Masks based on MORC2 or H32K9me3 intensity image mapped to H2B FRET map. **h** The ratio of the histone FRET fraction (indicative of compact chromatin) inside versus outside regions of high-intensity MORC2 or H3K9me3, as illustrated in panel g are plotted. For (**e–h**), Number of cells ($N$) = 24 cells for MORC2 and $N$ = 17 cells for H3K9me3, two biological replicates. The box and whisker plot in panel h shows the minimum, maximum, and sample median where **** indicates $P < 0.0001$, unpaired two-sided t-test. Source data are provided as a Source Data file.

most open chromatin regions that are present (Fig. 4g). In contrast, H3K9me3 IF shows that H3K9me3 foci colocalise with areas exhibiting high FRET, similar to our previous findings in HeLa cells[34] (Fig. 4h). This is consistent with our genomic studies[33], where the majority of MORC2 does not colocalise with H3K9me3, hinting a chromatin remodelling function of MORC2 independent of HUSH epigenetic silencing.

As MORC2 is predominantly localised at promoter regions, it may play a role in regulating gene transcription. To ask how MORC2 regulates gene expression, we performed RNAseq analysis on MORC2[WT] vs MORC2[KO] (knockout) HEK293T cells (Supplementary Fig 11c), using QL F-tests to compare gene-wise expression levels between MORC2[KO] and MORC2[WT] samples. We detected 97 up-regulated genes and 5 down-regulated genes in the MORC2[KO] compared with the MORC2[WT] sample (Supplementary Fig 11d), indicating that MORC2 is a localised silencer of the genome and does not cause large changes in gene expression. To assess the effect of MORC2[KO] on chromatin globally, we also perform ATAC-seq on MORC2[KO] cells. We found that 17 genes were differentially accessible, with 14 genes up-regulated in the MORC2[KO] sample (Supplementary Fig 11e), including several zinc finger (ZNF) genes that were reported previously[21]. Together, these findings suggest that MORC2 preferentially binds to open chromatin region where H3K9me3 is depleted.

To comprehensively evaluate the global impact of MORC2 overexpression on chromatin architecture, we conducted ATAC-seq analyses on HEK293T cells overexpressing MORC2-WT-meGFP and its variants. Cells overexpressing wild-type MORC2 exhibited minimal changes in chromatin accessibility (Supplementary Fig 11f). To further elucidate the role of MORC2 phosphorylation in chromatin compaction, we expanded our ATAC-seq analyses to include MORC2-WT-meGFP, MORC2-S6A-meGFP (phosphodead), and MORC2-S6D-meGFP (phosphomimetic) mutants. This comparative approach revealed 113 differentially accessible genes between the S6A and WT variants, while only 6 genes showed differential accessibility between S6D and WT (Supplementary Fig 11g). These findings suggest that the dephosphorylation of MORC2 may play a role in promoting chromatin compaction.

## MORC2-induced DNA compaction drives chromatin remodelling

Among all MORCs, MORC2 is most similar to MORC1 based on domain organisation. The *C. elegans* MORC1 has been shown to compact DNA[31], and based on its similarity to MORC2, the same could be plausible for the latter. To test this, we performed a single-molecule DNA compaction fluorescence assay by adding MORC2[WT], MORC2[PD] (phosphodead), MORC2[S87A] (ATP hydrolysis deficient), MORC2[N39A] (ATP binding deficient), MORC2[1-603] or MORC2[603-1032] proteins to a double biotinylated λ-DNA immobilised on a coverslip (Fig. 5a–g). In the presence of MORC2[WT] or MORC2[PD], punctae (formed by DNA compaction) appeared across the length of λ-DNA confirming that MORC2 compacts DNA (Fig. 5d–f, Supplementary Movies 3, 4). Applying an in-plane side flow[35] revealed that these formations were stably condensed clusters (Fig. 5b and Supplementary Movie 4), rather than DNA loop extrusion process. This indicates that MORC2

compacts DNA through a different mechanism than structural maintenance of chromosome (SMC) protein complexes, which achieve compaction by extruding DNA loops[36,37]. Even at lower MORC2 concentrations (0.1 – 5 nM) where DNA compaction transition occurs, MORC2 did not show a DNA loop. The punctae in full-length MORC2 constructs were absent in the CTD lacking MORC2[1-603] or absent in the N-terminal Domain lacking MORC2[604-1032] constructs, showing that MORC2 CTD and NTD are indispensable for its DNA compaction function (Fig. 5e, f). Surprisingly, both the lag time—measured from the point of flow injection to the onset of DNA compaction—and the compaction time—determined by exponential fitting of the cluster intensity increase—were approximately three times shorter for the MORC2[PD] mutant compared to MORC2[WT], indicating that the MORC2[PD] mutant compacts DNA three times faster than the wild-type (Fig. 5e–f, Supplementary Fig 12a–e, Supplementary Movies 3, 4). To directly assess the impact of phosphorylation, we further tested DNA compaction using dephosphorylated MORC2 (λ-phosphatase treated, LPP) and phosphorylated MORC2 (PAK1-treated) (Fig. 5h, i). The dephosphorylated MORC2 exhibited a ~3-fold faster compaction rate than MORC2[WT], whereas PAK1 treatment inhibited DNA compaction (Fig. 5j). These results confirm that phosphorylation of the PIM motif in the MORC2 CTD is a critical determinant of DNA compaction speed.

Next, we investigated the role of ATP in driving DNA compaction by MORC2. For this, we tested MORC2[S87A] and MORC2[N39A] mutants (Fig. 5d–f, Supplementary Fig 12, Supplementary Movies 6, 7). The lag time for DNA compaction by MORC2[S87A] was 3.5 times longer than that of MORC2[WT], with an eightfold increase in compaction time (Fig. 5e, Supplementary Fig 12d-e). Notably, the ATP-binding-deficient mutant (MORC2[N39A]) failed to compact DNA altogether (Fig. 5f). A 2D diagram plotting the DNA compaction kinetics against ATPase activity for various mutants and MORC2[WT] showed a strong correlation ($R^2$ = 0.93), providing compelling evidence that DNA compaction is driven by ATPase activity (Fig. 5g, Supplementary Movie 6). Therefore, we conclude that both ATP binding and hydrolysis are essential for MORC2-mediated DNA compaction.

To further verify these observations, we used atomic force microscopy (AFM) to image the MORC2-DNA complexes. We found that MORC2[WT] compacted DNA into a large DNA cluster, whereas MORC2[S87A], MORC2[N39A], MORC2[1-603] and MORC2[604-1032] variants did not (Fig. 5k-p, Supplementary Movies 8, 9). Clearly, the clusters were organised by numerous DNA loops, suggesting multiple MORC2 molecules exhibit DNA-trapping behaviour, similar observation were made from the AFM images of yeast cohesin SMC complex in presence of DNA[35,38]. We next focused on whether DNA binding causes structural changes in MORC2. The volume distribution analysis of MORC2 from the AFM images revealed that approximately 70% of MORC2 molecules exist in a dimeric form, formed through the engagement of two CTD hinge regions between monomeric MORC2 molecules (Fig. 6a-c). We identified two predominant configurations of MORC2: the O-shaped conformation, where the two ATPase domains ("heads") are dimerized while the hinge regions are engaged, and the V-shaped conformation, where the heads are dissociated but the hinge regions remain engaged (Fig. 6d, e, Supplementary Fig 13a). Interestingly, we observed that in

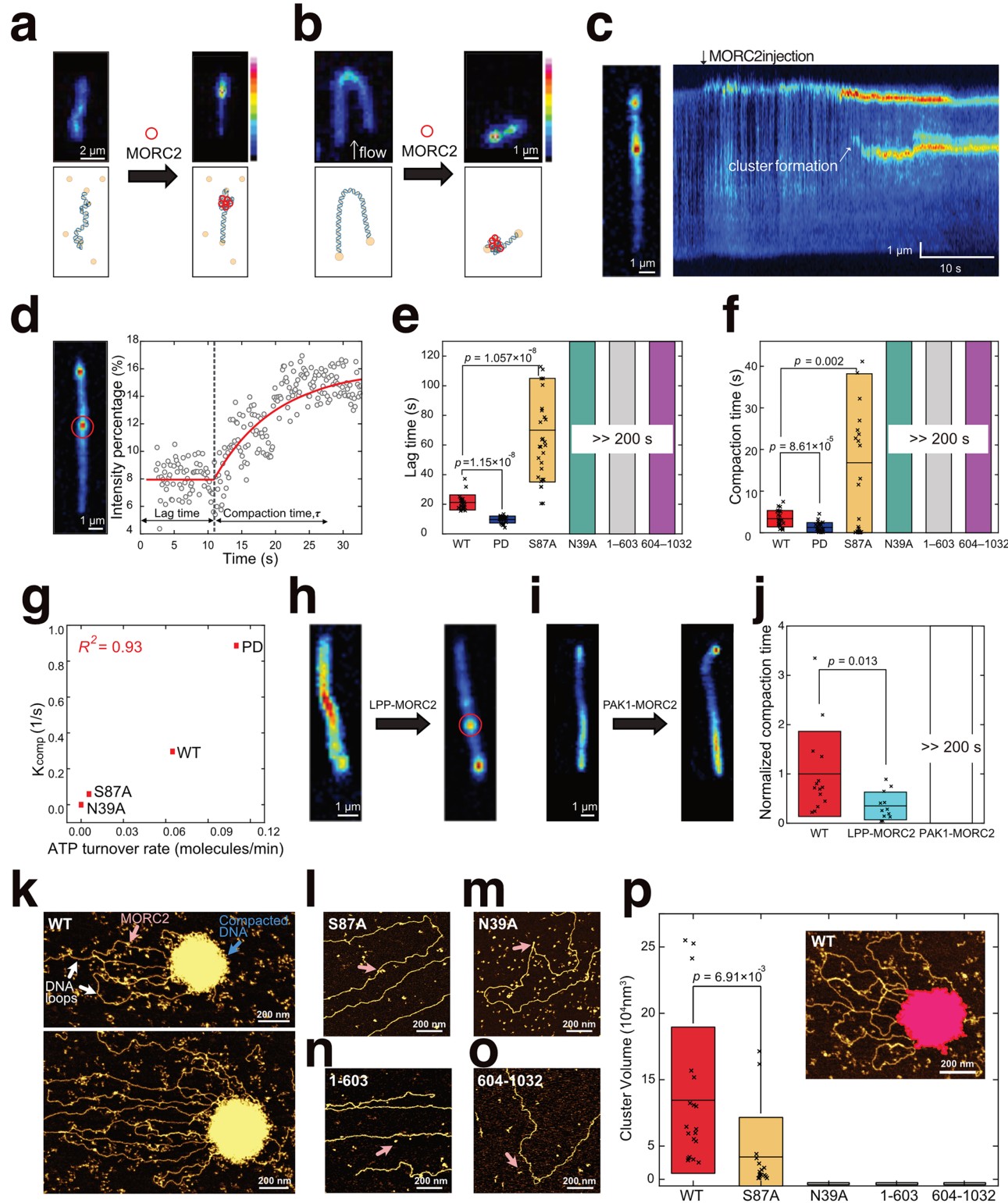

the presence of AMP-PNP, there is a higher prevalence of O-shaped conformations compared to V-shaped conformations (Fig. 6f). In contrast, the V-shaped conformation was more frequently observed with the ATP-binding-deficient mutant MORC2 N39A. With ATP, fewer O-shaped conformations were observed than with AMP-PNP, likely due to MORC2 cycling between ATP-bound, ADP-bound, and apo states. These results suggest ATP-binding induces O shaped conformations.

Next, we tested DNA-binding ability of our mutants and other variants in presence of ATP. Our results showed that the S87A mutant

of MORC2 exhibited higher DNA-binding affinity compared to the N39A mutant, suggesting that the ATP-bound state of MORC2 interact with DNA more than the ATP unbound state (Fig. 6g–i). Furthermore, we found that both the CTD and NTD of MORC2 are essential for DNA binding, as deletion mutants lacking either domain (MORC2[1-603] and MORC2[604–1032]) exhibited significantly reduced DNA binding at a MORC2 concentration of 10 nM (Fig. 6i, Supplementary Fig 13b).

Notably, about ~36% of O-shaped conformations was observed for MORC2[WT] in the presence of DNA and ATP or AMP-PNP (Fig. 6j–l)

**Fig. 5 | MORC2 compacts DNA in a manner dependent on ATP-hydrolysis and phosphorylation. a** Snapshots and schematics of doubly tethered SxO-labelled DNA, before reaction with MORC2 (left) and after (right). **b** Snapshot of a side-flow experiment, at the beginning of flow (left) and after compaction (right). **c** Snapshot and kymograph of DNA compaction. **d** Representative trace of DNA compaction mediated by MORC2. Intensity of DNA cluster (red circle) was fitted to a single exponential. **e** Lag time of WT, PD, S87A, 1–603 and 604–1032 (mean ± SD, number of technical replicates ($n$) = 23, 25, 31, 100 and 100 DNA molecules for WT, PD, S87A, 1–603 and 604–1032, respectively). **f** Compaction time of MORC2[WT], MORC2[PD], MORC2[S87A], MORC2[1–603] and MORC2[604–1032] (mean ± SD, number of technical replicates ($n$) = 23, 25, 31, 100 and 100 DNA clusters for MORC2[WT], MORC2[PD], MORC2[S87A], MORC2[1–603] and MORC2[604–1032], respectively). **g** Scatter plot of MORC2's ATP-turnover rate and DNA-compaction kinetics. **h, i** Snapshot images of DNA before and after applying LPP-treated MORC2 and PAK1-treated MORC2. LPP-treated

MORC2 induced clear DNA compaction while PAK1-treated MORC2 could not induce DNA compaction. **j** Compaction time (sec) of WT (in same reaction conditions as LPP and PAK1-treated MORC2 samples), LPP- treated MORC2, and PAK1-treated MORC2 (mean ± SD, $n$ = 14, 13, and 100 images for WT, LPP-MORC2, and PAK1-MORC2, respectively). **k–o** Dry AFM images of MORC2[WT] (**k**), MORC2[S87A] (**l**), MORC2[N39A] (**m**), MORC2[1–603] (**n**), and MORC2[604–1032] (**o**). Experiments were repeated independently 7, 3, 3, 3, and 3 times for (**k**)–(**o**), respectively. **p** Cluster volume with various conditions ($n$ = 19 clusters for MORC2[WT] and MORC2[S87A]. No clusters were observed for MORC2[N39A], MORC2[1–603] and MORC2[604–1032]. For (**e**), (**f**), (**g**), and (**p**), the centre line and bounds of the box represent the mean and SD, respectively. The $p$-values for these panels were obtained by the two-tailed unpaired $t$-test, with no adjustments for multiple comparisons. For **e–g**, **j**, and **p**, source data are provided as a Source Data file.

clearly clamping on to the DNA, thus corroborating our biochemical evidence for the same. It is worth noting that the frequency of V-shaped conformations was higher possibly due to the localization of O-shaped MORC2 within DNA clusters, making them more challenging to visualize clearly. This potential localization bias may contribute to an apparent overrepresentation of V-shaped conformations in the AFM images. Future studies are needed to investigate the structural dynamics of MORC2 with DNA.

Overall, these results, combined with biochemical and single molecule studies demonstrate that MORC2 clamps onto the DNA and uses ATP hydrolysis for DNA compaction (Fig. 6m).

## Discussion

Dominant variants in MORC2 can cause spinal muscular atrophy and the Charcot-Marie-Tooth (CMT) neuromuscular disease[39]. Further, MORC2 overexpression is associated with several cancers, including breast, gastric and liver cancers[14–16]. Notably, MORC2 phosphorylation on serine 739 in its CTD region by PAK1 kinase increases its recruitment to DNA damage sites and causes proliferation of gastric cancer cells[15,23]. In addition to phosphorylation, the MORC2 CTD undergoes other PTMs such as acetylation, PARylation and SUMOylation[22,40]. PTMs of CTD are a characteristic feature of all MORC proteins[5,23]. Despite mounting evidence highlighting the importance of MORCs CTD in various cellular processes, there remains a significant gap in our understanding of how the CTD contributes towards the molecular mechanisms underlying chromatin remodelling. Here we undertook a detailed study on full-length MORC2 and explored the functional implications of its phosphorylated CTD.

Firstly, our genomics and FLIM-FRET analyses showed that MORC2 preferentially binds open chromatin regions, suggesting that DNA, rather than nucleosomes, would be an ideal substrate for in vitro studies. MORC2 was previously shown to occupy heterochromatic regions upon lentiviral vector transduction and repress retrotransposons like LINE-1 marked by H3K9me3 via HUSH complex to initiate epigenetic silencing[18,20,21]. In contrast, our data indicate that MORC2 is predominantly accessible at the promoter regions, suggesting a broader role in transcriptional regulation. We show that full-length MORC2 exists predominantly as a dimer in vitro and its dimer population doubles in cells following DNA damage. Since MORC2 dimerisation is required for recruitment to DNA damage sites[41], it is plausible that MORC2 functions later in DNA repair when chromatin needs to be decondensed. Surprisingly, our proteomics analysis revealed that recombinant MORC2 expressed in insect cells is heavily phosphorylated at its CTD, though the specific insect cell kinases responsible remain unknown. We made a phosphodead construct with mutations of the top 6 phosphorylation sites to test the effect of phosphorylation on MORC2 function. All our cryoEM structures show AMP-PNP binding at the dimer interface similar to the previous crystal structure of MORC2 ATPase[11], consistent with ATP role in regulating ATPase domain dimerisation. MORC2 constructs containing either

phosphorylated or non-phosphorylated CTDs did not induce large structural changes in the ATPase domain upon DNA binding, revealing that the ATPase core is highly stable in the presence of AMP-PNP. The biggest structural changes we observed were rearrangements between the CC1 and CW-CC2 domains based on changes in crosslink clusters in these domains and the increased flexibility of the ATP lid as shown by increased deuterium uptake in this region upon DNA binding. This increased flexibility may reduce dimerisation and enable interaction with distal DNA regions, which could contribute to MORC2's proposed role in DNA compaction. The observed flexibility of CC1 inferred from its absence in the DNA-bound MORC2 cryoEM structures and crosslinking mass spectrometry, suggests that CC1 may aid in binding and compacting DNA. The measured DNA binding affinities of full-length MORC2, the ATPase domain, and the phosphodead mutant are in nanomolar range with dephosphorylated MORC2 having highest affinity. While our DNA competition assays indicate that the CTD enhances MORC2's ability to stably associate with DNA, the precise mechanism by which phosphorylation regulates this function remains to be fully determined.

Secondly, we identified multiple DNA binding sites across MORC2, with low-affinity sites located on CTD potentially facilitating genome surveillance in a sequence-independent manner. This mechanism could allow MORC2 to scan genome similarly to transcription factors before stabilizing specific DNA regions through its high-affinity site[42]. Our data suggest that MORC2 may trap DNA loops upon receiving a specific cellular signal, after which additional MORC2 molecules are likely recruited via the CTD. In this model, the low-affinity sites would transfer the DNA to the high-affinity sites, thereby enabling further trapping and stabilisation of DNA loops (Fig. 5m). This is consistent with our competition EMSA and recent study[43] showing that MORC2 is able to bind multiple DNA substrates. We propose that MORC2 has the ability to bridge multiple DNA substrates into compacted DNA, leading to gene silencing. These observations align with the previous studies on MORC1, which requires ATP binding for DNA loop-trapping[31]. However, unlike MORC1, our data indicate that ATP hydrolysis, rather than ATP binding alone, is required for MORC2-mediated DNA compaction. This DNA compaction is further enhanced by multiple DNA binding sites on MORC2, which have not yet been observed for MORC1. The ability of MORC2 to bind DNA through its multiple sites, combined with the flexibility of its CTD relative to the ATPase domain, explains why it is hard to determine DNA-bound MORC2 structures. This likely explains the absence of full-length and/ or DNA-bound structures for MORCs in general, indicating similar structural challenges for other MORC proteins[12,13]. Future single molecule experiments utilising fluorescently-tagged full-length MORC proteins will inform their DNA trapping functions and elucidate their roles in inducing gene silencing.

Finally, we provided direct visual evidence for MORC2 clamping onto the DNA, supporting our biochemical assays and demonstrating its ability to compact DNA into large clusters. Both the MORC2[WT] and

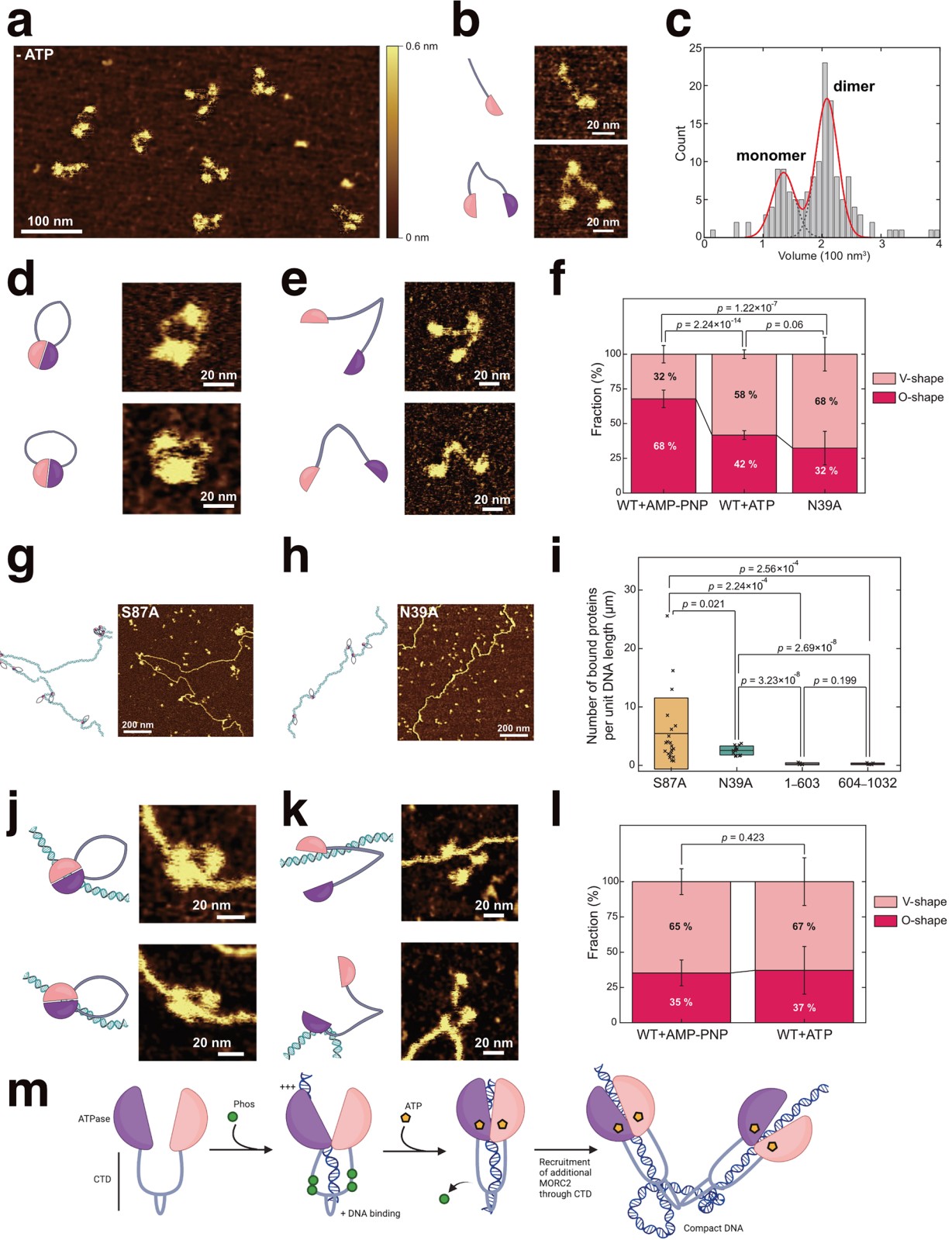

MORC2$^{PD}$ efficiently compacts DNA, whereas the N-terminal ATPase domain lacking the CC1 (1-265) or CC3-CC4 (1-603), and CTD (604-1032) do not. These findings demonstrate that NTD and CTD are indispensable for MORC2 function. Our data also shows that the phosphodead mutant compacts DNA three times faster than the WT, correlating with its two-fold higher ATP hydrolysis rate and higher DNA binding affinity. Consistent with these findings, λPPase-treated MORC2

binds DNA with higher affinity and compacts DNA faster than PAK1-treated MORC2. These observations suggest that DNA compaction by MORC2 is tightly regulated by its ATPase activity, phosphorylation state and DNA-binding properties. Our investigation of the MORC2 S87A mutant based on the patient variant S87L, which retains ATP binding but has significantly lower ATPase activity[11], showed significantly slower DNA compaction than MORC2$^{WT}$. This observation

**Fig. 6 | MORC2 changes conformation upon ATP binding via heads dimerization. a** Representative dry-AFM image of MORC2 in the absence of ATP. **b** Representative images of the monomer (top) and dimer (bottom) forms of MORC2. Experiments were repeated independently three times for (**a**) and (**b**). **c** Distributions of MORC2 volume, showing two distinct peaks, indicating two populations of monomers and dimers. Two gaussian fitting was used. Cartoons and representative images illustrating various O-shape of the MORC2 dimer (**d**) and V-shape of MORC2 dimer (**e**) in absence of DNA. **f** Relative occurrence of O-shaped and V-shaped conformations (number of technical replicates (*n*) = 252, 1001, and 68 individual proteins with AMP-PNP, AMP and the N39A mutant, respectively). Error bars represent counting errors. Cartoons and representative image of the S87A (**g**) and N39A mutants bound to DNA (**h**). **i** Number of DNA-bound MORC2 proteins per unit DNA length under various conditions: the S87A mutant (*n* = 21), N39A mutant (*n* = 13), the 1–603 fragment (*n* = 5), and the 604–1032 fragment (*n* = 5). The centre line and bounds of the box represent the mean and SD, respectively. Cartoons and representative images of O-shaped WT (**j**) and V-shaped WT (**k**) bound to DNA. **l.** Relative occurrence of O-shaped and V-shaped conformations bound to DNA (*n* = 119 molecules from 11 independent experiments and *n* = 35 molecules from 13 independent experiments with AMP-PNP and ATP, respectively). Error bars represent the counting errors for each protein shape. For (**f**), (**i**) and (**l**), the *p*-values were obtained by the one-tailed unpaired *t*-test. For (**c**), (**f**), (**i**) and (**l**), source data are provided as a Source Data file. **m** Model of DNA compaction mediated by MORC2 phosphorylation and ATP hydrolysis. A MORC2 molecule dimerised at CTD is shown, which subsequently gets dimerised at it NTD upon phosphorylation, DNA binding and ATP hydrolysis. Upon dephosphorylation, the DNA is clamped by MORC2 strongly. Eventually, another MORC2 molecule is recruited bringing in more DNA, thus leading to DNA compaction. For (**a–n**) the experiments were performed with two biological replicates. The panel m was created in BioRender. Tan, W. (2025) https://BioRender.com/11811kf.

was further complemented using the ATP-binding-deficient N39A mutant that showed no DNA compaction. These findings reinforce the critical role of ATP hydrolysis in MORC2 function and highlight differences between MORC2 and MORC1, which only requires ATP binding for DNA compaction[31]. Side-flow single-molecule experiments further revealed that MORC2 does not extrude DNA loops in a gradual loop growth but instead form DNA cluster formation, similar to findings for MORC1[31]. Given the structural similarities between MORC2 and MORC1, and the limited gene expression changes observed upon MORC2 knockout, it is likely that MORC2 compacts DNA by loop trapping specific DNA regions, such as the HUSH-associated H3K9me3 marks[21,22]. This contrasts with the global chromatin compaction by canonical SMC protein complexes, which extrudes supercoiled DNA to alter the 3D shape of the genome[44]. MORC1 can compact chromatinised DNA, but at a slower rate than naked DNA[31], so it will be intriguing to see whether MORC2 would do the same in future studies.

Our findings support the idea that phosphorylation-dependent regulation of MORC2 influences ATP hydrolysis, which in turn regulates DNA compaction. However, in contrast to previous findings that MORC2 is required for HUSH-mediated silencing of H3K9me3 region[21], our in vitro chromatin compaction of MORC2 was not shown to have functional significance in cells with regard to chromatin accessibility and transcription regulation. This may be due to different cell lines, phosphorylation state of MORC2, experimental design, methods for analysing repetitive regions in cells. Future assessment of MORC2 phosphorylation role on chromatin accessibility and retroviral elements in different cell systems may elucidate phosphorylation function in regulating transcriptional silencing.

Overall, our findings not only enhance our understanding of MORC2's function but also open new avenues for investigating the intricate relationship between protein phosphorylation and other PTMs and chromatin dynamics in cellular processes. While this work provides mechanistic insights into how CTD and its phosphorylation modulates MORC2 function, several questions remain. For instance, why does the majority of MORC2 occupy promoter regions in cells without external stimuli? How is MORC2 recruited to H3K9me3-marked heterochromatin regions? And how does MORC2 switches its role between DNA damage response and epigenetic silencing? Perhaps, studying the effect of all the PTMs in conjunction on MORC2 activity may answer some of these questions.

## Methods
### Protein expression and purification
Full-length human MORC2 (UniProt id: Q9Y6X9) and PAK1 (UniProt id: Q13153) cDNAs were synthesized by Epoch Life Sciences. For protein expression, full-length MORC2 and its variants (residues 1–265, 1–496, 496–603, 496–1032, 604–1032, 544-587 (CC2), 738-776 (CC3) 963-1020 (CC4), S87A, N39A, PD(S725A, S730A, S739A, S743A, S777A, S779A), S87L) and PAK1 were sub-cloned into a pACEBac1 vector with a

C-terminal 3 C protease-cleavage site followed by twin strepII-tag. MORC2 ATPase (residues 1-603) was cloned into pFastbac construct with 6xHis at the N-terminus and purified as previously described[29]. Each construct was transformed into EMBacY cells to generate a bacmid. Bacmid DNA was transfected into Sf9 cells using FuGENE transfection reagent (Promega) and virus was passaged twice in the same cell line before large-scale infection. Sf9 cells were a kind gift from the Glukhova lab at WEHI (negative for mycoplasma, identity not independently authenticated by us). For large-scale expression, Sf9 cells at a density of $2-2.5 \times 10^6$ cells per mL were infected with 1% (v/v) second passage (P2) virus and incubated at 27 °C for 50–60 h, 220 rpm.

The cells were harvested by centrifugation, resuspended in lysis buffer containing 50 mM HEPES, pH 8.0, 300 mM NaCl, 1 mM TCEP, 5% glycerol, 10 U per ml benzonase solution (Sigma), 1× cOmplete EDTA-free protease inhibitors (Roche) and 5 mM benzamidine hydrochloride. Cells were lysed by sonication. Clarified cell lysate was incubated with BioLock (IBA Lifesciences) and Strep-Tactin resin (IBA Lifesciences) for 1 h followed by wash with lysis buffer. All MORC2 constructs (except 1-495 and 495-603) and PAK1 proteins were eluted in elution buffer (100 mM HEPES pH 8.0, 300 mM NaCl, 1 mM TCEP, 5% glycerol and 8 mM desthiobiotin). Further purification was performed by 1 mL HiTrap Q Sepharose Fast Flow column (Cytiva) using a linear gradient of NaCl from concentration of 150 mM to 1 M in 50 mM HEPES pH 8.0, 1 mM TCEP, over 22 column volumes. For MORC2 1-495 and 495-603, this was followed by Heparin affinity chromatography (GE Healthcare) in the same buffer using a linear gradient of NaCl from concentration of 150 mM to 1 M over 20 column volumes. The final buffer for purified MORC2 constructs was 50 mM HEPES pH 8.0, -500 mM NaCl, 1 mM TCEP. The fractions containing MORC2 were pooled and concentrated using 10 kDa or 30 kDa MWCO Vivaspin Concentrator (Biostrategy), and further purified on size exclusion chromatography using the Superose 6 Increase 10/300 GL column (Cytiva).

### Size exclusion chromatography multi-angle light scattering (SEC-MALS)
Purified proteins were run on an XBridge Protein BEH SEC Column (Waters) coupled with DAWN HELEOS II light scattering detector and Optilab T-rEX refractive index detector (Wyatt Technology). The system was equilibrated in 50 mM HEPES pH 8.0, 300 mM NaCl, 1 mM TCEP and calibrated using bovine serum albumin (2 mg ml⁻¹) before analysis of experimental samples. For each experiment, 20 μl of purified protein (1 mg ml⁻¹) was injected onto the column and eluted at a flow rate of 0.5 ml min⁻¹. Experimental data were collected and processed using ASTRA (Wyatt Technology, v.7.3.19).

### Electrophoretic mobility shift assay (EMSA)
Different DNA substrates were formed by annealing oligonucleotides (synthesised by IDT, described in Supplementary Table 3) at 95 °C for

5 min and slowly cooling down to room temperature over ~2 h. The X0m1 oligonucleotide was 6FAM-, IRDye680- or IRDye800- labelled at 5' end. The Xom1 or P1 oligonucleotides were annealed with their corresponding X0m1.com or P7 oligonucleotides. DNA and protein concentration used are mentioned in the figure legends. In general, the DNA and protein were incubated in 10 μL reaction buffer (20 mM Tris pH 7.4, 75 mM NaCl, 5% glycerol, 0.005% NP-40, 0.5 mM MgCl2) at room temperature for 30 min. Any pre-treatment of proteins prior to EMSA are mentioned in their corresponding figure legends. For competitive EMSA experiments, 200 nM 101 bp linear DNA or 101 bp circular was incubated with 100 nM of MORC2 proteins at room temperature for 30 min, followed by addition of competitive IRDye800-dsDNA (90 bp) with increasing concentrations (50, 100, 200, 400, 800 nM) at room temperature for 10 minutes. The reaction was resolved by electrophoresis on a 6% non-denaturing poly-acrylamide gel in TBE (100 mM Tris, 90 mM boric acid, 1 mM EDTA) buffer and visualised by ChemiDoc (BioRad) using SYBR Gold stain or imaging in the 6FAM, IRDye680 or IRDye800 channels. During exposure, it was made sure none of the EMSA bands were oversaturated. Each experiment was performed two or three times (as indicated in the figure legends) and the percentage of shifted bands was quantified using ImageJ (as shown in Supplementary Fig 3a or Supplementary Fig 10a) and plotted on Prism. For all EMSA except competition EMSA, a 30 ×350 pixel box was drawn for the whole lane containing shifted (bound) and unshifted (unbound) DNA. The reported values are the percentage between the bound DNA and sum of total and unbound DNA. For competition EMSA, percentage of shifted IRDye800-DNA is calculated by measuring the intensities of bottom bands and fraction of loss of bands against the IRDye800-DNA inputs (50, 100, 200, 400 and 800 nM, Supplementary Fig 10a). All EMSA points are presented as mean ± standard deviation (SD).

### In vitro kinase and phosphatase assay

Ten μg of purified MORC2 and 0.1 μg of purified PAK1 were incubated in 60 μL of reaction buffer (20 mM Tris-HCl pH 7.4, 10 mM MgAc, 0.5 mM DTT, 0.05% Tween-20, 100 mM KCl and 0.2 mM ATP) for 30 min at 30 °C. For phosphatase experiments, 600 units of λ-protein phosphatase (NEB) was added to reactions together with 1 mM MnCl2 and incubated for 30 min at 30 °C. Ten μg of purified MORC2 was incubated in 60 μL of reaction buffer for 30 min at 30 °C as a negative control (labelled as "pre-treated"). All kinase reactions were performed in four technical replicates and quenched in 1X Laemmli sample buffer (Bio-rad) and run on 12.5% SuperSep Phos-tag gels (FujiFilm) containing 25 mM Tris, 250 mM glycine and 0.1% SDS running buffer. Gels were stained with InstantBlue Coomassie Protein Stain (Abcam). Gel bands containing phosphorylated or dephosphorylated MORC2 were excised for mass spectroscopy analysis.

### Mass spectrometry-based proteomics

Protein gel bands (number of technical replicates, $n = 3$ or 4) were carefully excised and were destained in a 50% acetonitrile (ACN) solution for 30 minutes at 37 °C. Subsequently, the gel band was dehydrated using 100% ACN at room temperature for 15 minutes. The ACN was aspirated, and the gel piece was further dried using a vacuum centrifuge (CentriVap, Labconco).

Proteins were reduced using 1 mM dithiothreitol (DTT) in 50 mM ammonium bicarbonate for 30 minutes at RT. Any excess DTT was carefully aspirated before adding 55 mM iodoacetamide to perform alkylation. Alkylation was performed at RT for 60 minutes. Following reduction and alkylation, the gel piece was washed again using 50% ACN twice, followed by a single wash with 100% ACN and drying the gel piece again.

Proteins within the gel piece were then subjected to overnight digestion using 500 ng trypsin in 50 mM ammonium bicarbonate, carried out at a temperature of 37 degrees. The following day, peptides were extracted using a solution of 60% ACN and 0.1% formic acid (FA). The collected peptides were subsequently lyophilised to dryness using a CentriVap (Labconco). The final step involved reconstituting the peptides in a 10 μl solution containing 0.1% FA and 2% ACN, preparing them for analysis via mass spectrometry.

The digested peptides were reconstituted in a solution containing 2% ACN and 0.1% formic acid (FA), and subsequently analysed using an Orbitrap Eclipse Tribrid mass spectrometer coupled with a Neo Vanquish LC. The samples were loaded onto a C18 fused silica column (inner diameter 75 μm, outer diameter 360 μm, and length 15 cm, with 1.6 μm C18 beads) packed into an emitter tip (IonOpticks) using pressure-controlled loading, with a maximum pressure of 1,500 bar. The system was interfaced to an Orbitrap Eclipse™ Tribrid mass spectrometer via Easy nLC source, and the peptides were electro-sprayed directly into the mass spectrometer.

For liquid chromatography (LC) separation, a linear gradient of solvent-B (99% acetonitrile) was applied at a flow rate of 400 nl/min. The gradient started at 2% solvent-B and increased to 30% over 5 minutes, followed by a gradient from 5% to 17% solvent-B for 15 minutes and from 17% to 34% solvent-B for the next 28 minutes. A column washing step at 85% solvent-B for 9 minutes concluded the 54-minute. Data was acquired in in a data-dependent acquisition (DDA) mode.

During MS1 acquisition, spectra were obtained in the Orbitrap with a resolution of 120,000 (R), standard normalized automatic gain control (AGC) target, Auto MaxIT, RF Lens set at 30%, and a scan range of 380–1400. Dynamic exclusion was implemented for 30 seconds, excluding all charge states for a given precursor. Data-dependent MS2 spectra were collected in the Orbitrap for precursors with charge states 3-8, utilizing a resolution of 50,000 (R), assisted HCD collision energy mode, normalized HCD collision energies set at 25% and 30%, normal scan range mode, normalized AGC target of 200%, and MaxIT of 150 ms.

The raw mass spectrometry (MS) data files underwent processing using MaxQuant version 2.0.1.0, incorporating the Andromeda search engine [23]. The analysis was conducted against the Human proteome fasta database obtained from UniProt in April 2022. Variable modifications included protein N-terminal acetylation, methionine oxidation, and phosphorylation on serine, threonine, and tyrosine (STY), while cysteine carbamidomethylation was selected as a fixed modification. Trypsin served as the specified enzyme for digestion, with an allowance for up to two missed cleavages. To ensure high confidence in the results, the false discovery rate (FDR) was set at 1% for both proteins and peptides. A peptide tolerance of 4.5 ppm was used during main search.

### Cryo-electron microscopy (cryoEM)

Purified proteins were vitrified by applying 3-4 μL purified protein (2 μM), either in presence of three-fold molar excess of 60 bp DNA or in its absence, to UltrAuFoil R1.2/1.3 grids (Quantifoil). The grids were glow discharged in the presence of amylamine in PELCO easiGlow. The grids were prepared using a Vitrobot Mark IV (ThermoFisher) at 4 °C and 100% humidity with blotting force of −3 N for 3 s. The MORC2$^{S87A}$ and MORC2$^{PD}$ grids were imaged on Titan Krios microscope operated at 300 keV using a Gatan K3 detector. The MORC2$^{1-603}$ grids were collected on Talos Arctica operating at 200 keV using a Gatan K2 detector.

### CryoEM image processing and model building

Image processing was performed in cryoSPARCv3 and v4[45]. For all datasets, the movies were aligned and averaged using patch motion correction, and contrast transfer function parameters were estimated using Patch CTF estimation. Particles were picked using Cryosparc "Blob Picker" tool or crYOLO[46], and the particle coordinates were transferred to cryoSPARC for 2D class average. After 2D classification of the auto-picked particles, selected classes were used to make an ab

initio model. Heterogenous refinement with a 'junk' model was used to clean up particles for higher resolution template. The particles were further refined using homogenous refinement protocol with imposed C2 symmetry. Final set of particles reaching 1.9 – 3.2 Å were used to perform non-uniform refinement with imposed C2 symmetry. The 3D maps were post-processed to automatically estimate and apply the B-factor and to determine the resolution by Fourier shell correlation between two independent half datasets using 0.143 criterion. Local resolution was estimated using CryoSparc "Local Resolution Estimation" Tool (version 4.5.1). Directional FSC and sphericity values for the maps were generated using the 3DFSC server (https://3dfsc.salk.edu/)[47].

The crystal structure of MORC2 ATPase domain (PDB: 5OF9) was used for modelling into MORC2$^{PD}$, MORC2$^{1-603}$, MORC2$^{PD-DNA}$ and MORC2$^{1-603-DNA}$ and the crystal structure of MORC2 S87L variant (PDB: 5OFB) was used for MORC2$^{S87A}$. Model placement and initial refinement was performed in Coot[48] and further refined using Phenix refinement[49]. ChimeraX (v.1.7) was used to color the surface of the cryoEM maps based on local resolution parameters obtained from Cryosparc. All images were rendered in ChimeraX (v.1.7).

### Generation of MORC2 KO HEK293 cells
Two guide RNAs (gRNAs) were designed using Synthego for human MORC2 (https://design.synthego.com/#/) to generate two biological bulk replicates. The gRNA sequences are UUCAGGGGCU-CAAUGCGCAU and UCAGGGGCUCAAUGCGCAUU. These gRNA were ordered from IDT. The SF cell line 4D-Nucleofector X Kit S (V4XC-2032) from Lonza was used to prepare a mix of 300 pmol of gRNA and 100 pmol of Cas9 protein. About 1 ×106 HEK293 cells were electroporated using Lonza Amaxa TM machine following the manufacturer's instruction. The transfected cells were cultured until they become confluent and split two more times before freezing down. The KO efficiency was measured by performing western blot analysis using anti-MORC2 antibody (Invitrogen #PA551172, Supplementary Fig 11c) on the cell lysate (~10,000 cells) from both the gRNAs. The gRNA1 and gRNA2 KO cells were used for our downstream assays. RNAseq, ATAC-seq and ChIP-seq were each performed on 2 independent WT replicates, and 2 independent KO cell lines generated using different MORC2-targeting sgRNA sequences.

### Fluorescence polarization ATPase assay
Fluorescence polarization ATPase assays were performed as outlined in ref. 50. A 10 μL reaction was set up in triplicates in 384-well low flange, black, flat-bottom plates (Corning) containing 7 μL reaction buffer (50 mM HEPES pH 7.5, 4 mM MgCl$_2$, 2 mM EGTA), 1 μL recombinant protein at concentrations ranging from 0.1-0.6 μM or SEC buffer control, 1 μL nuclease-free water and 1.25-10 μM ATP substrate. Reactions were incubated at 25 °C for 1 hour in the dark. Reactions were stopped by the addition of 10 μL detection mix (1X Detection buffer, 4 nM ADP Alexa Fluor 633 Tracer, 128 mg/mL ADP$^2$ antibody) and incubated for another hour in the dark. Fluorescence polarization readings (mP) were measured using an Envision plate reader (PerkinElmer Life Sciences) fitted with excitation filter 620/40 nm, emission filters 688/45 nm (s and p channels) and D658/fp688 dual mirror. Readings from a free tracer (no antibody) control were set as 20 mP as the normalization baseline of the assay for all reactions. The amount of ADP produced by each reaction was estimated by a 12-point standard curve, as outlined in the manufacturer's protocol. Data were plotted and analyzed in GraphPad Prism.

### Surface plasmon resonance (SPR)
SPR binding studies of MORC2 variants to DNA were performed using a Biacore S200 Instrument (Cytiva). Biotinylated 60 bp dsDNA was prepared by assembling oligonucleotides 1 (/5Biosg/ACG CTG CCG AAT TCT ACC AGT GCC TTG CTA GGA CAT CTT TGC CCA CCT GCA

GGT TCA CCC) and oligonucleotides 2 (GGG TGA ACC TGC AGG TGG GCA AAG ATG TCC TAG CAA GGC ACT GGT AGA ATT CGG CAG CGT). Biotinylated 60 bp dsDNA was diluted to 5 μg/mL in SPR running buffer (10 mM HEPES pH 7.4, 300 mM NaCl, 3 mM EDTA and 0.05% (v/v) Tween-20) to a final immobilisation level of 200-220 response units (RU) on the Streptavidin chip (Cytiva). An additional injection step with 1 min of Biolock (IBA) was used to block StrepII-tagged MORC2 from binding to the Streptavidin chip. A blank activation/deactivation was used for the reference surface. DNA binding studies were performed at 20 °C in SPR running buffer. Any pre-treatment of proteins prior to SPR are mentioned in their corresponding figure legends. All proteins were diluted to 1000 nM stock in SPR running buffer and prepared as an 8-point concentration series (2-fold serial dilution, 4-500 nM). Samples were injected in a multi-cycle run (flow rate 30 μL/min, contact time of 60 s, dissociation 120 s) with regeneration with 0.5 M EDTA buffer. Sensorgrams were double referenced, and steady-state binding data fitted using a 1:1 binding model using Biacore S200 Evaluation Software (Cytiva, v1.1). Representative sensorgrams and fitted dissociation constant ($K_D$) values, depicted as mean ± SEM ($n = 3$ independent experiments) are shown in Fig. 3b.

### Quantitative crosslinking mass spectrometry
Sulfo-SDA (ThermoFisher) was dissolved freshly in assay buffer (50 mM HEPES pH 8.0, 300 mM NaCl, 1 mM TCEP) to a final concentration of 0.5 mg/mL. Crosslinking reactions were performed in triplicate by incubating 0.5 mg/mL MORC2 in the presence of absence of AMP-PNP and DNA, with 0.5 mg/mL sulfo-SDA in a final volume of 10 μL at room temperature for 30 min. To activate sulfo-SDA, the samples were irradiated with a 1000 kV UV lamp at 356 nm on ice for 1 min. Crosslinked samples were analysed on a 3-8% NuPAGE Tris-acetate gel (Invitrogen), and gel bands corresponding to the crosslinked MORC2 were excised and digested as previously described above[51].

Reconstituted peptides were analysed on Orbitrap Eclipse Tribrid mass spectrometer that is interfaced with Neo Vanquish liquid chromatography system. Samples were loaded on to a C18 fused silica column (inner diameter 75 μm, OD 360 μm × 15 cm length, 1.6 μm C18 beads) packed into an emitter tip (IonOpticks) using pressure-controlled loading with a maximum pressure of 1,500 bar, that is interfaced to an Orbitrap Eclipse™ Tribrid (Thermo Scientific) mass spectrometer using Easy nLC source and electro sprayed directly into the mass spectrometer.

We then employed a linear gradient of 3% to 30% of solvent-B at 400 nl/min flow rate (solvent-B: 99% (by vol) acetonitrile) for 100 min and 30% to 40% solvent-B for 20 min and 35% to 99% solvent-B for 5 min which was maintained at 90% B for 10 min and washed the column at 3% solvent-B for another 10 min comprising a total of 145 min run with a 120 min gradient in a data dependent (DDA) mode. MS1 spectra were acquired in the Orbitrap (R = 120k; normalised AGC target = standard; MaxIT = Auto; RF Lens = 30%; scan range = 380–1400; profile data). Dynamic exclusion was employed for 30 s excluding all charge states for a given precursor. Data dependent MS2 spectra were collected in the Orbitrap for precursors with charge states 3-8 (R = 50k; HCD collision energy mode = assisted; normalized HCD collision energies = 25%, 30%; scan range mode = normal; normalised AGC target = 200%; MaxIT = 150 ms).

Quantitative crosslinking analysis was performed following the previously published method[51]. Raw data files were converted to MGF files using MS convert[52]. MGF files were searched against a fasta file containing the MORC2 sequence using XiSearch software[53] (version 1.7.6.7) with the following settings: with the following settings: cross-linker = multiple, SDA and noncovalent; fixed modifications = Carbamidomethylation (C); variable modifications = oxidation (M), sda-loop (KSTY) DELTAMASS:82.04186484, sda-hydro (KSTY) DELTA-MASS:100.052430; MS1 tolerance = 6.0ppm, MS2 tolerance =

20.0ppm; losses = $H_2O$, $NH_3$, $CH_3SOH$, CleavableCrossLinkerPeptide:MASS:82.04186484). False Discovery Rate (FDR) was performed with the in-built xiFDR set to 5%[54]. Identified crosslinks were converted to linear peptide sequences using the skyline convert tool and quantities were calculated with Skyline[51]. Data were visualised using the XiView software[55].

Data cleaning, pre-processing and statistical analysis were conducted using R (version 4.2.1). Decoy precursors and crosslinked peptides with Isotope.Dot.Product <0.8 were filtered out. Additionally, crosslinked peptides that quantified in at least 67% of replicates within at least one condition were considered for further analysis. Total of 910 crosslinked peptides were included in the analysis. The normalised area of crosslinked peptides were log-transformed and missing values were imputed using the Barycenter approach (v2-MNAR) as implemented in the msImpute package (v.1.8.0)[56]. Multivariate analysis, principal component analysis (PCA), was employed to identify any potential outliers. Differential analysis was carried out using the limma package (v. 3.54.2)[57]. A crosslinked peptide was deemed significantly differentially expressed if the FDR was ≤ 5% following Benjamini–Hochberg (BH) correction.

## Hydrogen deuterium exchange mass spectrometry
MORC2 ATPase domain (1-603) and CTD (496-1032) were assayed with 60 bp dsDNA (1:3 molar ratio of protein to DNA) or in absence of DNA. All samples had 2.5 mM AMP-PNP. HDX labelling of protein was performed at 20 °C for periods of 6, 60, 600, 6000 s using a PAL Dual Head HDX Automation manager (Trajan/LEAP) controlled by the ChronosHDX software (Trajan). A 4 μL of the sample (containing ~15 μM of protein) was transferred to 55 μL of non-deuterated (50 mM potassium phosphate buffer pH 7.4, 150 mM NaCl, in $H_2O$) or deuterated (50 mM potassium phosphate buffer pD 7.0, 150 mM NaCl, in $D_2O$) buffer and incubated for the respective time. Quenching was performed by adding 50 μL of the deuterated protein to 55 μL of quench buffer (50 mM potassium phosphate buffer, pH 2.3 containing 4 M guanidine hydrochloride) at 1 °C. For online pepsin digestion, 95 μL of the quenched sample was passed over an immobilized 2.1 × 30 mm Enzymate BEH pepsin column (Waters) equilibrated in 0.1% (v/v) formic acid in water at 100 μL/min. Proteolyzed peptides were captured and desalted by a C18 trap column (VanGuard BEH; 1.7 μm; 2.1 × 5 mm (Waters)) and eluted with acetonitrile and 0.1% (v/v) formic acid gradient (5% to 40% in 8 min, 40% to 95% in 0.5 min, 95% 1.5 min) at a flow rate of 40 μL/min and separation on an ACQUITY UPLC BEH C18 analytical column (1.7 μm, 1 × 100 mm (Waters)) delivered by ACQUITY UPLC I-Class Binary Solvent Manager (Waters).

For mass spectrometry, a SYNAPT G2-Si mass spectrometer (Waters) was used. Instrument settings were: 3.0 KV capillary and 40 V sampling cone with source and desolvation temperature of 100 and 40 °C respectively. The desolvation and cone gas flow was at 80 L/hr and 100 L/hr, respectively. High energy ramp trap collision energy was from 20 to 40 V. All mass spectra were acquired using a 0.4 s scan time with continuous lock mass (Leu-Enk, 556.2771 m/z) for mass accuracy correction. Data were acquired in $MS^E$ mode.

Raw data obtained from non-deuterated samples were processed using Protein Lynx Global Server (PLGS) v3.0 (Waters). This processing used a database that contained sequence information about *Sus scrofa* pepsin A and each MORC2 construct. PLGS processing parameters were 135 and 30 counts for low and elevated energy threshold, respectively. Primary digest reagent was set to non-specific and with oxidation methionine and phosphorylation of serine, threonine and tyrosine as variable modification. FDR was set to 1%. In the case of labelled data, the analysis was performed using DynamX 3.0, with 5000 minimum intensity criteria, 0.3 products per amino acid, 1 consecutive product, and error of 10 ppm. The file threshold was 3 out of 4. Data were manually curated (Supplementary Table 4). Heatmaps were made using in-house Python scripts (D'Arcy laboratory).

Structural mappings were done using scripts from DynamX in PyMOL version 2.5.

## Fluorescence microscopy
**Microscopy data acquisition.** All microscopy measurements were performed on an Olympus FV3000 laser scanning microscope coupled to an ISS A320 Fast FLIM box for fluorescence fluctuation data acquisition. A 60X water immersion objective 1.2 NA was used for all experiments and the cells were imaged at 37 degrees in 5% CO2. For single channel Number and Brightness (NB) measurement the monomeric pBacMam-meGFP-StrepII, which will be referred as "meGFP" (Addgene #160686), and pBacMam-MORC2-meGFP-StrepII tagged plasmids were excited by a solid-state laser diode operating at 488 nm and the resulting fluorescence signal directed through a 405/488/561 dichroic mirror to a photomultiplier detector (H7422P-40 of Hamamatsu) fitted with an eGFP 500/25 nm bandwidth filter. DNA double-strand breaks were induced by adding cell culture medium containing 0.5 μg/mL of neocarzinostatin (Sigma, Cat. No. N9162-100 μg) for 1 hour in a 37 °C tissue culture incubator. NB fluorescence fluctuation spectroscopy (FFS) measurement of the meGFP tagged MORC2 construct involved selecting a HEK293T MORC2 KO cell exhibiting low meGFP expression level which enabled observation of fluctuations in meGFP fluorescence intensity and then selecting a 10.6 μm region of interest (ROI) within that MORC2 KO cell's nucleus, which for a 256 ×256 pixel frame resulted in a pixel size of 41 nm (an oversampling of the point spread function of our diffraction limited acquisition). A frame scan acquisition (N = 100 frames) was then acquired with the pixel dwell time set to 12.5 μs, which resulted in a line time of 4.313 ms and a frame time of 1.108 s. HEK293T nucleus co-expressing the histone FRET pair H2B-eGFP and H2B-mCh that has been fixed with IF against MORC2 or H3K9me3. The FLIM imaging involved sequential imaging of a two-phase light path in the Olympus FluoView software. The first phase was set up to image H2B-eGFP and H2B-mCh via use of solid-state laser diodes operating at 488 and 561 nm, respectively, with the resulting signal being directed through a 405/488/561/6033 dichroic mirror to two internal GaAsP photomultiplier detectors set to collect 500–540 nm and 600–700 nm. The second phase was set up to image AF647 via use of solid-state laser diodes operating at 633 nm, with the resulting signal being directed through a 405/488/561/633 dichroic mirror to the internal GaAsP photomultiplier detector set to collect 600–700 nm. Then in each HEK293T nucleus selected, a FLIM map of H2B-eGFP was imaged within the same field of view (256×256-pixel frame size, 20 μs/pixel, 90 nm/pixel, 20 frame integration) using the ISS VistaVision software. This involved excitation of H2B-eGFP with the external pulsed 488 nm laser (80 MHz) and the resulting signal being directed through a 405/488/561/633 dichroic mirror to an external photomultiplier detector (H7422P-40 of Hamamatsu) that was fitted with a 520/50 nm bandwidth filter. The donor signal in each pixel was then subsequently processed by the ISS A320 FastFLIM box data acquisition card to report the fluorescence lifetime of H2B-eGFP. All FLIM data were pre-calibrated against fluorescein at pH 9 which has a single exponential lifetime of 4.04 ns.

**Number and brightness (NB) analysis.** The brightness of a fluorescently tagged protein is a readout of that protein's oligomeric state that can be extracted by a moment-based Number and brightness (NB) analysis of an FFS frame scan acquisition[33,34,58,59]. We imaged the whole nucleus containing meGFP or MORC2-eGFP, and selected a region of interest unbiasedly across each cell. In brief, within each pixel of a frame scan we have an intensity fluctuation that has an average intensity (first moment) and a variance (second moment). The ratio of these two properties describes the apparent brightness (B) of the molecules that gave rise to the intensity fluctuation. The true molecular brightness (ε) of the molecules is related to the measured apparent brightness (B) by $B = \varepsilon + 1$, where 1 is the brightness

contribution of the photon counting detector. Thus calibration of the apparent brightness of monomeric eGFP ($B_{monomer}$ = 1.15) enables determination of the molecular brightness of eGFP (($\varepsilon_{monomer}$ = 0.15) and extrapolation of the expected apparent brightness of MORC2-meGFP dimers ($B_{dimer}$ = 1.30) versus oligomers ($B_{oligomer}$ = 1.60). From definition of apparent brightness cursors that are centred to detect these different eGFP labelled species the fraction of pixels containing MORC2-meGFP dimer ($B_{dimer}$) versus MORC2-meGFP oligomer ($B_{oligomer}$) can be spatially mapped throughout a frame scan acquisition and quantified across multiple cells, as has been carried out for several other nuclear proteins[33,58]. Artefacts due to cell movement or photobleaching were subtracted from acquired intensity fluctuations via use of a moving average algorithm. All brightness calculations were carried out in SimFCS from the Laboratory for Fluorescence Dynamics (www.lfd.uci.edu).

**FLIM-FRET analysis.** The fluorescence decay recorded in each pixel of a FLIM image can be employed to detect FRET between two fluorescent proteins and was quantified here by the phasor approach to lifetime analysis[34]. In brief, each pixel in a FLIM image gives rise to a single point (phasor) in the phasor plot, and, when used in reciprocal mode, enables each point of the phasor pot to be mapped to each pixel of the FLIM image. Since phasors follow simple vector algebra, it is possible to determine the fractional contribution of two or more independent molecular species coexisting in the same pixel (e.g., autofluorescence and a fluorescent protein). In the case of two independent species, all possible weightings give a phasor distribution along a linear trajectory than joins the phasors of the individual species in pure form[32]. In the case of a FRET experiment where the lifetime of the donor molecule is changed upon interaction with an acceptor molecule, the realization of all possible phasors quenched with different efficiencies describe a curved trajectory in the phasor plot that follows the classical definition of FRET efficiency[32]. As described in previous papers to employ the phasor approach to FLIM for detection of FRET between fluorescent histones[34], the contribution of the background (cellular auto-fluorescence) and of the donor without acceptor were evaluated by using the rule of the linear combination, with the background phasor and donor unquenched determined independently. By linking two phasor cursors between these two terminal phasor locations, we can pseudo-colour each pixel of a FLIM map according to the exact FRET efficiency detected at that location. Furthermore, from linking two additional cursors between the donor phasor and background phasor, we can also quantify the contribution of cellular autofluorescence in each pixel. To increase phasor accuracy, a 3 ×3 spatial median filter was applied twice to the FLIM maps presented in before FRET analysis. All FLIM-FRET quantification was performed in the SimFCS software developed at the LFD.

**Immunofluorescence (IF) mask analysis of histone FRET.** To quantify the chromatin nanostructure associated with different MORC2 or H3K9me3, we applied an intensity threshold mask based on IF intensity images to FLIM maps pseudo-coloured according to histone FRET (compact chromatin) versus no FRET (open chromatin). This involved: (1) smoothing each HEK293T nucleus' IF image with a 3×3 spatial median filter, (2) transforming this smoothed image into a binary mask based on an intensity threshold that retains the top ~5% intensity pixels, (3) applying the IF-guided mask to its associated FLIM map pseudo-coloured according to histone FRET, and (4) quantification of the fraction of compact chromatin within high intensity MORC2/H3K9me3 region versus outside the IF-guided mask (nucleoplasm) in total.

**ChIP-seq**
ChIP-seq was performed as per[60] with amendments. Briefly, 5-10 ×10⁶ cells per IP were crosslinked by adding 1/10 volume of fresh 11% formaldehyde solution (50 mM HEPES-KOH pH 7.4, 100 mM NaCl, 1 mM

EDTA, 1 mM EGTA, 11% formaldehyde) to a 1 ×10⁶ cell suspension for 20 minutes at room temperature with mixing. 1/12.5 volumes of 2.5 M glycine was added to quench formaldehyde and cells were washed twice in ice-cold PBS. Crosslinked cells were lysed in Lysis Buffer 1 (50 mM HEPES-KOH pH 7.5, 140 mM NaCl, 1 mM EDTA, 10% (v/v) glycerol, 0.5% (v/v) IGEPAL CA-630, 0.25% (v/v) Triton X-100, 1x Roche cOmplete™, EDTA-free Protease Inhibitor Cocktail) for 10 minutes at 4 °C with rocking, pelleted at 1350 x *g* for 5 minutes at 4 °C, then Lysis Buffer 2 (10 mM Tris-HCl pH 8.0, 200 mM NaCl, 1 mM EDTA, 0.5 mM EGTA, 1x Roche cOmplete™, EDTA-free Protease Inhibitor Cocktail) for 10 minutes at room temperature with rocking, pelleted again at 1350 x *g* for 5 minutes at 4 °C, then resuspended in Lysis Buffer 3 (10 mM Tris-HCl pH 8.0, 100 mM NaCl, 1 mM EDTA, 0.5 mM EGTA, 0.1% (w/v) Na-Deoxycholate, 0.5% (v/v) N-lauroylsarcosine, 1x Roche cOmplete™, EDTA-free Protease Inhibitor Cocktail) and transferred to Covaris miniTUBEs. Cell lysates were sonicated using a Covaris E220 focused-ultrasonicator with the following settings to shear DNA to a size of 300-500 bp suitable for high-throughput sequencing: Fill Level 10, Duty Cycle 10, PIP 175, Cycles/Burst 200 for 80 seconds. 1/10 volume of 10% (v/v) Triton X-100 was then added to the sonicated lysate and cellular debris was pelleted at 20,000 x *g* for 10 minutes at 4 °C. Immunoprecipitation was then performed overnight at 4 °C with rotation using ProteinG Dynabeads (50 μL per IP) pre-coupled with antibody (5 μL per IP) in 0.5% BSA/PBS for >5 h. Antibodies used were anti-MORC2 Rabbit pAb (Invitrogen #PA5-51172), H3K9me3 Rabbit pAb (Abcam #8898), Rabbit IgG pAb (Abcam #ab46540). Following immunoprecipitation, beads were collected using a magnet and washed 5 times with Wash Buffer (50 mM HEPES-KOH pKa 7.55, 500 mM LiCl, 1 mM EDTA, 1% (v/v) IGEPAL CA-630, 0.7% (w/v) Na-Deoxycholate, then 1 time with Tris-EDTA-NaCl buffer (10 mM Tris-HCl pH 8.0, 1 mM EDTA, 50 mM NaCl). DNA was then eluted from beads by incubation in Elution Buffer (50 mM Tris-HCl pH 8.0, 10 mM EDTA, 1% (w/v) SDS) at 65 °C for 30 minutes with shaking. Bead-free supernatant was reverse crosslinked by incubation at 65 °C for 6-15 h, diluted with equal volume of TE Buffer (10 mM Tris-HCl pH 8.0, 1 mM EDTA), then digested sequentially with RNaseA (0.2 mg/mL) at 37 °C for 2 h and Proteinase K (0.2 μg/mL) at 55 °C for 2 h. Final immunoprecipitated DNA was then purified using Zymo ChIP DNA Clean & Concentrator kit and prepared for sequencing using Illumina Truseq ½ reaction library preparation kit. Final libraries were quantified using an Agilent Tapestation using a D1000 screentape and sequenced using an Illumina NextSeq2000 to generate libraries with 24 – 119 million paired-end 65 bp reads.

**ATAC-seq**
ATAC-seq was performed using the Omni-ATAC method as described previously[61]. For MORC2 overexpression studies, pBacMam-MORC2-meGFP, pBacMam-MORC2-S6A-meGFP (S725A, S730A, S739A, S743A, S777A, S779A), or pBacMam-MORC2-S6D-meGFP (S725D, S730D, S739D, S743D, S777D, S779D) was transfected to HEK293T cells or HEK293T MORC2 KO cells with Lipofectimine 2000 for 72 hours. Briefly, 50,000 cells were pelleted at 500 x *g* for 5 minutes at 4 °C then lysed on ice for 3 minutes in 50 μL ATAC-RSB Buffer (10 mM Tris-HCl pH 7.4, 10 mM NaCl, 3 mM MgCl₂) also containing 0.1% (v/v) IGEPAL CA-630, 0.1% (v/v) Tween-20, 0.01% (w/v) Digitonin. After lysis, 1 mL of ATAC-RSB containing only 0.1% (v/v) Tween-20 but no IGEPAL CA-630 or Digitonin was added, and nuclei were pelleted at 500 x *g* for 10 minutes at 4 °C. All supernatant was aspirated and discarded, and each pellet of nuclei was resuspended in 50 μL of transposition mix (25 μL 2x TD Buffer, 2.5 μL Tn5 transposase (100 nM final), 16.5 μL PBS, 0.1 μL 5% digitonin, 0.5 μL 10% Tween-20, 5 μL H₂O) and incubated at 37 °C for 30 minutes exactly in a thermomixer with 1000 RPM mixing. DNA was then purified using Zymo DNA Clean & Concentrator-5 Kit to generate 20 μL of product which was pre-amplified for 5 cycles using NEBNext 2x Master Mix (#E7649A) and indexed primers as described previously[62]. A 5 μL of pre-amplified DNA was then used in a 15 μL qPCR

reaction using Promega GoTaq qPCR Master Mix to determine the required number of additional cycles to yield minimally amplified DNA for optimal library diversity for next-generation sequencing (determined to be 3-4 additional cycles for these libraries). Amplified DNA was then purified using NucleoMag NGS Clean-up and Size Select beads (Macherey-Nagel) using a 0.5x ratio of beads to DNA to initially clear large DNA fragments from the library and then a 1.5x ratio of beads to supernatant to adsorb DNA fragments of appropriate size for sequencing. DNA-bound beads were then washed twice with 80% EtOH and size-selected DNA was eluted in nuclease-free water. Final libraries were quantified using an Agilent Tapestation using a D5000 screentape and sequenced using an Illumina NextSeq2000 to generate ~39 M paired-end 65 bp reads per library.

### RNA-seq
Total RNA was isolated from 1 ×106 cells using RNeasy Plus Mini Kit (Qiagen) with genomic DNA removal. RNA was confirmed to be of high quality using an Agilent Tapestation using an RNA screentape (RIN > 9.8 for these samples) and 500 ng was prepared for RNA-seq using Truseq RNA library preparation protocol with poly-A capture (Illumina). Final libraries were quantified using an Agilent Tapestation using a D1000 screentape and sequenced using an Illumina Next-Seq2000 to generate libraries with 64 – 78 million paired-end 65 bp reads.

### Bioinformatics analyses of RNA-seq samples
Alignment of RNA-seq reads to the human reference genome (T2T-CHMv2.0[63]) was performed with 'subread' (v.2.0.6[64]) using default options for paired-end read data. Read counting was performed with 'featureCounts' with default options for paired-end read data and using strict RefSeq gene annotation. Differential gene expression analysis was performed using the 'edgeR' package (v.3.42.4[65]), with lowly expressed genes removed via 'filterByExpr' and library normalization with the TMM method[66]. Quasi-likelihood F-tests were performed to assess gene-level differential expression between knock-out and wild type samples while controlling for the false discovery rate at the 0.05 nominal level[67].

### Bioinformatics analyses of ATAC-seq samples
ATAC-seq sequence reads were trimmed with 'trimgalore' (v.0.6.10 [https://www.bioinformatics.babraham.ac.uk/projects/trim_galore/]) and aligned to the human reference genome T2T-CHMv2.0 with 'bowtie2' (v2.4.4). Post-alignment filtering was performed to remove unmapped reads and read mates, not primary alignments, reads failing platform, orphan reads, read pairs mapping to different chromosomes, and low-quality reads (MAPQ < 30), while keeping only properly paired reads. Duplicated reads were marked and removed with Picard-tools (v.2.26.11 [http://broadinstitute.github.io/picard/]). Peak calling was performed with MACS (v2[68]) in paired-end mode using default options. The number of peaks identified in WT and KO samples was 50,883 and 68,357, respectively. Reads were counted on genome-wide non-overlapping windows of 250 bp in size using the 'csaw' package (v.1.34.0[69]) and its function 'windowCounts' for all human chromosomes. The midpoint of each fragment was used for counting. Below we load the read count data. Background windows were removed via 'filterByExpr' and library normalization was performed with the TMM method. Quasi-likelihood F-tests were performed to assess window-level differential accessibility between knock-out and wild type samples while controlling for the false discovery rate at the 0.05 nominal level. Window-level results were summarized to the region-level and promoter-level via 'mergeResults' and 'overlapResults' functions, respectively. Promoter regions were defined as the 2kbp window around the transcription start site of each annotated gene.

### Bioinformatic analyses of ChIP-seq samples
ChIP-seq sequence reads were trimmed with 'trimgalore' and aligned to the human reference genome T2T-CHMv2.0 with 'bowtie2'. Post-alignment filtering was performed to remove unmapped reads and read mates, not primary alignments, reads failing platform, orphan reads, read pairs mapping to different chromosomes, and low-quality reads (MAPQ < 30), while keeping only properly paired reads. Duplicated reads were marked and removed with Picard-tools. Peak calling was performed with MACS (v2[68]) in paired-end mode using default options for MORC2 library, and with options '−broad −broad-cutoff 0.1' for the histone modification H3K9me3 to detect broad peaks. IgG libraries were used as controls during peak calling. The number of peaks identified in H3K9me3 and WT samples were 116,550 and 15,366, respectively.

Data analysis was performed in R (v.4.3.0 [https://www.R-project.org/]). Upset plots were generated with the 'UpSetR' package (v.1.4.0[70]). Heatmap and signal profile plots were created with 'deepTools' (v.3.5.1[71]) using merged alignments. Read enrichment coverage was computed in 10 bp windows using extended and centered reads and using read per genomic content normalization (1x depth).

### Preparation of biotin-labelled λ-DNA
Circular λ-DNA lacking methylation (N3011S, Promega) underwent modification through Taq ligase (M0208L) and New England Biolabs (NEB) buffer (B0208S). This process involved the incorporation of complementary oligomers JT41 (5′(P)-GGGCGGCGACCT-3′(Biotin)) and JT42 (5′(P)-AGGTCGCCGCCC-3′(Biotin)) from Integrated DNA Technologies (IDT). The resultant DNA, now labeled with biotin, underwent purification using the AKTA pure system (Cytiva), followed by size exclusion chromatography using Sephacryl S-500 HR resin (Cytiva).

### Single-molecule measurements
We adhered to an established protocol for a single-molecule fluorescence assay[35]. In the assay preparation, we created 6 channels on glass slides by drilling 12 holes on the side. The slides underwent cleaning with 10% detergent, acetone, and Mili-Q water, followed by Piranha solution treatment. Subsequently, they were PEGylated using a 1:80 ratio of biotin-PEG to PEG, incubating in a sodium bicarbonate solution (0.1 M NaHCO3, pH 8.5) for 12-24 hours. After washing with Milli-Q water and drying with nitrogen gas, additional PEGylation with MS(PEG) solution was performed, with the only difference being the incubation time (1-24 hours). Finally, the channels were partitioned with double-sticky tapes and the edges sealed with epoxy glue.

To immobilize 48.5-kbp double-biotinylated λ-DNA, Streptavidin (100 μg/ml) in T50 buffer (40 mM Tris pH7.5 and 50 mM NaCl) was injected through the channels, incubating for 2 minutes to induce Streptavidin biotin-PEG binding. Channels were washed with T50 buffer, and 48.5-kbp double-biotinylated λ-DNA in reaction buffer (40 mM Tris, 50 mM NaCl, 1 mM MgCl2, 0.5 mM TCEP, BSA (0.25 mg/ml), 50 nM Sytox Orange (SxO, S11368 Thermofisher) was applied at a flow speed of 8 μl/m. After DNA tethering, unbound DNA was removed by flowing imaging buffer with 200 μM Desthiobiotin to block exposed biotin-binding sites of the surface-immobilized streptavidin and strepII-tag of MORC2. To monitor MORC2-induced DNA compaction, 50 nM MORC2 in the reaction buffer was flowed into the channels at a speed of 20 μl/m for up to 3 minutes. Real-time DNA compaction was imaged using a 1 mW 561-nm HILO mode laser for SxO excitation. Image recording was conducted with an Olympus UPlanXApo 100x /1.45 lens and an Andor iXon Ultra 897 EM CCD. To obtain kymographs for the subsequent intensity fluctuation analysis, the intensity values of n pixels (n is proportional to the region of

interest (ROI)) from the ROI perpendicular to the extended DNA were summed in each frame (Fig. 5c).

## Single-molecule fluorescence measurement analysis

We measure the DNA compaction kinetics using two approaches 1) cluster-intensity analysis (Fig. 5d–j, Supplementary Fig. 12f–h) and 2) fluctuation analysis (Supplementary Fig. 12a–e) to ensure reliable results.

For cluster-intensity analysis, we first defined the region of interest (ROI) of a cluster area along a compacted DNA using the final frame of recorded images. By reverse-tracking the cluster area through preceding frames, we established the ROI for each frame. The total intensity of the pixels within the ROI was quantified, and the DNA-cluster size in base pairs was deduced by normalizing the cluster intensities to the total intensity of the DNA image and multiplying by 48.5 kbp (the length of λDNA length). We quantified the lag time, defined as the time difference between MORC2 injection and the start of DNA compaction (Fig. 5d, e), and DNA-compaction time was determined by fitting a single-exponential function on the cluster intensities from starting of DNA compaction to saturation (Fig. 5d–f).

For the fluctuation analysis, we confirmed the kinetics obtained from cluster-intensity analysis using a complementary method that reduces potential bias from low-resolution cluster definition. Specifically, as the cluster size or number of clusters increased, the intensity fluctuation decreased. Here, "fluctuation", referred to the variance of the intensity region between adjacent frames. To quantitatively assess the kinetics of DNA cluster formation quantitatively, the mean intensity values below the 50th percentile within the DNA ROI was considered as background noise and subtracted from the DNA signal. For a focused examination of DNA variance, the background intensity was standardized to zero (any 'black (background)' in the DNA figures signifies zero intensity). To ensure uniform DNA length across all frames, the sum of the DNA signal was equalized after background removal. For calculating the sum of variance intensity, the overlapped region between the current and previous frames was compared, and only the variant segments were utilized. In order to determine DNA-compaction time by fluctuation, the data was smoothed using Savitzky-Golay with a moving window of 150 or 300 and a polynomial order of 2, followed by normalization. The starting point was established where the DNA signal exhibited a significant initial fluctuation followed by a decrease, indicating the onset of compaction. Finally, an exponential decay equation was fitted to the fluctuation data to ascertain the compaction time.

## AFM imaging and analysis

To visualize MORC2 conformations, we used 10 nM MORC2 in a reaction buffer containing 40 mM Tris (pH 7.5), 50 mM NaCl, 1 mM MgCl2, and 0.5 mM TCEP with 1 mM ATP, with 1 mM AMP-PNP or without ATP. The sample were deposited onto 0.00001 % (w/v) poly-O-lysine treated mica (Fig. 5k–p, Fig. 6). On the other hands, for visualizing MORC2/DNA clusters using an AFM, we combined 10 nM MORC2 with λ-DNA (48.5 kbp, Promega) at a concentration of 10 ng/μL. The mixture was prepared in a reaction buffer containing 40 mM Tris (pH 7.5), 50 mM NaCl, 1 mM MgCl2, and 0.5 mM TCEP, and incubated for 10 minutes in an E-tube. The prepared samples were then deposited onto mica treated with 0.00001% (w/v) poly-O-lysine. Following a brief rinse with 3 mL of Milli-Q water, the sample was dried using N2 gas blown from a gas gun.

The AFM imaging of the dried sample was performed using a Multimode-8 AFM (Bruker) equipped with a Nanoscope VI controller and Nanoscope version 10.0 software. The ScanAsyst-Air-HR cantilevers (Bruker) used for imaging had a nominal stiffness of 0.4 N/m and a tip radius of 2 nm. Imaging was conducted in the Peak-Force Tapping mode at an oscillation frequency of 8 kHz, with a peak-force setpoint below 150 pN to minimize interaction between the tip and the sample.

For processing the AFM images, Gwyddion version 2.63 software was employed. Background subtraction and transient noise filtering were performed initially. To ensure accurate background subtraction, regions containing particles were masked, and polynomial background corrections (planar or line-by-line) were applied to unmasked areas. Any horizontal scars caused by feedback instabilities or protein adhering to the AFM tip were removed. Plane background subtraction was then applied, and the blind tip estimation method was used to reconstruct the surface and reduce the broadening effects caused by tip convolution.

To quantify dimer and monomer fractions of MORC2 (Fig. 6b, c), we measured particle volumes of MORC2-only images. Using Gwyddion, each protein in the AFM images was masked, and the grain volume was measured, which was considered the volume of a single MORC2 complex. By fitting the MORC2-volume distributions with a two-gaussian fit, we could estimate the number of monomers and dimers of MORC2. In the case of counting conformations of MORC2, we manually analyzed and categorized the conformations (Fig. 6b).

To identify clusters, a cluster-volume threshold was applied based on the volume of a single MORC2 protein (Figs. 5k–p and 6c). For cluster volume measurements, the regions where proteins bound to DNA were manually defined, and their volumes were quantified (Fig. 5p, inset). A threshold value was set at seven standard deviations above the average volume of a single MORC2-dimer complex to ensure a confidence level exceeding 99.9%. Regions with volumes exceeding this threshold were classified as clusters.

## Estimation of the number of MORC2 binding on a unit DNA length

To estimate the number of MORC2 molecules bound to a unit length of DNA, the total number of MORC2 molecules associated with DNA was quantified, and the corresponding DNA length within the image area was calculated. The DNA length was determined using custom made Python scripts developed for automated image analysis. The preprocessing method is improvised from the previous research[38]. The image was repetitively smoothed 3 to 8 times with a Gaussian filter ($\sigma = 2$ nm, window size = 5 nm) to reduce noise and ensure that DNA pixels were connected. This procedure also removed grains at the DNA boundaries, which hindered accurate skeletonization. Pixels where the z-height is higher than 0.1 nm but lower than 2 nm were collected as a DNA, and protein regions were discriminated by size and removed using skimage.measure module. Remaining image was skeletonized using skimage.morphology module. The lengths of DNA skeletons were measured using quasi-Euclidean distance. Detailed parameters were adjusted considering the noise level of the image. Number of protein binding sites was obtained by summing the number molecules individually bound to DNA, and the number of molecules inside the protein cluster. Individual molecules were identified manually as blobs on DNA, larger than 10 nm in size, consistent with the dimensions of a MORC2 dimer. To measure the number of proteins inside the cluster, the volume of the cluster measured in the previous section is divided by the volume of the MORC2 dimer (Fig. 6c). Each microscopy image yielded a single data point for analysis. A one-sided paired t-test was employed to evaluate the statistical significance of differences between distributions of MORC2 binding associated with different mutants.

## Reporting summary

Further information on research design is available in the Nature Portfolio Reporting Summary linked to this article.

## Data availability

CryoEM maps have been deposited in the EM Data Bank with the following accession codes: EMD-45474 (MORC2[PD]) [https://www.ebi.ac.uk/pdbe/entry/emdb/EMD-45474], EMD-45477 (MORC2[1-603])

[https://www.ebi.ac.uk/pdbe/entry/emdb/EMD-45477], EMD-45476 (MORC2^(PD-DNA)) [https://www.ebi.ac.uk/pdbe/entry/emdb/EMD-45476], EMD-45478 (MORC2^(1-603-DNA)) [https://www.ebi.ac.uk/pdbe/entry/emdb/EMD-45478] and EMD-45475 (MORC2^(S87A)) [https://www.ebi.ac.uk/pdbe/entry/emdb/EMD-45475]. Atomic coordinates have been deposited in the Protein Data Bank with the accession codes 9CDF (MORC2^PD), 9CDI (MORC2^(1-603)), 9CDH (MORC2^(PD-DNA)), 9CDJ (MORC2^(1-603-DNA)), and 9CDG (MORC2^(S87A)). The top-ranking atomic coordinates generated by AlphaFold2 for the full-length MORC2 homodimer shown in this study is available in ModelArchive (https://www.modelarchive.org) with accession code ma-1125z [https://www.modelarchive.org/doi/10.5452/ma-1125z]. The raw mass spectrometry data for MORC2 phosphorylation are available from ProteomeXchange via the PRIDE partner repository with identifier PXD053383; and PXD053379 for HDX. All crosslinking mass spectrometry data are available at JPOST with identifier PXD053380. All the ChIPseq, RNAseq and ATACseq data are deposited at NCBI's Genome Omnibus with the GEO Series accession number GSE274916 [https://www.ncbi.nlm.nih.gov/geo/query/acc.cgi?acc=GSE274916]. All other data are available in the main text or as part of the Supplementary or supplementary materials. Original gels, blot images and numerical data used to generate plots are provided in the source data. All unique materials are available upon request with completion of a standard Materials Transfer Agreement from the corresponding authors. Source data are provided with this paper.

## Code availability

The custom Python scripts used in AFM data analysis are deposited in Zenodo (https://doi.org/10.5281/zenodo.15347133)[72].

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

## Acknowledgements

We acknowledge the use of transmission electron microscopes at the Monash University Ramaciotti Centre for Cryo-Electron Microscopy, and the Ian Holmes Imaging Centre at the Bio21 Molecular Science and Biotechnology Institute. We thank the WEHI Cryo-EM Platform, the WEHI Research Computing Platform and the Monash University MASSIVE high-performance computing facility for providing facilities and support. We acknowledge the WEHI Proteomics facility and Protein Production facility. We would like to thank the Melbourne Mass Spectrometry and Proteomics Facility at The Bio21 Molecular Science and Biotechnology Institute at The University of Melbourne. We acknowledge the technical assistance of Stephen Wilcox, Sarah MacRaild and Lilian Wong. We thank Lori Passmore and MRC Laboratory of Molecular Biology, Cambridge, for purchasing the MORC2 synthetic construct. We thank Matthew E. Call, Peter E. Czabotar, Terence Tang, Chloe Gerak, Mihin Perera, Andrew Deans and Pablo Alcon for critical comments on the manuscript. We would like to acknowledge Angel Zhuo Chen and Juri Rappsilber for providing suggestions on the quantitation experimental design for crosslinking mass spectrometry and helping set up the data analysis workflow at WEHI. This work was supported by the Charcot-Marie-Tooth Australia Research Grant, and an

NHMRC Investigator Grant (2026635) to WT; NHMRC Ideas Grant (2010571) to CRK; FSHD Society Fellowship to AG; NHMRC Investigator Grant (2025645) and Chan Zuckerberg Initiative Grant (2021-237445) to GKS; Snow Medical fellowship and CSL Centenary fellowship to SV; NHMRC Investigator Grant (1194345) to MEB; and NIH NIGMS R35GM133751 grant to SD. We acknowledge the Suh Kyungbae Foundation, Creative-Pioneering Researchers Program through Seoul National University, the Brain Korea 21 Four Project grant funded by the Korean Ministry of Education, Samsung Electronics Co., Ltd. (Project Number IO220811-01964-01), and the National Research Foundation of Korea (Project Number RS-2023-00212694, RS-2023-00265412, RS-2023-00218318, and RS-2023-00301976) to JKR. SS is supported by funds from WEHI, WEHI Ventures, the estate of Akos and Marjorie Talon, The University of Melbourne Attraction and Retention Funds, the NHMRC Investigator grant (GNT 2016827), the Australian Research Council Discovery Project grant (DP250100450) and the US Department of Defence Rare Cancer Research Concept Award (HT9425-24-1-0922).

## Author contributions

WT designed and carried out all biochemical, crosslink mass spectrometry, biophysical experiments, prepared samples for cryoEM and analysed the cryoEM data. HV, AL and SS prepared cryoEM samples and collected cryoEM data. HV and SS analysed cryoEM data. JQL and EH performed the FLIM-FRET and FCS experiments. WT and ADG performed the Fluorescence Polarisation ATPase assays. CRK and WT performed ChIP-seq, ATAC-seq, RNA-seq data collection. TMJ performed ATAC-seq. PLB, GKS and RSA analysed ChIP-seq, ATAC-seq and RNA-seq data. VV and LFD performed proteomics experiment and data analysis. WT, TD and LFD performed crosslink mass spectrometry experiments. WT, TD, JY and LFD performed crosslink mass spectrometry data analysis. CSA and SS performed HDX experiments. CS, SS, PSD and SD analysed the HDX data. LDC, SS and SJV performed MORC2 knocked out experiments. CD provided nucleosomes. MEB contributed to interpretation. JVP, KWM, JYL and JKR performed the single molecule DNA compaction assay and atomic force microscopy. SS conceived and supervised the project. WT and SS analysed the data and wrote the manuscript with contributions from all authors.

## Competing interests

The authors declare no competing interests.
