## [Transparent Peer Review file · Nature Communications]

MORC2 is a phosphorylation-dependent DNA compaction machine

Corresponding Author: Dr Shabih Shakeel

Version 0:

Reviewer comments:

Reviewer #1

(Remarks to the Author)

MORC proteins are one of the chromatin remodelers that are involved in various aspects of epigenetic silencing and transcription along with DNA damage responses. The authors of this manuscript realized that the C-terminal domain (CTD) of human MORC2 is an area of extensive post-translation modifications. They employed various biochemical, cryoEM-based structural, biophysical, cellular, and single-molecule approaches to obtain mechanistic insights about how MORC2 protein functions and contributes to chromatin remodeling depending on the phosphorylation status of CTD. The authors show that there are multiple DNA binding domains in MORC2 and the protein acts as a dimer both in vitro and (likely) in vivo. Employing structural and crosslinking tools revealed that DNA binding to MORC2 results in structural changes with different degrees (e.g., less pronounced conformational changes at the ATPase core domain). Interestingly, the phosphorylation status at the C-terminal domain (CTD) somehow affects ATPase activity which occurs at the N-terminal domain (GHKL domain). Additionally, it impacts the speed of DNA compaction. The authors also show that MORC2 acts as a DNA sliding clamp and its preferred in vivo localization areas are open chromatin regions. Overall, the study by Tan et al. deepens our understanding of how MORC2 protein acts and will help our understanding of other MORC proteins as well.

* Line 72-73: Grammar

* Line 111-112: "We mapped and quantified NB of eGFP tagged MORC2 plasmid in a MORC2 knockout HEK293T cell line." Do you mean MORC2 protein rather than MORC2 plasmid?

* Line 252-266: Smart experimental design to show that full-length MORC2 acts as a DNA sliding clamp. Here are our question. Do the authors think that MORC2 is like a closed circle (or a closed circular ring) that encompasses DNA (when ATP or AMP-PNP is present)? If so, have the authors seen the circle-like conformations from AFM images (Fig 5)? It sounds like a silly question, but we are simply trying to understand.

* Line 297: Extended Data Fig 10e should be Extended Data Fig 10f.

* Line 966-967: "the measured apparent brightness (B) by, where I is" – Looks like some words were missing after "by".

* Line 969-973: The authors were discussing HP1 α rather than MORC2. Obviously, this is an error possibly occurred during the manuscript editing process.

* Identifying and utilizing an ATP hydrolysis mutant S87A significantly enhanced their logic since S87L mutant still has a residual ATP hydrolysis activity.

* As mentioned in Line 325-327 and shown in Extended Data Fig 5f, MORC2(S87A) is capable of binding DNA and ATP although it does not hydrolyse ATP. In the AFM data (Fig 5d panel 2), we are unsure which bright dots are DNA-bound MORC2(S87A). If there are any, please indicate the MORC2 mutant using pink arrows (in panel 2) as the authors did for WT in Fig 5d (panel 1).

* SEC-MALS analyses for purified MORC2 proteins convincingly demonstrate that the full length of MORC2 is a homodimer

and the CTD is a dimerization domain. The authors investigated the functional form of MORC2 in vivo and claimed that dimerized MORC2 may be the functional form. According to “Methods”, the authors tagged MORC2 with eGFP and extracted its brightness information (actually, intensity fluctuation information) while utilizing the brightness of monomeric eGFP as a reference. (In the Extended Data Fig 1 legend, it says that HEK293T MORC2 KO cell expressing eGFP was used as a monomer calibration.) However, in the main text, everything was simply referred to as “eGFP”. It is uncertain whether the authors indeed used the monomeric version of eGFP or they simply assumed that eGFP is a monomer. If they indeed used the monomeric version of eGFP, please clearly state in the main text that what they used was a monomeric eGFP. It is especially important since eGFP itself is prone to dimerize (PMID 11988576; PMID 38409224; also see PMID 22484850 for the fluorescence tag clustering in general). If the authors used a “regular” eGFP, they need to re-do the experiments with MORC2 labeled with the “monomeric” eGFP before claiming the dimeric status of MORC2 in vivo.

* The legends of Extended Data Fig 1 are “e-f. Quantification of the fractional contribution of MORC dimer (green), and oligomer (red) in WT vs. MORC2 KO cell”. However, there are not any green or red colors in those two figure panels. In fact, the Extended Data Fig 1 e-f are about in vivo dimer and oligomer fractions with and without neocarzanostatin (NCS). The figure panels and the legends do not match.

* Line 114-116: About 20% of the MORC2 population is present as dimer. And about 1% is present as oligomer according to Extended Data Fig 1 (in the absence of NCS). Does that mean that ~80% and ~60% of the MORC2 population are monomers in the absence and presence of NCS, respectively? If so, do the authors think that the functional protein form is a dimer although there are more monomers in vivo? This question is closely related to another question above relating to using eGFP as the monomeric form reference.

* ChIP-seq, ATAC-seq, FLIM-FRET, and RNAseq convincingly show that MORC2 tends to bind H3K9me3-depleted regions. The tendency of MORC2’s binding to open chromatin is intriguing given that MORC2 associates with HUSH complex.

* Line 288: eGFP-H2B and mCherry-H2B (FP at the N-terminus), Line 942: H2B-eGFP and H2B-mCH (capital H) (FP at the C-terminus), Line 944-945: H2B-eGFP and H2B-mCh (small H) (FP at the C-terminus) Make your notations consistent.

* Line 130 claims that dephosphorylated MORC2 may be a stronger DNA binder. In Line 244-245, it says that MORC2(PD) hydrolyzes ATP 2-fold faster than the wild-type MORC2. Fig 5c shows that DNA compaction time for PD is shorter than that of wild-type. Those three results from different approaches seem to agree with each other. It will be interesting and informative to see discussions on how the first two results are intimately related to the last result.

* Line 1145-1146: “To obtain kymograph” -> Fig 5 does not contain any kymograph figure. If the authors needed kymographs for the subsequent intensity fluctuation analyses, rewording would clarify the situation. For example, “To obtain kymographs for the subsequent analysis” instead of “To obtain kymographs”.

* For single-molecule assay, Sytox-Orange-labeled double-biotinylated lambda-DNA was tethered and introduced MORC-2 proteins at 20 ul/m (ul/min). Then, the authors used intensity fluctuation changes to indirectly determine when DNA compaction occurs. This approach led the authors to conclude that phosphodead MORC2 compacts DNA 3-times faster than WT MORC2 in a statistically significant manner. We are not convinced of this important conclusion. Because of the presence of flow while applying MORC2 protein, DNA is biased towards the buffer flow direction while two DNA ends are still tethered. This is obvious in the provided movies where the intensity of the upper part of DNA is roughly twice higher since two DNA segments overlap due to flow. We guess that the flow will make the determination of DNA compaction onset complicated as the authors also indirectly implicated this difficulty by saying that “Although the appearance of clusters was sometimes not obvious due to the resolution limit.” If DNA compaction is a slow process, the uncertainty may be still fine. But, both WT and PD MORC2 seem to be robust (fast) DNA compactor, even some uncertainty might lead to big uncertainty in the DNA compaction time. Furthermore, the compaction time of WT and PD (Fig 5c) were obtained from only 11 and 6 events, and any purified protein has slightly different activities for each purification (although using exactly the same protocols).

To help us and the readers convinced, at minimum, (1) more information on the data analyses such as an SI figure of intensity fluctuation graph with the exponential decay fitting and (2) more data points (instead of 6 and 11 DNAs – ideally from multiple experiments (for sure) with proteins from more than 1 purification (not required)) are suggested.

* Line 1204-1205: We are thankful for the authors depositing various data including CryoEM maps and atomic coordinates for readers. As a minor point, it says that “Original gels, blot images and numerical data used to generate plots are provided in the source data.” We downloaded all the files for review, but the original gels and blot images are missing. (What we see is chopped gel images such as Extended Data Fig 1a.)

Reviewer #2

(Remarks to the Author)

The Microchidia (MORC) family of chromatin remodellers are associated with chromatin compaction and epigenetic silencing. The manuscript by Tan et al. describes a mechanistic study of MORC2, where they aimed to characterize the functional roles of MORC2 domains in dimerization, DNA binding and DNA compaction, and how these roles are regulated by phosphorylation. A strength of this manuscript is the multiple avenues of investigation presented, which provide complementary datasets that inform on the MORC2 mechanism of action. Overall, the work provides some interesting insights into human MORC2 function that can be evaluated against related published work (most notably Kim et al. 2019 on

c.elegans MORC-1); however, there are significant problems with aspects of the manuscript that should be addressed before publication. In particular, the authors should take care to ensure that the data presentation is clear, that the writing conveys the intended messages and that conclusions are not overstated.

Major concerns:

Cellular 'number and brightness (NB)' analysis - whilst evidence in the manuscript (and elsewhere) support the conclusion that MORC2 forms a dimer, there are several issues with this part of the paper:

1. The paper should include relevant controls / calibration datasets (eGFP cells).
2. The authors state that they obtain pixel resolution but the interpretable spatial resolution of the data (not pixel size) should be defined to make this meaningful for the reader.
3. ED figure 1e and 1f appear to plot the fraction of pixels assigned to dimers (-NCS = ~10 %; +NCS = ~20 %) or oligomers (-NCS = ~0.2 %; +NCS = ~0.5 %) - what is being shown should be made clearer in the figure/legend (fractions of what?; and what values are indicated by the box plot?) and the figures should include individual data points.
4. The values in the box plots (dimers -NCS = ~10 %; +NCS = ~20 %) do not seem to match those in the text (dimers -NCS = ~20 %; +NCS = ~40 %).
5. The methods section refers to the wrong protein (HP1). The authors should confirm that the methods and statistics given here refer to calibrations and experiments carried out for this study. They select cells with low eGFP, how are fractions of monomer vs oligomer impacted by absolute eGFP intensity?
6. Finally, as the results appear to plot the fraction of pixels assigned to each state of the protein, to what extent does the selection of area impact the results? The authors should include a panel showing the areas selected for analysis for the two conditions.

MORC2 as a sliding clamp – whilst it is entirely possible that MORC2 forms a dimer that encloses and can slide on DNA, as shown for c.elegans MORC-1, there are several issues with the assay used to test this hypothesis in this manuscript, meaning the data do not substantiate the claim. Problems include (but are not limited to):

1. What is being quantified in the gels? The way in which the data has been analysed and interpreted are very unclear. Initially 100 nM of the MORC2 protein is mixed with 25 nM of DNA, meaning that maximally 25% of the protein is DNA bound (assuming a 1:1 interaction). When the lower concentrations of competitor DNA (25 nM) are added, they are likely to interact with the free protein (no competition required), so how these results should be interpreted (particularly in the case of the circular DNA substrate experiment) is not clear.
2. At the higher concentration of competitor (20-fold excess), 74% of the competitor DNA is apparently shifted by the protein (MORC2 FL, 60-bp linear DNA condition), whereas a 1:1 interaction would predict a maximum of 5%. How do the authors explain this result?
3. The quantification of the result with the truncated 1-603 MORC2 also seems highly improbable based on a visual inspection of the images in the figures (50% competitor DNA is quantified as shifted).
4. The comparison between the linear and circular DNA is further complicated by the fact that the DNA constructs are not equivalent in length (and possibly not sequence?). ED fig. 9 suggests that MORC2 interacts with longer DNA with higher affinity, confounding interpretation of the results.
5. The labelling of the lanes of the gels appear to be incorrect and inconsistent between the main figure and the equivalent extended data figure.
6. The concentrations plotted in Fig. 3e, described in the methods and described in the figure and ED figure legends are inconsistent.
7. The length of circular DNA used differs in the manuscript, figure and ED figure legends and methods (95, 101 or 105 bp).
8. In general, the authors need to consider whether this approach addresses the question being asked and if so, provide a much more comprehensive description of how the experimental setup and results can be interpreted to support their hypotheses.

MORC2 FLIM-FRET –

1. Line 288-290 – IF staining with MORC2 or (not and) H3K9me3 antibodies. The data does not show that MORC2 is localized with regions depleted of H3K9me3 (they are imaged in different experiments), it shows that is localized with areas of lower FRET according to the current data interpretation. H3K9me3 is correlated with regions of higher FRET, so only indirect conclusions can be drawn.
2. Fig. 4h – It is not clear what this figure is showing. The y-axis is apparently in units of pixels and the data appear more suited to a box plot representation than the lines shown. It is not clear why there are only 4 cells recorded for the H3K9me3 data. It is difficult to interpret these results.

Other comments:

- Care should be taken with all figures to ensure that they are appropriately clear, labelled (incl. axis, units etc.) and consistent with the legends, text, and methods sections. The current version of the manuscript has multiple errors and inconsistencies that need to be addressed but are not listed exhaustively here.
- In the discussion, many hypotheses generated by data in the paper are conveyed as statement of fact, when the data do not support this (e.g. lines 388-389: "As MORC2 is a DNA sliding clamp that preferentially binds to open DNA...", the authors need to draw a clearer distinction between proposed mechanisms that are consistent with the data and hard conclusions that are conclusively demonstrated.
- For all quantified EMSAs, the region of the gel that has been quantified should be indicated. The current quantification data

is very hard to interpret.

- Fig 1b would be easier to interpret if it included a panel with a MORC2 colored by domain (like Fig. 1a)
- Lines 168-171 are not clear.
- Several experiments use the MORC2PD mutant to evaluate the role of phosphorylation in regulating MORC2. Given the authors show that phosphorylation state of MORC2 can be manipulated with PAK1 kinase or PPase, it would be more direct and informative to assess the role of phosphorylation on MORC2 function after treatment of the protein with these enzymes. Phosphorylation or dephosphorylation should be demonstrated.
- Lines 72-75 aren't clear and would benefit from rewriting.
- Line 321-323 – “indicating phosphorylation of PIM motif in MORC2 CTD is crucial for dictating the speed at which DNA compaction would occur” – This result shows that mutating these residues increases the speed on DNA compaction in this assay. This implies that phosphorylation (if the WT protein in this assay was phosphorylated) might negatively regulate the speed of compaction. The authors could directly test this hypothesis by testing compaction with WT MORC2 that is either phosphorylated (after PAK1 kinase treatment) or not phosphorylated (after PPase treatment).
- Line 325-328 – Similarly to the above, the authors test the role of ATP hydrolysis in DNA compaction using a mutant (the methods seem to indicate that the assays were performed without ATP, but this is presumably just an omission in the writing). Whilst this is a valid approach, supporting data +/- ATP or ATP analogues would also help to confirm this finding, particularly given that MORC-1 has been shown to compact DNA in a manner that is nucleotide binding but not hydrolysis dependent (Kim et al. 2019).
- Figure 5f – Three AFM images are proposed to show MORC2 interactions with DNA (lines 335-337), however these images are only anecdotal evidence of the possible shape of MORC2 and do not address the stated aim of focusing on ‘whether DNA binding changes the oligomerization state of MORC2’. Supporting data showing analysis of MORC2 alone, in the presence and absence of AMP-PNP (or ATP), to compare structural rearrangements upon ATPase dimerization would be informative. Scale bars and comparison to the scaled cryo-EM structures of MORC2 would help demonstrate that the particles being shown are MORC2.
- ED Figure 6f is very unclear and does not currently add anything to the interpretation of the data. What is the relevance of the ‘representative intra-protomer crosslinks’? How many (and which) crosslinks identified experimentally did not fit the AlphaFold prediction? Why was the AlphaFold structure (no DNA) validated with the crosslinking data with DNA? Perhaps a comparison of each of the crosslinking datasets with the model would be more informative, highlighting how many experimentally derived crosslinks fit or do not fit the model (within a given distance tolerance), and where these are found in the structure.

Reviewer #3

(Remarks to the Author)

Reviewer #4

(Remarks to the Author)

In this manuscript Tan et al provide an in-depth biochemical and structural characterization of the chromatin remodeler MORC2, as well as functional analysis in the context of genomic binding and gene expression regulation. They show that MORC2 is a functional dimer and its dimerization depends on the flexible C-terminal domain (CTD). ATP binding increases the proximity of the two protomers. MORC2 is a DNA sliding clamp with DNA binding sites located in the ATPase domain (CC1, CW-CC2) and CTD (CC4); using X-link mass spectrometry and HDMX the authors show reorganization of ATPase and CTD regions upon DNA binding. Phosphorylation of the CTD reduces ATPase activity without affecting DNA binding. The authors further show that MORC2 can compact DNA in vitro in ATPase-dependent manner. The relevance of this effect is unclear though as ChIP experiments showed that MORC2 generally binds to promoter regions and open chromatin with only a small portion found at H3K9me3-enriched heterochromatic loci. Moreover, MORC2 knock-outs cells don't show major changes in transcription and chromatin accessibility, suggesting that MORC2 exerts local rather than global effects on chromatin.

Overall, a lot of data is provided in this manuscript but insights into the functional connection of these findings are missing. Specifically:

1) Are the genes that are upregulated in RNAseq and more accessible in ATACseq for MORC2 KO cells also bound by MORC2 based on ChIP and contain H3K9me3 marks in WT cells? If so, what is the function of these genes and why would only this small subset require MORC2-mediated compaction? Did you analyse the differential expression of repetitive elements in your RNA-seq data?

2) In PMID: 23260667 they showed that MORC2 promotes chromatin relaxation in ATPase-dependent manner after radiation-induced DNA damage based on the micrococcal nuclease assay. In PMID: 31616951 they show that MORC2 induces chromatin relaxation even without DNA damage and that this is dependent on its PARylation. In this manuscript you show that MORC2 promotes chromatin compaction in vitro. Cellular effects are missing. I think it's crucial to explain how the same chromatin remodeler can have two opposite effects on the chromatin state through its ATPase activity. This must be tightly regulated – how? Would overexpression of MORC2 increase chromatin compaction and make it more global? What is the phenotype of the phospho-dead mutant in cells, which shows higher compaction in vitro? I think that showing the cellular relevance of the in vitro compaction effect is crucial.

3) 'The neuronal MORC2 mutation, R252W results in defective ATPase activity and hyperactive transcriptional silencing^{11,21}, suggesting that ATP hydrolysis drives chromatin remodelling during HUSH-mediated silencing.' ATPase activity was reported to be reduced (not defective) in Douse et al. 2018. Higher efficiency of HUSH-dependent epigenetic silencing when ATP hydrolysis is reduced would suggest that ATPase activity relaxes chromatin (rather than compacting it).

4) The EMSA experiment in Extended Data Fig 2 is not conclusive. The example EMSA in panel b for the phosphatase-treated samples shows overall lower amount of DNA and the signal for the highest concentration of MORC2 (200 nM) is very weak. In panel c quantification this sample is completely missing? EMSA should be repeated and complete quantification provided. EMSA in Extended Data Fig 5f is much cleaner. A more quantitative assay could also be performed (e.g., SPR as in Fig 3). Please also indicate the length of DNA (I assume 60 bp).

The conclusions from the current results that dephosphorylated MORC2 is a stronger DNA binder are at variance with (i) the authors' results from Fig 3b whereby SPR showed similar binding affinity for WT and phospho-dead mutant, and (ii) the published data (PMID: 23260667) showing that MORC2 phosphorylation after DNA damage at S379 promotes chromatin binding.

5) Structure results should always start with data for the WT and full-length constructs to rationalize the use of mutants. Why was the structure of the full-length ATPase-dead mutant not included in Figure 1? If phosphorylation of the CTD affects dimer flexibility, why didn't the authors dephosphorylate the protein before preparing the grids? I cannot comment on the quality of cryo-EM datasets and I hope that another reviewer will cover these aspects.

6) EMSA experiment in Extended Data Fig. 8a for 496-1032 is unclear as it seems that DNA got stuck in the wells (there is no unbound DNA. Thus the corresponding curve in Fig 2d is unreliable. In Fig. 2d the curve for 496-603 isn't visible. Please clarify.

7) EMSA from Extended Data Fig. 9 should be shown before the first EMSA in Extended Data Fig 2 to justify the use of 60 bp DNA for further experiments.

Minor (mainly text editing for clarity)

1) line 72/72: Remove: 'There are several'

2) line 74/75: This statement is unclear and should be rephrased: 'resulting in histone H2AX phosphorylation to cause chromatin remodeling'

A suggestion how to rephrase: ...and promotes chromatin relaxation and gH2AX phosphorylation.

3) line 90: domains

4) line 90: possesses

Version 1:

Reviewer comments:

Reviewer #1

(Remarks to the Author)

In the revised manuscript, the authors satisfactorily addressed this reviewer's comments and concerns. They went extra miles and provided the convincing revised manuscript.

Reviewer #2

(Remarks to the Author)

In their revised manuscript and accompanying rebuttal, Tan et al. have addressed many of the points raised in my initial review and improved their manuscript. I am grateful for their efforts in this regard. Overall, the manuscript includes a lot of interesting data and comes together to produce an informative study that advances our understanding of MORC2, its interaction with DNA and its regulation. The results are mostly consistent with another recently published study (Fendler et al. 2024 PMID : 39739841 ; with the exception of the impact of DNA binding on ATPase activity), supporting many of the conclusions. However, several important issues remain or arise from the new data that has been included. In particular, the new EMSA experiments presented in Figure 3 and ED Figure 10 have several problems.

Major:

Figure 3c/d and ED Fig 10 have many serious problems with the experimental design, data and analysis. They cannot be used to draw any meaningful interpretations. Problems include (but are not limited to):

In the rebuttal, the authors state "In the new experiment (Fig 3c-d) we observe that ~25% of the competitor DNA displaced linear DNA." and in the main text "the IRDye800-labeled DNA effectively displaced the linear DNA but not the circular DNA". However, this is not what the experiment shows. The experiment does not measure displaced DNA, it measures binding of a

secondary DNA species. This manuscript describes how MORC2 has multiple DNA binding sites. Moreover, according to the quantification provided, both Fig. 3c/d and ED Fig. 10 suggest that >40% of 'competitor DNA', which is added in 10-fold molar excess over total protein (assuming the 80 nM protein concentration is for the dimer?) is shifted, suggesting that each MORC2 complex can interact with multiple DNA molecules. Another recent study convincingly shows MORC2 binding/entrapping multiple DNAs (Fendler et al. 2024 PMID : 39739841). Knowing that MORC2 can bind to multiple DNAs, and only measuring the signal of the competitor DNA, how do the authors interpret their data? Could any conclusions about competition or MORC2 acting as a sliding clamp be made on the basis of this data? The authors should consider these questions before addressing the technical problems with the data included in the manuscript.

In the rebuttal, the authors state "Given that we now use longer length DNA, therefore, there is a possibility of more than one MORC2 molecule binding to a single DNA". This may be possible, but in this particular experiment, I think the opposite is more likely to be true, that each MORC2 is binding to multiple DNAs.

Most seriously, the data included in Fig 3c (in particular), is unusable. As is evident from the figure panels and the source data, the gels have overrun and very little of the free DNA is present in the gel / images. It is not clear to me what the authors are suggesting they have quantified. comparison between column 1 (or panels in ED Fig 10) and columns 2 and 3 of Fig. 3c, shows that the free IRDye-DNA that should be present at the bottom of the gel is largely absent.

The data are apparently representative of experiments run twice, but it is unclear what is plotted in Fig 3d. Are these averages of the quantification or just quantification from the data shown in the figure. There appear to be some partial error bars, but it is not clear what these represent?

Although less severe (the competitor DNA is still visible), the data for ED Fig 10 also has issues with the gels and the way in which they have been cropped, which has removed signal. The quantification also does not appear reasonable. Visually, it seems inconceivable that ~50% of the IRDye-DNA is 'shifted' in the 800 nM conditions, when there is such a strong band for the free DNA compared to a relatively weak band for the MORC2-DNA. This indicates a problem with the quantification, but it remains unclear precisely what is being quantified (ED Fig. 3a begins to address this, but doesn't show exactly what is quantified – is it loss of substrate, or formation of product?

For all EMSAs reported in the manuscript, the authors need to ensure that the gels are not inappropriately cropped, and that pixel values are not saturated in the images that are being quantified, particularly in the highest concentrations of DNA. If pixels are saturated, the full extent of the signal cannot be quantified. This appears to be what is shown in ED Fig. 3a, where certain peaks indicate saturation.

The middle column of ED Fig. 10 shows multiple bands for MORC2(ATPase) + linear DNA. How do the authors explain these? They are currently labelled as 'Free DNA' in the figure, but it appears that this sample contains DNA+MORC2. The experiments in both figures should also include a condition for the free linear or circular DNAs before mixing with the MORC2 to demonstrate the position of the free-DNA band.

There are other issues with the design these experiments, the data presented and the interpretation, but overall, these assays appear unusable.

Minor:

Extended data figure 1d/e:

In their rebuttal the authors state that they have clarified the description of the NB analysis in the methods, but I cannot see where this has been done.

In their rebuttal, the authors did not address my question regarding how the selection of the area impacts the results of the monomer/dimer/oligomer NB analysis. The authors include the same panels showing examples of areas selected as were included in the first manuscript, but it is not clear why these areas were chosen. How would the results compare if the whole nuclei were analysed, rather than selected regions?

Line 1621 missing text: "data acquisition was recorded is. Column"

ED Fig. 4a,b:

The '% DNA bound' numbers in (a) do not appear to correspond with the equivalent numbers plotted in (b) – see highest two concentrations for 'untreated' and 'PAK1' conditions.

ED Fig. 4a (untreated), 4d (WT), 5e (WT) and 9a (1-1032) appear to be repeats of the same (or similar) experiments, but with different labelling and different results. Are 4a (untreated), 4d (WT), 5e (WT) replicates, or have the proteins been treated differently, for example with phosphatase treatment? One figure legend says the data is a representative result from 3 independent experiments, one says 2 independent experiments and the other seems to be a single experiment. If these differ, it needs to be clear how they differ. If they are repeats of the same experiment under the same conditions, then the variability in the results needs to be explained. ED Fig. 9a appears to be a very similar assay but with different protein concentrations used in the titration (based on the quantification shown in Fig. 2d).

In general, the manuscript would greatly benefit from consistency of notation between the different protein constructs used.

Similar to the above, have MORC2 mutants (e.g. S87A in ED Fig. 5f) been treated with phosphatase before being used for EMSA and SPR experiments? This wasn't obvious from the text, figure legend or methods.

ED Fig. 4c / Fig 3b:

Has the 'WT' sample in 3b been pre-treated with λ PPase? Or is it the same as the 'untreated' sample in ED Fig. 4c? This should be clear in the description in the text and in the figure legends. If the 'WT' sample used in 3b has not been treated with λ PPase, why is there such a difference in the measured affinity with the 'untreated' sample in ED Fig. 4c? It should be made clear in the methods and throughout the manuscript whether the constructs being used have been pre-treated in anyway before being used in the assays, to distinguish them from the 'untreated' purified sample.

Description of SPR conditions in methods: "a 8-point concentration series (2-fold serial dilution, 7-500 nM)" and figure legend "protein concentrations of 0, 4, 8, 16, 33, 63, 125, 250 and 500 nM" are inconsistent. This is a small detail, but small errors like this (and other more significant errors like those listed above) were found throughout in both first and second drafts of the manuscript, making it difficult to assess the methods/data and reducing confidence in the quality of the work.

Fig 4 – FLIM-FRET:

The manuscript asserts that it uses these experiments "to directly visualize MORC2 binding to chromatin", but these experiments do not directly visualize MORC2 binding to chromatin, or its colocalization with (or not) H3K9me3. The experiments image MORC2 or H3K9me3 and histones. They then draw a correlation between the co-localisation of MORC2 or H3K9me3 with relative proportion of chromatin compaction measured by FRET. The authors state that "MORC2 is localised with areas of lower histone FRET" and "H3K9me3 is present at areas with high FRET", but again, this doesn't seem to be what the data shows. The figure plots the fraction of pixels exhibiting FRET (as an indication of compacted chromatin) in areas that colocalise with MORC2 or H3K9me3 (foci) or do not colocalise (nucleoplasm). Whilst there appears to be a large difference in the proportion of compacted chromatin in the nuclei of each cell imaged (ranging from about 5%-36%; possibly reflecting the stage of the cell cycle?), there appears to be little difference between the average fraction of compacted chromatin inside vs outside of the foci and between MORC2 foci vs the H3K9me3 foci, particularly when taking into account the different scaling of the y-axes. The two plots should be shown on the same scale and reporting the normalised fraction of FRET as displayed in Liang et al. 2024 PMID: 38265456 Fig. 2h would be informative).

Similarly, the text in the results section (lines 417-420) is confusing and I am not sure that it accurately conveys what the figure shows. The text suggests that MORC2 is found in the 'nucleoplasm', whilst H3K9me3 is found in 'foci', however, my understanding of the figure, based on the Methods section, is that it compares "quantification of the fraction of compact chromatin within high intensity MORC2/H3K9me3 region versus outside the IF-guided mask (nucleoplasm) in total". Could the authors please clarify this in the text and figure legend.

The y-axis on the plots shows 'FRET faction', but I believe this should be 'FRET fraction (% pixels)'

Reviewer #3

(Remarks to the Author)

Reviewer #4

(Remarks to the Author)

Thank you for providing the revised manuscript. Unfortunately, the new experiments do not provide the much-needed link between in vitro assays and cellular function of MORC2. There are several remaining issues, which needs to be at least dealt with in writing.

1) page 8, line 76: 'suggesting that ATP hydrolysis relaxes chromatin during HUSH-mediated silencing'
This statement doesn't make sense. Chromatin relaxation and silencing don't go together. Please rephrase: 'suggesting that ATP hydrolysis impairs HUSH-mediated silencing by promoting chromatin relaxation.'

2) 'These modifications lead to histone H2AX phosphorylation, which in turn relaxes chromatin to facilitate the DNA damage response.'

This is wrong. gH2AX doesn't lead to chromatin relaxation. Please rephrase: 'These modifications promote chromatin remodeling and DNA damage response.'

3) Extended Data 4D: only one replicate? All figures: number of replicates should be indicated, data points should be shown.

4) Labels in main figures are messed up.

5) 'MORC2 was previously shown to repress retrotransposons like LINE-1 marked by H3K9me3 to initiate epigenetic silencing. Our data aligns with this finding...'

It does not.

'Our findings highlight ATP hydrolysis-driven DNA compaction as a major function of MORC2, which is consistent with its role in gene silencing, presumably at promoters where it is localised.'

Again, your cellular data doesn't support this claim.

'The differences in gene expression and chromatin accessibility patterns between the wild-type, phosphodead and phosphomimetic mutants suggest that MORC2 phosphorylation plays a key role in chromatin structure regulation.'

Your phosphomutant reconstitution data does not show major effects on chromatin accessibility. Therefore, you cannot state that phosphorylation plays a key role in chromatin structure regulation.

The new experiments did not demonstrate a functional importance of in vitro chromatin compaction, which is in contrast with previous findings that MORC2 is required for HUSH-mediated silencing of H3K9me3 regions (PMID: 28581500). This can be due to i) a different cell line, ii) lack of analysis of retroviral elements, iii) suboptimal assessment of chromatin accessibility (DIVA approach was shown to be superior to ATAC-seq for the analysis of repetitive regions in PMID: 28581500). Before acceptance of this manuscript, it is crucial that the authors comment on this in the Abstract and the Discussion by providing the above-mentioned possible reasons and by clearly stating that 'In contrast to previous work, in vitro chromatin compaction by MORC2 was not shown to have functional significance in cells with regard to chromatin accessibility and transcription regulation.'

Version 2:

Reviewer comments:

Reviewer #2

(Remarks to the Author)

In their revised manuscript, the authors have significantly improved the quality of the data and analysis for key experiments, thus addressing my major concerns.

Response to reviewers' comments

We appreciate all the comments from the reviewers. Here is our point-by-point response in blue.

Reviewer #1 (Remarks to the Author):

MORC proteins are one of the chromatin remodelers that are involved in various aspects of epigenetic silencing and transcription along with DNA damage responses. The authors of this manuscript realized that the C-terminal domain (CTD) of human MORC2 is an area of extensive post-translation modifications. They employed various biochemical, cryoEM-based structural, biophysical, cellular, and single-molecule approaches to obtain mechanistic insights about how MORC2 protein functions and contributes to chromatin remodeling depending on the phosphorylation status of CTD. The authors show that there are multiple DNA binding domains in MORC2 and the protein acts as a dimer both in vitro and (likely) in vivo. Employing structural and crosslinking tools revealed that DNA binding to MORC2 results in structural changes with different degrees (e.g., less pronounced conformational changes at the ATPase core domain). Interestingly, the phosphorylation status at the C-terminal domain (CTD) somehow affects ATPase activity which occurs at the N-terminal domain (GHKL domain). Additionally, it impacts the speed of DNA compaction. The authors also show that MORC2 acts as a DNA sliding clamp and its preferred in vivo localization areas are open chromatin regions. Overall, the study by Tan et al. deepens our understanding of how MORC2 protein acts and will help our understanding of other MORC proteins as well.

We thank the reviewer for appreciating and acknowledging the importance of our work.

* Line 72-73: Grammar

Addressed in lines 77-78

* Line 111-112: "We mapped and quantified NB of eGFP tagged MORC2 plasmid in a MORC2 knockout HEK293T cell line." Do you mean MORC2 protein rather than MORC2 plasmid?

Yes, we meant MORC2 protein. We have now changed it to protein instead of plasmid.

* Line 252-266: Smart experimental design to show that full-length MORC2 acts as a DNA sliding clamp. Here are our question. Do the authors think that MORC2 is like a closed circle (or a closed circular ring) that encompasses DNA (when ATP or AMP-PNP is present)? If so, have the authors seen the circle-like conformations from AFM images (Fig 5)? It sounds like a silly question, but we are simply trying to understand.

We appreciate reviewer's comment on our DNA competition assay to show that MORC2 is sliding clamp. Based on other reviewers' comments we have further improved those assays.

We also thank the reviewer on using AFM to provide direct evidence for MORC2 forming a closed circular ring in presence of ATP or its analog. Our new AFM data confirm that MORC2 adopts a more closed, circular conformation upon binding AMPPNP or ATP, likely due to dimerization of its ATPase domains ("heads") (new Fig. 6a–f). In contrast, the ATP-binding-deficient N39A mutant, which cannot dimerize via its ATPase domain, exhibits a more open, V-shaped structure with dissociated heads (Extended Data Fig. 13a). These findings align with our cryo-EM studies, which also reveal ATP-dependent ATPase domain dimerization (Fig. 1b).

Furthermore, the ATP-binding-deficient N39A mutant does not bind DNA, whereas the S87A mutant, which binds but does not hydrolyze ATP, retains DNA-binding ability (new Fig. 6g–i). Consistently, ATP-dependent DNA compaction by MORC2 WT and the PD mutant is absent in both the ATPase-dead S87A and ATP-binding-deficient N39A mutants (Fig. 5e–f).

As the reviewer anticipated, we did observe closed, circular MORC2 upon DNA binding (Fig. 6j). However, this conformation appears at a lower frequency, possibly due to its localization within DNA clusters, making visualisation more challenging.

Collectively, these findings support the idea that ATP binding promotes a circular or clamp-like structure in MORC2, enhancing its DNA-binding affinity and facilitating efficient DNA compaction.

* Line 297: ED Fig 10e should be ED Fig 10f.

We have changed the call out to the correct figure and panel (now ED Fig 11d)

* Line 966-967: "the measured apparent brightness (B) by, where I is" – Looks like some words were missing after "by".

Addressed in line 2480

* Line 969-973: The authors were discussing HP1 α rather than MORC2. Obviously, this is an error possibly occurred during the manuscript editing process.

Corrected in lines 2482-2486

* Identifying and utilizing an ATP hydrolysis mutant S87A significantly enhanced their logic since S87L mutant still has a residual ATP hydrolysis activity.

Thanks for appreciating our use of the S87A mutant. We have emphasised your point in lines 165-166.

* As mentioned in Line 325-327 and shown in ED Fig 5f, MORC2(S87A) is capable of binding DNA and ATP although it does not hydrolyse ATP. In the AFM data (Fig 5d panel 2), we are unsure which bright dots are DNA-bound MORC2(S87A). If there are any, please indicate the MORC2 mutant using pink arrows (in panel 2) as the authors did for WT in Fig 5d (panel 1).

We appreciate the reviewer's request for clearer identification of DNA-bound MORC2 (S87A) in the AFM data. In response, we have now labelled MORC2 (S87A) in the revised Fig. 5l and provided additional evidence of its DNA-binding ability in Fig. 6g and 6i. Furthermore, we have included a schematic representation and quantified the number of bound proteins per unit DNA length in Fig. 6i to enhance clarity.

* SEC-MALS analyses for purified MORC2 proteins convincingly demonstrate that the full length of MORC2 is a homodimer and the CTD is a dimerization domain. The authors investigated the functional form of MORC2 in vivo and claimed that dimerized MORC2 may be the functional form. According to "Methods", the authors tagged MORC2 with eGFP and extracted its brightness information (actually, intensity fluctuation information) while utilizing the brightness of monomeric eGFP as a reference. (In the ED Fig 1 legend, it says that HEK293T MORC2 KO cell expressing eGFP was used as a monomer calibration.) However, in the main text, everything was simply referred to as "eGFP". It is uncertain whether the authors indeed used the monomeric version of eGFP or they simply assumed that eGFP is a monomer. If they indeed used the monomeric version of eGFP, please clearly state in the main text that what they used was a monomeric eGFP. It is especially important since eGFP itself is prone to dimerize (PMID 11988576; PMID 38409224; also see PMID 22484850 for the fluorescence tag clustering in general). If the authors used a "regular" eGFP, they need to re-do the experiments with MORC2 labeled with the "monomeric" eGFP before claiming the dimeric status of MORC2 in vivo.

We can confirm that the eGFP tagged to MORC2 constructs was monomeric. To make it clear we have changed the notation to meGFP in every instance of its use.

* The legends of ED Fig 1 are "e-f. Quantification of the fractional contribution of MORC dimer (green), and oligomer (red) in WT vs. MORC2 KO cell". However, there are not any green or red colors in those two figure panels. In fact, the ED Fig 1 e-f are about in vivo dimer and oligomer fractions with and without neocarzanostatin (NCS). The figure panels and the legends do not match.

The ED 1e-f quantification is based on dimer and oligomer from ED Fig 1d. We have added supporting data to panel d to clarify this and re-written the figure legend with new ED Fig 1 d-e (please note: we have merged old panels e and f into one).

* Line 114-116: About 20% of the MORC2 population is present as dimer. And about 1% is present as oligomer according to ED Fig 1 (in the absence of NCS). Does that mean

that ~80% and ~60% of the MORC2 population are monomers in the absence and presence of NCS, respectively? If so, do the authors think that the functional protein form is a dimer although there are more monomers in vivo? This question is closely related to another question above relating to using eGFP as the monomeric form reference.

Yes, it is likely that the majority of the MORC2 population exists in a monomeric state, an estimate which is similar to the one determined recently for another chromatin remodeller, heterochromatin protein 1a (HP1a) (Lou et al, NAR 2024: <https://doi.org/10.1093/nar/gkae720>). Lou et al discovered that HP1a in its monomeric form facilitated nucleosome spacing whereas the dimer is essential for chromatin compaction and maintenance. Similarly, MORC2 monomer may also have yet-to-be-discovered functions, although the chromatin remodelling is likely performed by MORC2 dimer. Furthermore, the measurements done in our analysis considers the global level of MORC2 in cells, where we cannot differentiate the oligomeric state of MORC2 bound to chromatin vs the freely diffusing one. Perhaps almost all chromatin-bound MORC2 is in dimer form as indicated by our in vitro and single molecule experiments. To further strengthen our claim of MORC2 existing as a dimer, we quantified the monomer and dimer fractions of MORC2 recombinant protein using AFM by measuring the volume of individual particles (Fig 6b-c). Our analysis revealed that the volume of a monomer was approximately $130 \pm 3 \text{ nm}^3$ (mean \pm SD), while the volume of a dimer was approximately $210 \pm 1 \text{ nm}^3$ (mean \pm SD). The results showed that about 70% of MORC2 particles exist as dimers, indicating that the majority of MORC2 molecules adopt a dimeric form. Taken together, these results suggest that within the context of chromatin remodelling, it is highly likely that the MORC2 function is dimer dependent.

* ChIP-seq, ATAC-seq, FLIM-FRET, and RNAseq convincingly show that MORC2 tends to bind H3K9me3-depleted regions. The tendency of MORC2's binding to open chromatin is intriguing given that MORC2 associates with HUSH complex.

Yes, indeed we were surprised to see a large proportion of MORC2 binding to the H3K9me3-depleted regions. However, we did observe a small population of MORC2 binding to H3K9me3 (Fig 4d, ED Fig 11a-b), similar to a previous study by Tchasovnikarova et al, Nature Genetics 2017 (PMID: 28581500). This MORC2 occupation of the H3K9me3 sites may increase when retroviruses integrate into the heterochromatin region of the genome where they are actively silenced by HUSH.

* Line 288: eGFP-H2B and mCherry-H2B (FP at the N-terminus), Line 942: H2B-eGFP and H2B-mCh (capital H) (FP at the C-terminus), Line 944-945: H2B-eGFP and H2B-mCh (small H) (FP at the C-terminus) \ Make your notations consistent.

We have made the notations consistent throughout the manuscript.

* Line 130 claims that dephosphorylated MORC2 may be a stronger DNA binder. In Line 244-245, it says that MORC2(PD) hydrolyzes ATP 2-fold faster than the wild-type

MORC2. Fig 5c shows that DNA compaction time for PD is shorter than that of wild-type. Those three results from different approaches seem to agree with each other. It will be interesting and informative to see discussions on how the first two results are intimately related to the last result.

Yes, we agree that is an important observation and we have explicitly mentioned it in Discussion section, lines 949-1218.

* Line 1145-1146: “To obtain kymograph” -> Fig 5 does not contain any kymograph figure. If the authors needed kymographs for the subsequent intensity fluctuation analyses, rewording would clarify the situation. For example, “To obtain kymographs for the subsequent analysis” instead of “To obtain kymographs”.

We apologize for the oversight in omitting the kymograph in the original submission. In the revised manuscript, we have added the kymograph to illustrate DNA compaction mediated by MORC2 WT in the presence of ATP (Fig 5c). The kymograph clearly illustrates that prior to the addition of MORC2, DNA intensities were evenly distributed, whereas cluster intensities increased after MORC2 addition, indicating DNA compaction. The corresponding explanation has been updated in line 2742 to reflect this addition.

* For single-molecule assay, Sytox-Orange-labeled double-biotinylated lambda-DNA was tethered and introduced MORC-2 proteins at 20 ul/m (ul/min). Then, the authors used intensity fluctuation changes to indirectly determine when DNA compaction occurs. This approach led the authors to conclude that phosphodead MORC2 compacts DNA 3-times faster than WT MORC2 in a statistically significant manner. We are not convinced of this important conclusion. Because of the presence of flow while applying MORC2 protein, DNA is biased towards the buffer flow direction while two DNA ends are still tethered. This is obvious in the provided movies where the intensity of the upper part of DNA is roughly twice higher since two DNA segments overlap due to flow. We guess that the flow will make the determination of DNA compaction onset complicated as the authors also indirectly implicated this difficulty by saying that “Although the appearance of clusters was sometimes not obvious due to the resolution limit.” If DNA compaction is a slow process, the uncertainty may be still fine. But, both WT and PD MORC2 seem to be robust (fast) DNA compactor, even some uncertainty might lead to big uncertainty in the DNA compaction time. Furthermore, the compaction time of WT and PD (Fig 5c) were obtained from only 11 and 6 events, and any purified protein has slightly different activities for each purification (although using exactly the same protocols). To help us and the readers convinced, at minimum, (1) more information on the data analyses such as an SI figure of intensity fluctuation graph with the exponential decay fitting and (2) more data points (instead of 6 and 11 DNAs – ideally from multiple experiments (for sure) with proteins from more than 1 purification (not required)) are suggested.

We appreciate the reviewer's insightful concerns regarding the potential impact of flow effects and the limited number of events in our original analysis. To address these issues, we have included additional representative real-time fluctuation traces (ED Fig 12a–c). To further validate our conclusions, we performed an alternative analysis by measuring cluster intensities on the DNA (Fig 5d–j and ED Fig 12f–h). Following DNA compaction, distinct clusters became visible, enabling us to quantify cluster intensities and track real-time intensity changes within these regions. Since the experiment was conducted for 3 minutes following MORC2 injection, the same conditions (flow duration and flow speed) were applied to all control samples. Additionally, we expanded the dataset to comprise over 30 events from at least 5 independent experiments using proteins from at least two different purification batches. This complementary analysis yielded results consistent with our original fluctuation-based approach, reinforcing the robustness of our conclusions.

Additionally, we independently measured the lag time (defined as the time from flow injection to the onset of DNA compaction) and the compaction time (determined by exponential fitting of the cluster intensity increase) (Fig 5d–f). Both the lag-time analysis and the compaction-time analysis confirmed that the phosphodead (PD, S6A) MORC2 mutant compacts DNA approximately three times faster than the wild type, demonstrating that our findings are not artifacts of flow.

* Line 1204-1205: We are thankful for the authors depositing various data including CryoEM maps and atomic coordinates for readers. As a minor point, it says that “Original gels, blot images and numerical data used to generate plots are provided in the source data.” We downloaded all the files for review, but the original gels and blot images are missing. (What we see is chopped gel images such as ED Fig 1a.)

We apologise for this oversight; we have now uploaded all source data.

Reviewer #2 (Remarks to the Author):

The Microchidia (MORC) family of chromatin remodellers are associated with chromatin compaction and epigenetic silencing. The manuscript by Tan et al. describes a mechanistic study of MORC2, where they aimed to characterize the functional roles of MORC2 domains in dimerization, DNA binding and DNA compaction, and how these roles are regulated by phosphorylation. A strength of this manuscript is the multiple avenues of investigation presented, which provide complementary datasets that inform on the MORC2 mechanism of action. Overall, the work provides some interesting insights into human MORC2 function that can be evaluated against related published work (most notably Kim et al. 2019 on *c.elegans* MORC-1); however, there are significant problems with aspects of the manuscript that should be addressed before publication. In particular, the authors should take care to ensure that the data presentation is clear, that the writing conveys the intended messages and that conclusions are not overstated.

Major concerns:

Cellular 'number and brightness (NB)' analysis - whilst evidence in the manuscript (and elsewhere) support the conclusion that MORC2 forms a dimer, there are several issues with this part of the paper:

1. The paper should include relevant controls / calibration datasets (eGFP cells).

We have added the eGFP monomer cell as relevant control/calibration data in ED Fig 1d.

2. The authors state that they obtain pixel resolution but the interpretable spatial resolution of the data (not pixel size) should be defined to make this meaningful for the reader.

The pixel size for the number of brightness data is 41 nm and is added to figure legend for ED Fig 1d.

3. ED figure 1e and 1f appear to plot the fraction of pixels assigned to dimers (-NCS = ~10 %; +NCS = ~20 %) or oligomers (-NCS = ~0.2 %; +NCS = ~0.5 %) - what is being shown should be made clearer in the figure/legend (fractions of what?; and what values are indicated by the box plot?) and the figures should include individual data points.

We have plotted individual data points in ED Fig 1e and updated the figure legend showing the quantification of the fraction of pixels containing MORC2 dimer or oligomers from the NB data acquisition represented in ED Fig 1d.

4. The values in the box plots (dimers -NCS = ~10 %; +NCS = ~20 %) do not seem to match those in the text (dimers -NCS = ~20 %; +NCS = ~40 %).

Thanks for pointing this oversight. We have fixed it in lines 132-133.

5. The methods section refers to the wrong protein (HP1 α). The authors should confirm that the methods and statistics given here refer to calibrations and experiments carried out for this study. They select cells with low eGFP, how are fractions of monomer vs oligomer impacted by absolute eGFP intensity?

We have corrected the method section with the correct protein and clarified the NB analysis in the methods.

6. Finally, as the results appear to plot the fraction of pixels assigned to each state of the protein, to what extent does the selection of area impact the results? The authors should include a panel showing the areas selected for analysis for the two conditions.

We have added the panel showing the areas selected for analysis in ED Fig 1d. The cursors used to highlight monomers, dimers and oligomers is derived from a monomeric eGFP control. We now show the intensity versus brightness scatter plot with the eGFP calibrated cursors superimposed (ED Fig 1d-e). The figure legend is also updated.

MORC2 as a sliding clamp – whilst it is entirely possible that MORC2 forms a dimer that encloses and can slide on DNA, as shown for *c.elegans* MORC-1, there are several issues with the assay used to test this hypothesis in this manuscript, meaning the data do not substantiate the claim. Problems include (but are not limited to):

1. What is being quantified in the gels? The way in which the data has been analysed and interpreted are very unclear. Initially 100 nM of the MORC2 protein is mixed with 25 nM of DNA, meaning that maximally 25% of the protein is DNA bound (assuming a 1:1 interaction). When the lower concentrations of competitor DNA (25 nM) are added, they are likely to interact with the free protein (no competition required), so how these results should be interpreted (particularly in the case of the circular DNA substrate experiment) is not clear.

We have added the workflow for quantification of the percent DNA shifted in ED Fig 3a.

We acknowledge the reviewer's concern regarding the experimental limitation where excess free protein may bind to the competitor DNA. To address this, we conducted a more rigorous experiment by measuring the percentage of 90 bp IRDye8-labeled DNA bound to MORC2 and compared it to conditions where MORC2 was preincubated with 101 bp linear or circular DNA in equimolar ratio (Fig 3c-d, ED Fig 10). This experimental design is consistent with our previous work (competition with linear DNA; Alcon et al., NSMB 2020; PMID: 32066963) and studies from another group (competition with circular DNA; Wang et al., Nature 2020; PMID: 32269332).

It is important to note that the maximum length of IRDye8-labeled DNA we could obtain was 90 bp, which is only 10 bp shorter than the preincubation DNA. This minor difference is insufficient to accommodate an additional MORC2 molecule on to a DNA, ensuring accurate interpretation of the competition results.

To provide further evidence for MORC2's sliding clamp observation, we collected atomic force microscopy (AFM) data on MORC2. We observed MORC2 forms a closed, circular conformation encircling the DNA which we call O-shape (new Fig 6j). Additionally, we also observed V-shaped MORC2 conformation (the ATPase domain has not dimerise) which may be the 'before' or 'after' ATP hydrolysis state of MORC2. The frequency of O-shaped MORC2 was however low, this is likely due to the localization of O-shaped MORC2 within DNA clusters, making them more challenging to visualize clearly (Fig 6l).

As a negative control for this observation, we used the MORC2 N39A mutant (ATP-binding-deficient mutant), which cannot dimerise through its ATPase domain. This mutant displayed a more open, V-shaped structure (Fig 6f, ED Fig 13a).

Overall, these latest biochemical and biophysical observations support that MORC2 does clamp on to the DNA and is consistent with observations made for MORC1 and MORC3 although the exact role of CTD in DNA clamping for those MORCs was not investigated.

2. At the higher concentration of competitor (20-fold excess), 74% of the competitor DNA is apparently shifted by the protein (MORC2 FL, 60-bp linear DNA condition), whereas a 1:1 interaction would predict a maximum of 5%. How do the authors explain this result?

In the new experiment (Fig 3c-d) we observe that ~25% of the competitor DNA displaced linear DNA. Given that we now use longer length DNA, therefore, there is a possibility of more than one MORC2 molecule binding to a single DNA.

3. The quantification of the result with the truncated 1-603 MORC2 also seems highly improbable based on a visual inspection of the images in the figures (50% competitor DNA is quantified as shifted).

In the ED Fig 10, we observed almost 100% displacement of pre-incubated linear with the competitor DNA and about 50% of circular DNA. This suggests that MORC2 lacking C-terminal is not able to clamp onto the DNA efficiently.

4. The comparison between the linear and circular DNA is further complicated by the fact that the DNA constructs are not equivalent in length (and possibly not sequence?). ED fig. 9 suggests that MORC2 interacts with longer DNA with higher affinity, confounding interpretation of the results.

We have repeated the experiments with matching DNA length (101 bp linear or circular DNA), and 90bp IRDye800-DNA (maximum length for a fluorescently labelled dsDNA available commercially from a vendor).

5. The labelling of the lanes of the gels appear to be incorrect and inconsistent between the main figure and the equivalent ED figure.

Corrected in Fig 3c, ED Fig 9a-b and ED Fig 10a.

6. The concentrations plotted in Fig. 3e, described in the methods and described in the figure and ED figure legends are inconsistent.

Corrected in Fig 3d and ED Fig 10, and in the methods.

7. The length of circular DNA used differs in the manuscript, figure and ED figure legends and methods (95, 101 or 105 bp).

We apologise for this mistake. The circular DNA length is 101 bp and we have corrected it in methods section and figure legends.

8. In general, the authors need to consider whether this approach addresses the question being asked and if so, provide a much more comprehensive description of how the experimental setup and results can be interpreted to support their hypotheses.

We have re-designed these experiments which incorporates three changes: 1) addition of new sample: measure of IRDye800-DNA incorporation into MORC2 to make a better comparison when it is used as a competitor DNA; 2) the preincubation of DNA with protein at equimolar ratio prior to competition so that there is no excess protein left; and 3) use of similar length of preincubating and competitor DNA to rule out any effect of affinity of MORC2 towards a DNA of specific length.

Additionally, these new experiments are in line with previously published DNA competition assays for other proteins (Alcon et al., NSMB 2020; PMID: 32066963 and Wang et al., Nature 2020; PMID: 32269332). Finally, we have tested the outcomes of our biochemical results with an orthogonal method of AFM that provided visual proof of MORC2 clamping onto the DNA (Fig 6j). We hope that these new results would be satisfactory to the reviewers' comments on this important observation.

MORC2 FLIM-FRET –

1. Line 288-290 – IF staining with MORC2 or (not and) H3K9me3 antibodies. The data does not show that MORC2 is localized with regions depleted of H3K9me3 (they are imaged in different experiments), it shows that is localized with areas of lower FRET according to the current data interpretation. H3K9me3 is correlated with regions of higher FRET, so only indirect conclusions can be drawn.

We have clarified the FRET findings in lines 331-334. We are comparing the fraction of histone FRET pixels (red pixels) inside MORC2 foci versus outside of the MORC2 foci (nucleoplasm), using the MORC2 immunofluorescence (IF) signal intensity threshold. The data show that the MORC2 foci region has a lower FRET fraction compared to the nucleoplasm in the same nucleus, leading us to conclude that MORC2 is associated with open chromatin. H3K9me3 serves as a positive control for the experiment, as we have previously demonstrated that regions with high H3K9me3 intensity are associated with more compact chromatin (Liang et al., 2024, Chromosoma; PMID: 38265456).

2. Fig. 4h – It is not clear what this figure is showing. The y-axis is apparently in units of pixels and the data appear more suited to a box plot representation than the lines shown. It is not clear why there are only 4 cells recorded for the H3K9me3 data. It is difficult to interpret these results.

We have replotted Fig 4h by quantifying fraction of FRET in foci or nucleoplasm with 17 cells for H3K9me3 vs 24 MORC2 cells. The figure legend is updated to clarify

quantification of Fig 4h.

Other comments:

- Care should be taken with all figures to ensure that they are appropriately clear, labelled (incl. axis, units etc.) and consistent with the legends, text, and methods sections. The current version of the manuscript has multiple errors and inconsistencies that need to be addressed but are not listed exhaustively here.

Thanks for the comments. We have made sure that all the figures are consistent with other sections of the manuscript.

- In the discussion, many hypotheses generated by data in the paper are conveyed as statement of fact, when the data do not support this (e.g. lines 388-389: “As MORC2 is a DNA sliding clamp that preferentially binds to open DNA...”, the authors need to draw a clearer distinction between proposed mechanisms that are consistent with the data and hard conclusions that are conclusively demonstrated.

We have made several changes in the Discussion section to address the raised concerns.

- For all quantified EMSAs, the region of the gel that has been quantified should be indicated. The current quantification data is very hard to interpret.

We have added the quantification method in ED Fig 3a.

- Fig 1b would be easier to interpret if it included a panel with a MORC2 colored by domain (like Fig. 1a)

We have updated Fig 1a and Fig 1b to include domain diagram.

- Lines 168-171 are not clear.

We have clarified them in line 249.

- Several experiments use the MORC2PD mutant to evaluate the role of phosphorylation in regulating MORC2. Given the authors show that phosphorylation state of MORC2 can be manipulated with PAK1 kinase or γ PPase, it would be more direct and informative to assess the role of phosphorylation on MORC2 function after treatment of the protein with these enzymes. Phosphorylation or dephosphorylation should be demonstrated.

We conducted mass spectrometry analysis on MORC2 treated with λ -phosphatase and found that serine 743 remained resistant to dephosphorylation (new Supplementary Data Table 1). As such, λ -phosphatase treatment does not fully represent the completely dephosphorylated state of MORC2. To address the reviewer's concern, we

have now included EMSA and SPR analyses of untreated, PAK1-treated, and λ -phosphatase-treated MORC2 (ED Fig 4a-c), along with DNA compaction assays (Fig 5h-j).

Our new findings align with results from MORC2 WT (endogenously phosphorylated in insect cells) and the phosphodead (PD) mutant, where λ -phosphatase-treated MORC2 shows higher DNA-binding affinity and compaction activity compared to PAK1-treated MORC2. Given this consistency and the ease of purification of WT and PD mutants, we prefer using them to demonstrate the majority of MORC2 functions.

- Lines 72-75 aren't clear and would benefit from rewriting.

We have rewritten the sentences in lines 77-80.

- Line 321-323 – “indicating phosphorylation of PIM motif in MORC2 CTD is crucial for dictating the speed at which DNA compaction would occur” – This result shows that mutating these residues increases the speed on DNA compaction in this assay. This implies that phosphorylation (if the WT protein in this assay was phosphorylated) might negatively regulate the speed of compaction. The authors could directly test this hypothesis by testing compaction with WT MORC2 that is either phosphorylated (after PAK1 kinase treatment) or not phosphorylated (after γ PPase treatment).

We thank the reviewer for this insightful comment. To address this point, we compared the DNA-compaction kinetics of wild-type MORC2 in both dephosphorylated (after λ PPase treatment) and phosphorylated (after PAK1 kinase treatment) states. Our results demonstrated that dephosphorylated MORC2 exhibited significantly higher DNA-compaction activity, whereas phosphorylated MORC2 showed no compaction under the same conditions. These findings support the hypothesis that phosphorylation negatively regulates both the speed and efficiency of DNA compaction. We have included these new results in the revised manuscript (Fig 5h-j).

- Line 325-328 – Similarly to the above, the authors test the role of ATP hydrolysis in DNA compaction using a mutant (the methods seem to indicate that the assays were performed without ATP, but this is presumably just an omission in the writing). Whilst this is a valid approach, supporting data +/- ATP or ATP analogues would also help to confirm this finding, particularly given that MORC-1 has been shown to compact DNA in a manner that is nucleotide binding but not hydrolysis dependent (Kim et al. 2019).

We thank the reviewer for this valuable comment. To strengthen our findings, we included an additional ATPase mutant, N39A, which completely blocks ATP binding to MORC2 (Fig 5e–g). Our results showed that this mutant does not compact DNA at all, whereas the hydrolysis-deficient S87A mutant compacts DNA, albeit at a slower rate. This strongly supports the conclusion that DNA compaction is ATP-dependent. In addition, we observed clear correlations between ATPase activity and DNA-compaction

kinetics, strongly supporting ATPase-activity dependence of MORC2-mediated DNA compaction (Fig 5g).

Furthermore, we tested the DNA compaction in presence and absence of ATP as suggested by the reviewer. While we did not observe significant differences in DNA compaction kinetics by MORC2 WT +/- ATP during the initial stages, we did observe notable differences in the DNA-compaction time (Response to Reviewers Fig 1a). By washing away background buffer using microfluidics, we found that DNA clusters formed by MORC2 in presence of ATP remained stable, whereas clusters formed in the absence of ATP were unstable and dissociated after washing (Response to Reviewers Fig 1b-c). This clearly indicates that stable DNA compaction is dependent on ATP.

We hypothesize that the lack of a significant difference in compaction kinetics between wild-type MORC2 with and without ATP could be attributed to residual ATP bound to the ATP-binding pocket of MORC2 after purification, as MORC2 exhibits low ATPase activity even without supplementing ATP. This residual ATP may influence compaction kinetics despite the absence of externally added ATP.

In the main manuscript, we emphasized the role of ATP in DNA compaction by presenting data from two ATPase mutants, N39A and S87A, which together demonstrate that MORC2-induced DNA compaction is ATP-dependent. We have revised the relevant sections accordingly to reflect these findings.

Response to Reviewers Figure 1. Stability of compacted DNA in presence or absence of ATP post buffer wash. **a.** DNA compaction time of MORC2 WT with and without ATP (mean \pm SD, $n = 34, 23$ DNA clusters, respectively.) The p -value was obtained by two-tailed unpaired Student's t-test. **b.** Snapshots of DNA in buffer flow with and without ATP. **c.** Plot showing the fraction of compacted DNA after 8-minute of buffer wash.

- Figure 5f – Three AFM images are proposed to show MORC2 interactions with DNA

(lines 335-337), however these images are only anecdotal evidence of the possible shape of MORC2 and do not address the stated aim of focusing on ‘whether DNA binding changes the oligomerization state of MORC2’. Supporting data showing analysis of MORC2 alone, in the presence and absence of AMP-PNP (or ATP), to compare structural rearrangements upon ATPase dimerization would be informative. Scale bars and comparison to the scaled cryo-EM structures of MORC2 would help demonstrate that the particles being shown are MORC2.

We thank the reviewer for these insightful comments and suggestions. In response, we have conducted additional AFM experiments to further investigate the structural rearrangements of MORC2 upon DNA binding.

First, in the absence of ATP, the majority of MORC2 molecules exist as dimer via their CTD, adopting what we describe as a V-shaped conformation (Fig 6a-c). Upon addition of AMP-PNP or ATP, a significant proportion of MORC2 molecules transitions to an O-shaped conformation, characterized by dimerised ATPase domains (“heads”) (Fig 6d-f). The DNA-bound MORC2 in the presence of ATP or AMP-PNP, showed ~ 36 % O-shapes. The relatively low proportion of O-shaped conformations in DNA-bound conditions is due to the formation of MORC2/DNA clusters upon ATP binding, which likely creates a crowded environment in the AFM images, limiting the visibility of individual MORC2 conformations and underestimating their apparent frequency.

Furthermore, analyses of the ATP-binding-deficient N39A mutant of MORC2 revealed that it fails to adopt an O-shaped conformation and does not bind DNA (Fig 6h-i, ED Fig13a), indicating that ATP binding is essential for MORC2-DNA interaction.

We have added scale bars to all AFM images presented in this paper. However, since our cryo-EM analysis resolves only the ATPase domain of MORC2, it does not provide a direct comparison to the full-length MORC2 molecules observed in the AFM images. Therefore, we have chosen not to include scaled cryo-EM structures alongside the AFM images.

- ED Figure 6f is very unclear and does not currently add anything to the interpretation of the data. What is the relevance of the ‘representative intra-protomer crosslinks’? How many (and which) crosslinks identified experimentally did not fit the AlphaFold prediction? Why was the AlphaFold structure (no DNA) validated with the crosslinking data with DNA? Perhaps a comparison of each of the crosslinking datasets with the model would be more informative, highlighting how many experimentally derived crosslinks fit or do not fit the model (within a given distance tolerance), and where these are found in the structure.

The sentence that mentions ‘representative intra-protomer crosslinks’ should have read “The solid lines indicate intra-protomer crosslinks and dashed lines indicate inter-protomer crosslinks. We did not use AlphaFold3.0, which supports protein-DNA complex prediction due to potential commercial interest we have in future for drug campaign

against MORC2. Therefore, we used the AlphaFold2 structure with 'no DNA' to compare the number of crosslinks that satisfy the distance length of sulfo-SDA between the samples with and without DNA. We have updated the ED Figure 6f to show the quantitatively enriched crosslinks from 'DNA' and 'non-DNA' samples along with the graph showing the quantification of crosslinks that are within the median distance expected for sulfo-SDA crosslinker.

Reviewer #3 (Remarks to the Author):

Reviewer #4 (Remarks to the Author):

In this manuscript Tan et al provide an in-depth biochemical and structural characterization of the chromatin remodeler MORC2, as well as functional analysis in the context of genomic binding and gene expression regulation. They show that MORC2 is a functional dimer and its dimerization depends on the flexible C-terminal domain (CTD). ATP binding increases the proximity of the two protomers. MORC2 is a DNA sliding clamp with DNA binding sites located in the ATPase domain (CC1, CW-CC2) and CTD (CC4); using X-link mass spectrometry and HDMX the authors show reorganization of ATPase and CTD regions upon DNA binding. Phosphorylation of the CTD reduces ATPase activity without affecting DNA binding. The authors further show that MORC2 can compact DNA in vitro in ATPase-dependent manner. The relevance of this effect is unclear though as ChIP experiments showed that MORC2 generally binds to promoter regions and open chromatin with only a small portion found at H3K9me3-enriched heterochromatic loci. Moreover, MORC2 knock-outs cells don't show major changes in transcription and chromatin accessibility, suggesting that MORC2 exerts local rather than global effects on chromatin.

Overall, a lot of data is provided in this manuscript but insights into the functional connection of these findings are missing. Specifically:

1) Are the genes that are upregulated in RNAseq and more accessible in ATACseq for MORC2 KO cells also bound by MORC2 based on ChIP and contain H3K9me3 marks in WT cells? If so, what is the function of these genes and why would only this small subset require MORC2-mediated compaction? Did you analyse the differential expression of repetitive elements in your RNA-seq data?

There are six genes that are upregulated in RNA-seq and more accessible in ATAC-seq in MORC2 KO cells also overlap with MORC2 binding sites and H3K9me3 marks in WT cells. These genes are ZNF22-AS1, ZNF22, LINC00858, MIR3171HG, LINC02994 and

RRAGA. Additionally, we have plotted the differentially expressed genes overlapping with MORC2 binding sites in ED Fig. 11e.

Notably, these genes are primarily zinc finger genes, which may play roles in chromatin compaction and transcriptional regulation. One possible explanation for why only this small subset requires MORC2-mediated compaction is that these genes reside in regions enriched for H3K9me3, where MORC2 may reinforce silencing by maintaining a compact chromatin state. Unlike other repressed genes that may rely on redundant silencing mechanisms, these specific loci might be more dependent on MORC2 due to their chromatin context or regulatory complexity.

Regarding repetitive elements, our RNA-seq samples were generated using poly(A) selection, capturing only mRNA, which prevents the analysis of non-gene components.

2) In PMID: 23260667 they showed that MORC2 promotes chromatin relaxation in ATPase-dependent manner after radiation-induced DNA damage based on the micrococcal nuclease assay. In PMID: 31616951 they show that MORC2 induces chromatin relaxation even without DNA damage and that this is dependent on its PARylation. In this manuscript you show that MORC2 promotes chromatin compaction in vitro. Cellular effects are missing. I think it's crucial to explain how the same chromatin remodeler can have two opposite effects on the chromatin state through its ATPase activity. This must be tightly regulated – how? Would overexpression of MORC2 increase chromatin compaction and make it more global? What is the phenotype of the phospho-dead mutant in cells, which shows higher compaction in vitro? I think that showing the cellular relevance of the in vitro compaction effect is crucial.

We appreciate the reviewer's insightful comments on the dual role of MORC2 in chromatin regulation and the need to establish the cellular relevance of our in vitro compaction findings.

To test whether MORC2 overexpression leads to global chromatin compaction, we transfected HEK293 cells with a MORC2 WT plasmid and performed ATAC-seq. Differential analysis revealed only six downregulated genes compared to the lipofectamine control, suggesting that MORC2 overexpression alone does not result in widespread chromatin compaction (ED Fig. 11f).

To further explore the regulation of MORC2-mediated compaction, we examined the phospho-dead (S6A) and phospho-mimic (S6D) mutants in MORC2-KO HEK293 cells (ED Fig. 11g). ATAC-seq differential analysis showed that the S6A mutant, which exhibits higher compaction in vitro, led to 32 downregulated and 11 upregulated regions, indicating increased chromatin compaction relative to MORC2 WT. This aligns with our single-molecule assays. In contrast, the S6D mutant resulted in only 9 downregulated and 1 upregulated region, suggesting a slight increase in compaction compared to WT.

The apparent contradiction between MORC2's roles in chromatin relaxation (as shown in PMID: 23260667 and PMID: 31616951) and chromatin compaction (our study) likely

reflects a highly regulated, context-dependent function of its ATPase activity as mentioned by the reviewer. One key mechanism of this regulation appears to be post-translational modifications (PTMs), such as phosphorylation, which may act as a molecular switch modulating MORC2's ability to remodel chromatin depending on the extent of phosphorylation. In our study, the phosphodead S6A mutant showed increased chromatin compaction, while the phosphomimic S6D mutant had a milder effect, suggesting that extensive phosphorylation may prevent excessive compaction and promote chromatin relaxation when required. Indeed, even subtle changes in phosphorylation status have profound effects on MORC2 function, as highlighted in Response to Reviewers Fig. 2, reinforcing the idea that MORC2 is tightly regulated by PTMs.

Beyond phosphorylation, other modifications such as PARylation (PMID: 31616951), SUMOylation (PMID: 36793866), and acetylation (PMID: 32112098) have been implicated in enhancing MORC2's DNA damage response activity. It is possible that there is crosstalk between these modifications and phosphorylation, allowing MORC2 to switch between its roles in chromatin relaxation and compaction. Additionally, interactions with chromatin remodellers (such as the HUSH complex) and DNA repair factors may further fine-tune this switch. For example, in response to DNA damage, MORC2 may transition to a relaxed chromatin state to facilitate repair, whereas under normal conditions, it may contribute to heterochromatin maintenance through localized compaction.

Taken together, our findings suggest that MORC2's ability to mediate chromatin compaction or relaxation is tightly regulated through phosphorylation and other PTMs, allowing it to function dynamically in response to different cellular contexts. We have hinted at the importance of PTMs in MORC2 function in lines 80-83, 114-116, 816 and 1241-1243.

3) 'The neuronal MORC2 mutation, R252W results in defective ATPase activity and hyperactive transcriptional silencing^{11,21}, suggesting that ATP hydrolysis drives chromatin remodelling during HUSH-mediated silencing.'

ATPase activity was reported to be reduced (not defective) in Douse et al. 2018. Higher efficiency of HUSH-dependent epigenetic silencing when ATP hydrolysis is reduced would suggest that ATPase activity relaxes chromatin (rather than compacting it).

Thanks for the comment. We have corrected in lines 75-77.

4) The EMSA experiment in ED Fig 2 is not conclusive. The example EMSA in panel b for the phosphatase-treated samples shows overall lower amount of DNA and the signal for the highest concentration of MORC2 (200 nM) is very weak. In panel c quantification this sample is completely missing? EMSA should be repeated and complete quantification provided. EMSA in ED Fig 5f is much cleaner. A more quantitative assay could also be performed (e.g., SPR as in Fig 3). Please also indicate the length of DNA (I assume 60 bp).

Thanks for the suggestions. We have repeated EMSA and performed SPR for PAK1, phosphatase treated and WT MORC2 in ED Fig 4a-c.

The conclusions from the current results that dephosphorylated MORC2 is a stronger DNA binder are at variance with (i) the authors' results from Fig 3b whereby SPR showed similar binding affinity for WT and phospho-dead mutant, and (ii) the published data (PMID: 23260667) showing that MORC2 phosphorylation after DNA damage at S379 promotes chromatin binding.

The reviewer has raised an important point. To address this, we have repeated the SPR experiments from Fig. 3b multiple times and now consistently observe that the PD mutant exhibits a five-fold increase in DNA-binding affinity compared to WT. To further substantiate this finding, we conducted the following experiments:

1. EMSA analysis of WT vs. PD mutant, which confirms that the PD mutant has a higher DNA-binding affinity (Extended Data Fig. 4d-e).
2. EMSA and SPR analyses of untreated, PAK1-treated, and λ -phosphatase-treated MORC2, demonstrating that dephosphorylated MORC2 (λ -phosphatase-treated) binds DNA more strongly (Extended Data Fig. 4a-c).
3. Single-molecule DNA compaction assays on untreated, PAK1-treated, and λ -phosphatase-treated MORC2, showing that λ -phosphatase-treated MORC2 exhibits higher compaction activity compared to PAK1-treated MORC2 (Fig. 5h-j).

Additionally, we believe the reviewer may have meant S739 rather than S379. To test whether S739 phosphorylation modulates chromatin binding, we purified MORC2 S739D (phosphomimetic) and S739A (phospho-dead) mutants. We found that the S739D mutant exhibits a two-fold higher ATPase activity than S739A (S739D: $k_{\text{cat}} = 0.083 \pm 0.02 \mu\text{M ADP}/\text{min}/\mu\text{M}$; S739A: $k_{\text{cat}} = 0.028 \pm 0.02 \mu\text{M ADP}/\text{min}/\mu\text{M}$) (Response to Reviewers Fig. 2a-b). This higher ATPase activity correlates with increased DNA binding (S739D: $K_{\text{D}} = 37.7 \pm 34 \text{ nM}$; S739A: $K_{\text{D}} = 170 \pm 21.8 \text{ nM}$) (Response to Reviewers Fig. 2c-d), suggesting that S739 phosphorylation influences DNA binding via its effect on ATP hydrolysis.

However, the MORC2 PD mutant (with six serine residues, including S739, mutated) exhibits even higher ATPase activity ($k_{\text{cat}} = 0.11$ vs. $0.08 \mu\text{M ADP}/\text{min}/\mu\text{M}$) and stronger DNA binding ($K_{\text{D}} = 5.35 \pm 3.4 \text{ nM}$, Fig. 3b) than S739D. This suggests a tightly regulated mechanism in which full dephosphorylation of MORC2 enhances DNA binding and correlates with increased ATPase activity.

We believe that future work systematically generating phosphomimetic and phosphodead mutants will be crucial to pinpointing the specific phosphorylation states that induce the transition to stronger DNA binding and higher ATPase activity.

Response to Reviewers Figure 2. Phosphorylation mutations on MORC2 S739 influences its ATPase and DNA binding affinity. a-b. MORC2 S739A and S739D mutant ATPase activity. **c.** EMSA gel showing 0, 25, 50, 100, and 200 nM MORC2 protein binding to 25 nM IRDye700-labelled 60bp dsDNA. **d-e.** SPR analysis of MORC2 S739A and S739D mutant.

5) Structure results should always start with data for the WT and full-length constructs to rationalize the use of mutants. Why was the structure of the full-length ATPase-dead mutant not included in ure 1? If phosphorylation of the CTD affects dimer flexibility, why didn't the authors dephosphorylate the protein before preparing the grids? I cannot comment on the quality of cryo-EM datasets and I hope that another reviewer will cover these aspects.

We have moved structure of full-length ATPase dead mutant S87A to the Fig 1b. Since phosphatase-treated MORC2 exhibits similar behaviour to the PD mutant - where all phosphorylation sites are mutated to alanine (ED Fig 4a-b; Fig 5h-j) - and considering that phosphatase treatment was unable to completely remove all phosphorylation sites (as shown in the new Supplementary Data Table 1), we have opted to use the PD mutant for our cryo-EM studies.

6) EMSA experiment in ED Fig. 8a for 496-1032 is unclear as it seems that DNA got stuck in the wells (there is no unbound DNA. Thus the corresponding curve in Fig 2d is unreliable. In Fig. 2d the curve for 496-603 isn't visible. Please clarify.

We have repeated the experiments and updated the ED Fig 9 and Fig 2d.

7) EMSA from ED Fig. 9 should be shown before the first EMSA in ED Fig 2 to justify the use of 60 bp DNA for further experiments.

We have moved the figure to ED Fig 3.

Minor (mainly text editing for clarity)

1) line 72/72: Remove: 'There are several'

Amended in line 77

2) line 74/75: This statement is unclear and should be rephrased: 'resulting in histone H2AX phosphorylation to cause chromatin remodeling'

A suggestion how to rephrase: ...and promotes chromatin relaxation and gH2AX phosphorylation.

Changed in lines 78-80

3) line 90: domains

Changed in line 95

4) line 90: possesses

Changed in line 95

Response to reviewers' comments

We appreciate all the comments from the reviewers. Here is our point-by-point response in blue.

Reviewer #1 (Remarks to the Author):

In the revised manuscript, the authors satisfactorily addressed this reviewer's comments and concerns. They went extra miles and provided the convincing revised manuscript.

We thank the reviewer for their kind words and appreciating our efforts, and also for taking the time to review our manuscript.

Reviewer #2 (Remarks to the Author):

In their revised manuscript and accompanying rebuttal, Tan et al. have addressed many of the points raised in my initial review and improved their manuscript. I am grateful for their efforts in this regard. Overall, the manuscript includes a lot of interesting data and comes together to produce an informative study that advances our understanding of MORC2, its interaction with DNA and its regulation. The results are mostly consistent with another recently published study (Fendler et al. 2024 PMID : 39739841; with the exception of the impact of DNA binding on ATPase activity), supporting many of the conclusions. However, several important issues remain or arise from the new data that has been included. In particular, the new EMSA experiments presented in Figure 3 and ED Figure 10 have several problems.

Major:

Figure 3c/d and ED Fig 10 have many serious problems with the experimental design, data and analysis. They cannot be used to draw any meaningful interpretations. Problems include (but are not limited to):

In the rebuttal, the authors state "In the new experiment (Fig 3c-d) we observe that ~25% of the competitor DNA displaced linear DNA." and in the main text "the IRDye800-labeled DNA effectively displaced the linear DNA but not the circular DNA". However, this is not what the experiment shows. The experiment does not measure displaced DNA, it measures binding of a secondary DNA species. This manuscript describes how MORC2 has multiple DNA binding sites. Moreover, according to the quantification provided, both Fig. 3c/d and ED Fig. 10 suggest that >40% of 'competitor DNA', which is added in 10-fold molar excess over total protein (assuming the 80 nM protein concentration is for the dimer?) is shifted, suggesting that each MORC2 complex can interact with multiple DNA molecules. Another recent study

convincingly shows MORC2 binding/entrapping multiple DNAs (Fendler et al. 2024 PMID : 39739841). Knowing that MORC2 can bind to multiple DNAs, and only measuring the signal of the competitor DNA, how do the authors interpret their data? Could any conclusions about competition or MORC2 acting as a sliding clamp be made on the basis of this data? The authors should consider these questions before addressing the technical problems with the data included in the manuscript.

In the rebuttal, the authors state "Given that we now use longer length DNA, therefore, there is a possibility of more than one MORC2 molecule binding to a single DNA". This may be possible, but in this particular experiment, I think the opposite is more likely to be true, that each MORC2 is binding to multiple DNAs.

We acknowledge the reviewer's point that our initial wording suggested measurement of displaced DNA, whereas the assay actually measures the binding of a secondary DNA species. We have revised our text accordingly to accurately reflect this distinction. We agree that MORC2 can interact with multiple DNA molecules due to its multiple DNA binding sites, as also supported by Fendler et al. (2024). Consequently, we have removed our initial claim regarding MORC2 sliding as a cause for competitor DNA uptake. Instead, we now emphasize that the observed binding differences are likely influenced by the topological nature of the DNA substrates.

Our calculations indeed consider MORC2 as a dimer.

Given the experimental constraints, measuring competitor DNA signal remains the most robust quantification approach. As we could not source a fluorescently labeled 101 bp long linear DNA of a different color for preincubation, and due to a small length difference of 11 bp between unlabelled and IRDye-labeled DNA, distinguishing these bands using SYBR Gold staining is challenging. However, we have quantified the amount of circular DNA in complex with MORC2 using SYBR Gold, showing that it remains consistent across different competitor DNA concentrations (**Response to Reviewers Fig. 1**). Similarly, with new quantification of competitor DNA where we measured the uptake of competitor DNA by quantifying it against the free DNA, we observe a marked difference in uptake where MORC2 is preincubated in linear vs circular DNA (Fig 3c and ED Fig 10). We have confirmed that the DNA-only controls follow a linear trend across all the DNA concentrations, which ensure there is no under or over estimations (**Response to Reviewers Fig. 2**). We mention in the manuscript that these differences in competitor DNA uptake are likely due to the topological nature of these two substrate types. This supports our model of MORC2 clamping onto DNA, further corroborated by our AFM imaging providing direct visual proof of MORC2 clamping to DNA (Fig 6j).

We believe that by using the longer DNA both scenarios are likely occurring (1) multiple MORC2 molecules binding to a single DNA, and (2) a single MORC2 binding multiple DNA molecules. The latter is true due to multiple DNA binding sites on the MORC2 as

highlighted by the reviewer whereas the former is consistent with our findings that the shortest DNA length MORC2 binds is about 29 bp (ED Fig. 3b).

We have cited Fendler et al paper in lines 668-669 and propose future investigation of DNA trapping mechanisms using single molecule in the discussion (lines 678-679).

Response to Reviewers Figure 1. Quantification of circular DNA bound to MORC2^{WT} and MORC2¹⁻⁶⁰³. **a.** Schematic of competition EMSA (top) and SYBR gold stained native PAGE gels (bottom) showing 100 nM MORC2 WT (left) and 1-603 (right) proteins pre-incubated with 200 nM circular DNA, followed by incubation with 50, 100, 200, 400 and 800 nM competition IRDye800 DNA (Note: these are the same representative SYBR gold stained gels as shown in Fig3c column 4 and Fig 10b column 4). Blue box indicates region of circular DNA selected for ImageJ quantification. Arrow indicates circular DNA used as input for quantification of circular DNA bound. **b.** The percentage of IRDye800-DNA bound was calculated as shown in EDFig 10a. WT is MORC2^{WT} same as in Fig 3a and 1-603 is MORC2¹⁻⁶⁰³ as in EDFig 10b.

Response to Reviewers Figure 2. The DNA-only controls are within the dynamic range and follow a linear trend. a-b. ImageJ quantification of 50, 100, 200, 400 and 800 nM IRDye800-dsDNA inputs for MORC2^{WT} (a) and MORC2¹⁻⁶⁰³ (b) proteins from three independent experiments. A standard curve of 50, 100, 200, 400 and 800 nM IRDye800 dsDNA was generated by performing simple linear regression on Prism. The points are represented as mean \pm standard deviation (n=3).

Most seriously, the data included in Fig 3c (in particular), is unusable. As is evident from the figure panels and the source data, the gels have overrun and very little of the free DNA is present in the gel / images. It is not clear to me what the authors are suggesting they have quantified. comparison between column 1 (or panels in ED Fig 10) and columns 2 and 3 of Fig. 3c, shows that the free IRDye-DNA that should be present at the bottom of the gel is largely absent.

We acknowledge that the original gels were overrun and have repeated the experiments to ensure robust data quality. The newly acquired gels avoid run-off issues, and quantification has been conducted more rigorously.

The data are apparently representative of experiments run twice, but it is unclear what is plotted in Fig 3d. Are these averages of the quantification or just quantification from the data shown in the figure. There appear to be some partial error bars, but it is not clear what these represent?

We clarify that the data in Fig. 3d represents mean values from three independent experiments, with error bars showing standard deviation. Some error bars were previously not visible due to their small size. To enhance transparency, we have now uploaded raw EMSA gels and quantified bands from independent experiments as Source Data.

Although less severe (the competitor DNA is still visible), the data for ED Fig 10 also has issues with the gels and the way in which they have been cropped, which has removed signal. The quantification also does not appear reasonable. Visually, it seems inconceivable that ~50% of the IRDye-DNA is 'shifted' in the 800 nM conditions, when there is such a strong band for the free DNA compared to a relatively weak band for the MORC2-DNA. This indicates a problem with the quantification, but it remains unclear precisely what is being quantified (ED Fig. 3a begins to address this, but doesn't show exactly what is quantified – is it loss of substrate, or formation of product?

Due to variability in regions demarcating the shifted bands of DNA in complex with MORC2, we have changed our quantification method. We are quantifying the formation of product by measuring the free competitor DNA left in the samples containing MORC2 against the DNA-only control. This is because we are unambiguously able to box out the region around the unbound DNA in our lanes containing MORC2-DNA samples. We

believe this is more accurate way of describing the amount of MORC2-bound competitor DNA.

For all EMSAs reported in the manuscript, the authors need to ensure that the gels are not inappropriately cropped, and that pixel values are not saturated in the images that are being quantified, particularly in the highest concentrations of DNA. If pixels are saturated, the full extent of the signal cannot be quantified. This appears to be what is shown in ED Fig. 3a, where certain peaks indicate saturation.

We have ensured that gels are not inappropriately cropped and that all EMSA images remain within a linear dynamic range. To confirm that no pixel saturation occurs, we utilized the Chemidoc feature “highlight saturated pixels”, which displays areas with saturated signal intensity (higher than a measurable range) in red, and adjusted exposure settings accordingly. For example (**Response to Reviewers Fig. 2**). For newly acquired data (Fig 3c and ED Fig 10), we manually set the Chemidoc exposure time to 0.5 s to avoid overexposure. Additionally, light intensity settings were adjusted for previous images to prevent oversaturation. All EMSA gels included in this manuscript are without the saturated pixels. These updated methodological details are now reflected in the manuscript.

The middle column of ED Fig. 10 shows multiple bands for MORC2(ATPase) + linear DNA. How do the authors explain these? They are currently labelled as ‘Free DNA’ in the figure, but it appears that this sample contains DNA+MORC2. The experiments in both figures should also include a condition for the free linear or circular DNAs before mixing with the MORC2 to demonstrate the position of the free-DNA band.

The multiple bands observed in the middle column of ED Fig. 10 likely arise from MORC2’s multiple DNA binding surfaces, DNA modifications, or the presence of multiple protein molecules binding a single DNA.

We have included several controls including protein only, unlabelled linear and circular DNA, preincubated protein and DNA, protein and competitor DNA in Fig 3c and ED Fig 10b. We also have included an accompanying gel containing competitor DNA-only control at the concentrations used in the competition assays. These control gels are run and imaged side by side with the competition gels.

There are other issues with the design these experiments, the data presented and the interpretation, but overall, these assays appear unusable.

We understand the reviewer’s concerns with the experiments shown in Fig 3c/d and ED Fig. 10. We have repeated the experiments three times in Fig 3c and ED Fig 10a ensuring there are no DNA run offs and none of the bands are oversaturated. We also quantified them in a more robust manner.

Minor:

Extended data figure 1d/e:

In their rebuttal the authors state that they have clarified the description of the NB analysis in the methods, but I cannot see where this has been done.

We apologise that this was not clear previously. We have now further clarified the description of the NB analysis in the methods (lines: 1489–1501) and ED Fig1d legend (lines: 920–921).

In their rebuttal, the authors did not address my question regarding how the selection of the area impacts the results of the monomer/dimer/oligomer NB analysis. The authors include the same panels showing examples of areas selected as were included in the first manuscript, but it is not clear why these areas were chosen. How would the results compare if the whole nuclei were analysed, rather than selected regions?

We thank the reviewer for this question. As is now explained in the methods section we measure the apparent brightness (B) distribution of free eGFP ($B=1.15$) to determine the apparent brightness of monomeric MORC2-eGFP and molecular brightness (ϵ) of eGFP ($\epsilon=0.15$). Then given that $B=\epsilon+1$ we extrapolate the apparent brightness of MORC2-eGFP dimers ($B=2 \times 0.15 + 1 = 1.30$) and oligomers (e.g., tetramer $B=4 \times 0.15 + 1 = 1.6$) to enable definition of cursors positioned at these B positions that have widths determined by the eGFP calibration experiment, as has been demonstrated to work on control constructs of eGFP that are dimeric (2GFP) and pentameric (5GFP) in previous publications that we now cite (Lou et al. 2020. Nature Comm, Lou et al. 2024. NAR). In terms of why we analyse nuclear regions of interest (ROIs) across the cell nucleus instead of just analysing the entire cell nucleus, this is to avoid edge effects from the nuclear envelope that can introduce slow time scale artefacts into the fluctuation analysis upon cell drift. In addition, imaging the whole nucleus will reduce the resolution (micrometre instead of nanometre) and prevent us from analysing the eGFP at single molecule level. We have reflected these changes in the methods (lines: 1489–1501).

Line 1621 missing text: "data acquisition was recorded is. Column"

We have corrected this.

ED Fig. 4a,b:

The '% DNA bound' numbers in (a) do not appear to correspond with the equivalent numbers plotted in (b) – see highest two concentrations for 'untreated' and 'PAK1' conditions.

We apologize for the confusion due to the same colour of error bars for both PAK1 and pre-treated MORC2. We have changed the colours of PAK1 (red) and pre-treated previously called untreated (black). The variation of error bars between three experiments are shown, the data is represented as mean \pm standard deviation (n=2, from 2 gels). Data points where we don't see error bars are because the error bar is too small to see on the plot. The ED4a is also updated by calculating shifted bands on EMSA gels with pixel intensities in the linear dynamic range. We have uploaded the 2 raw EMSA gels and quantification from 2 independent experiments in Source data.

ED Fig. 4a (untreated), 4d (WT), 5e (WT) and 9a (1-1032) appear to be repeats of the same (or similar) experiments, but with different labelling and different results. Are 4a (untreated), 4d (WT), 5e (WT) replicates, or have the proteins been treated differently, for example with phosphatase treatment? One figure legend says the data is a representative result from 3 independent experiments, one says 2 independent experiments and the other seems to be a single experiment. If these differ, it needs to be clear how they differ. If they are repeats of the same experiment under the same conditions, then the variability in the results needs to be explained. ED Fig. 9a appears to be a very similar assay but with different protein concentrations used in the titration (based on the quantification shown in Fig. 2d).

ED Fig4d (WT), 5e and 9a (1-1032) are different experiments with different batches of MORC2 WT proteins, while ED 4a (untreated) is WT treated with same phosphorylation buffer as PAK1 and LPP treated samples. We have changed the treated MORC2 to "pre-treated" in ED Fig 4a. All WT proteins were matching controls of different experiments under the same conditions, except with different protein concentrations and conducted separately to compare WT and other constructs (or to test with different DNA lengths), therefore, some variability is expected. We have changed all labels of MORC2 WT protein to WT. All protein concentrations are now stated in the figure legends and clarified in the methods. The protein concentration for ED Fig9a is now noted in the figure legend (used for quantification of Fig 2d). As the MORC2 WT EMSAs are different experiments and have been repeated different times according to the experiments aims (i.e. comparing different MORC2 truncations, or comparing different DNA lengths), the number of experiments is stated separately in all figure legends.

In general, the manuscript would greatly benefit from consistency of notation between the different protein constructs used.

We have clarified the use of WT in Fig 2, ED Fig 9 and "pre-treated" MORC2 protein in ED Fig 4. The methods section of "In vitro kinase and phosphatase assay" (line 1223) have also been updated to clarify the use of "pre-treated" MORC2 protein. We have tried to maintain consistency of notation between different protein constructs throughout the manuscript.

Similar to the above, have MORC2 mutants (e.g. S87A in ED Fig. 5f) been treated with phosphatase before being used for EMSA and SPR experiments? This wasn't obvious from the text, figure legend or methods.

None of the proteins were pre-treated with phosphatase except the ones used in ED Fig 4a-c, which has been clarified in the methods and the figure legend.

ED Fig. 4c / Fig 3b:

Has the 'WT' sample in 3b been pre-treated with λ PPase? Or is it the same as the 'untreated' sample in ED Fig. 4c? This should be clear in the description in the text and in the figure legends. If the 'WT' sample used in 3b has not been treated with λ PPase, why is there such a difference in the measured affinity with the 'untreated' sample in ED Fig. 4c? It should be made clear in the methods and throughout the manuscript whether the constructs being used have been pre-treated in anyway before being used in the assays, to distinguish them from the 'untreated' purified sample.

Only MORC2 in ED Fig 4a-c are pre-treated with phosphatase and kinase buffer, and we have now clarified that change in the ED Fig 4, and methods section of "In vitro kinase and phosphatase assay". The difference in measured affinity of pre-treated sample vs WT in Fig 3b arise potentially arise due to the additives in the kinase and phosphatase reaction buffer such as 10 mM MgAc, 1 mM MnCl₂, 100 mM KCl or 0.05% Tween-20), which are not present in experiments done in Fig 3b. It has been made clear throughout the manuscript where the proteins are pre-treated.

Description of SPR conditions in methods: "a 8-point concentration series (2-fold serial dilution, 7-500 nM)" and figure legend "protein concentrations of 0, 4, 8, 16, 33, 63, 125, 250 and 500 nM" are inconsistent. This is a small detail, but small errors like this (and other more significant errors like those listed above) were found throughout in both first and second drafts of the manuscript, making it difficult to assess the methods/data and reducing confidence in the quality of the work.

We did carefully go through the manuscript in the first revision, we apologise that we failed to identify this small error. We have corrected the typo (should read 4-500 nM) in the methods. We have again gone through the manuscript carefully to make sure such errors have been eliminated to the best of our ability.

Fig 4 – FLIM-FRET:

The manuscript asserts that it uses these experiments "to directly visualize MORC2 binding to chromatin", but these experiments do not directly visualize MORC2 binding to chromatin, or its colocalization with (or not) H3K9me3. The experiments image MORC2 or H3K9me3 and histones. They then draw a correlation between the co-localisation of MORC2 or

H3K9me3 with relative proportion of chromatin compaction measured by FRET. The authors state that “MORC2 is localised with areas of lower histone FRET” and “H3K9me3 is present at areas with high FRET”, but again, this doesn’t seem to be what the data shows. The figure plots the fraction of pixels exhibiting FRET (as an indication of compacted chromatin) in areas that colocalise with MORC2 or H3K9me3 (foci) or do not colocalise (nucleoplasm). Whilst there appears to be a large difference in the proportion of compacted chromatin in the nuclei of each cell imaged (ranging from about 5%-36%; possibly reflecting the stage of the cell cycle?), there appears to be little difference between the average fraction of compacted chromatin inside vs outside of the foci and between MORC2 foci vs the H3K9me3 foci, particularly when taking into account the different scaling of the y-axes. The two plots should be shown on the same scale and reporting the normalised fraction of FRET as displayed in Liang et al. 2024 PMID: 38265456 Fig. 2h would be informative).

We thank the Reviewer for this comment. It is true that the experiment reported in Fig. 4e-h does not directly visualise MORC2 binding to chromatin but rather visualises the compaction of the chromatin environment that MORC2 associates with, and in comparison, to H3K9me3. We also agree that the language surrounding foci versus nucleoplasm was confusing and have modified this text accordingly in the manuscript (lines 495–502). In short, we found that MORC2 foci (i.e., regions of high MORC2 intensity) colocalise with more open regions of chromatin (i.e., low FRET) when compared to the surrounding nucleoplasm (i.e., regions outside of high MORC2 intensity), unlike H3K9me3 foci which colocalise with more compact regions of chromatin (i.e., high FRET) when compared to the surrounding nucleoplasm.

We have changed Fig 4h and updated the panel in h to have the same y axes like in Liang et al. 2024. Chromosoma. We show there is a difference in the proportion of MORC2 vs H3K9me3 foci (Fig 4h).

Similarly, the text in the results section (lines 417-420) is confusing and I am not sure that it accurately conveys what the figure shows. The text suggests that MORC2 is found in the ‘nucleoplasm’, whilst H3K9me3 is found in ‘foci’, however, my understanding of the figure, based on the Methods section, is that it compares “quantification of the fraction of compact chromatin within high intensity MORC2/H3K9me3 region versus outside the IF-guided mask (nucleoplasm) in total”. Could the authors please clarify this in the text and figure legend.

We have clarified this point in new figure legend 4h and in the manuscript (lines 495–502).

The y-axis on the plots shows ‘FRET faction’, but I believe this should be ‘FRET fraction (% pixels)’

Thanks for noticing the typo. We have now added new Fig 4h to show the ratio of the

histone FRET fraction (indicative of compact chromatin) inside versus outside regions of high-intensity MORC2 or H3K9me3.

Reviewer #3 (Remarks to the Author):

Reviewer #4 (Remarks to the Author):

Thank you for providing the revised manuscript. Unfortunately, the new experiments do not provide the much-needed link between in vitro assays and cellular function of MORC2. There are several remaining issues, which needs to be at least dealt with in writing.

1) page 8, line 76: 'suggesting that ATP hydrolysis relaxes chromatin during HUSH-mediated silencing'

This statement doesn't make sense. Chromatin relaxation and silencing don't go together. Please rephrase: 'suggesting that ATP hydrolysis impairs HUSH-mediated silencing by promoting chromatin relaxation.'

Thanks for the suggestion. We have made that change.

2) 'These modifications lead to histone H2AX phosphorylation, which in turn relaxes chromatin to facilitate the DNA damage response.'

This is wrong. gH2AX doesn't lead to chromatin relaxation. Please rephrase: 'These modifications promote chromatin remodeling and DNA damage response.'

We have changed it on line 80.

3) Extended Data 4D: only one replicate? All figures: number of replicates should be indicated, data points should be shown.

ED 4d had two replicates with very small error bars (n=2) as stated in the figure legend ED4d. We have added the raw EMSA gels of the two independent experiments in source data.

4) Labels in main figures are messed up.

We have changed the labels to ensure consistency of font size in the main and ED figures.

5) 'MORC2 was previously shown to repress retrotransposons like LINE-1 marked by

H3K9me3 to initiate epigenetic silencing. Our data aligns with this finding...'
It does not.

We have changed this sentence to not claim that our data aligns with the finding about MORC2 repressing retrotransposons.

'Our findings highlight ATP hydrolysis-driven DNA compaction as a major function of MORC2, which is consistent with its role in gene silencing, presumably at promoters where it is localised.'

Again, your cellular data doesn't support this claim.

We agree with this observation and have deleted the sentence.

'The differences in gene expression and chromatin accessibility patterns between the wild-type, phosphodead and phosphomimetic mutants suggest that MORC2 phosphorylation plays a key role in chromatin structure regulation.'
Your phosphomutant reconstitution data does not show major effects on chromatin accessibility. Therefore, you cannot state that phosphorylation plays a key role in chromatin structure regulation.

We agree with this observation and have deleted the corresponding statements.

The new experiments did not demonstrate a functional importance of in vitro chromatin compaction, which is in contrast with previous findings that MORC2 is required for HUSH-mediated silencing of H3K9me3 regions (PMID: 28581500). This can be due to i) a different cell line, ii) lack of analysis of retroviral elements, iii) suboptimal assessment of chromatin accessibility (DIVA approach was shown to be superior to ATAC-seq for the analysis of repetitive regions in PMID: 28581500). Before acceptance of this manuscript, it is crucial that the authors comment on this in the Abstract and the Discussion by providing the above-mentioned possible reasons and by clearly stating that 'In contrast to previous work, in vitro chromatin compaction by MORC2 was not shown to have functional significance in cells with regard to chromatin accessibility and transcription regulation.'

We have incorporated the above suggestions in the Discussion section and explicitly mentioned in the Abstract that the DNA compaction we observed is in vitro.